# ERα-associated translocations underlie oncogene amplifications in breast cancer

Jake June-Koo Lee[1,2,3✉], Youngsook Lucy Jung[1,4,14], Taek-Chin Cheong[5,14], Jose Espejo Valle-Inclan[6], Chong Chu[1], Doga C. Gulhan[1,2], Viktor Ljungström[1], Hu Jin[1], Vinayak V. Viswanadham[1], Emma V. Watson[7,8,9], Isidro Cortés-Ciriano[6], Stephen J. Elledge[2,7,9,10], Roberto Chiarle[5,11], David Pellman[2,10,12,13] & Peter J. Park[1,2✉]

Focal copy-number amplification is an oncogenic event. Although recent studies have revealed the complex structure[1–3] and the evolutionary trajectories[4] of oncogene amplicons, their origin remains poorly understood. Here we show that focal amplifications in breast cancer frequently derive from a mechanism—which we term translocation–bridge amplification—involving inter-chromosomal translocations that lead to dicentric chromosome bridge formation and breakage. In 780 breast cancer genomes, we observe that focal amplifications are frequently connected to each other by inter-chromosomal translocations at their boundaries. Subsequent analysis indicates the following model: the oncogene neighbourhood is translocated in G1 creating a dicentric chromosome, the dicentric chromosome is replicated, and as dicentric sister chromosomes segregate during mitosis, a chromosome bridge is formed and then broken, with fragments often being circularized in extrachromosomal DNAs. This model explains the amplifications of key oncogenes, including *ERBB2* and *CCND1*. Recurrent amplification boundaries and rearrangement hotspots correlate with oestrogen receptor binding in breast cancer cells. Experimentally, oestrogen treatment induces DNA double-strand breaks in the oestrogen receptor target regions that are repaired by translocations, suggesting a role of oestrogen in generating the initial translocations. A pan-cancer analysis reveals tissue-specific biases in mechanisms initiating focal amplifications, with the breakage–fusion–bridge cycle prevalent in some and the translocation–bridge amplification in others, probably owing to the different timing of DNA break repair. Our results identify a common mode of oncogene amplification and propose oestrogen as its mechanistic origin in breast cancer.

Copy-number amplification is a common mode of oncogene activation in cancer[1]. In contrast to large-scale copy-number gains such as chromosome arm-scale aneuploidies, oncogene amplifications are frequently focal with high amplitude[5,6], suggesting distinct causal mechanisms. Previous work has established that cancer cells can take different evolutionary paths to acquire high-level copy-number amplification. In some cases, the oncogenes are linearly amplified after a single DNA double-strand break (DSB) through the breakage–fusion–bridge (BFB) cycle[2]—iterative cycles of chromosome breakage, DNA replication, sister chromatid fusion and dicentric chromosome bridge formation that results in another breakage. More recently, it was shown that high-level amplifications can also originate from chromothripsis, the phenomenon of massive chromosomal fragmentation and rearrangement, through

the formation of extrachromosomal circular DNAs[3,7–11] (ecDNAs). Chromothripsis and BFB cycles are often intertwined because chromothripsis can generate DNA breaks that initiate BFBs and the BFB cycle can precipitate chromothripsis[4,12,13]. Despite these advances, the initial mutational events leading to focal oncogene amplifications remain poorly understood.

Breast cancer is one of the cancer types in which the focal amplification of oncogenes has a crucial role in oncogenesis[14]. In many breast cancers, bona fide oncogenes such as *HER2* (also known as *ERBB2*) and cyclin D1 (*CCND1*) undergo focal amplification, defining clinically relevant subgroups[14,15]. These focal amplifications typically occur early in breast oncogenesis, probably contributing to the transition from atypical ductal hyperplasia to ductal carcinoma in situ[16,17]. Late

[1]Department of Biomedical Informatics, Harvard Medical School, Boston, MA, USA. [2]Ludwig Center at Harvard, Harvard Medical School, Boston, MA, USA. [3]Department of Medicine, Memorial Sloan Kettering Cancer Center, New York, NY, USA. [4]Division of Genetics and Genomics, Boston Children's Hospital, Boston, MA, USA. [5]Department of Pathology, Boston Children's Hospital and Harvard Medical School, Boston, MA, USA. [6]European Molecular Biology Laboratory, European Bioinformatics Institute, Hinxton, UK. [7]Department of Genetics, Harvard Medical School, Boston, MA, USA. [8]Department of Systems Biology, University of Massachusetts Chan Medical School, Worcester, MA, USA. [9]Division of Genetics, Department of Medicine, Brigham and Women's Hospital, Boston, MA, USA. [10]Howard Hughes Medical Institute, Chevy Chase, MD, USA. [11]Department of Molecular Biotechnology and Health Sciences, University of Torino, Torino, Italy. [12]Department of Cell Biology, Harvard Medical School, Boston, MA, USA. [13]Department of Pediatric Oncology, Dana-Farber Cancer Institute, Boston, MA, USA. [14]These authors contributed equally: Youngsook Lucy Jung, Taek-Chin Cheong. ✉e-mail: leej39@mskcc.org; peter_park@hms.harvard.edu

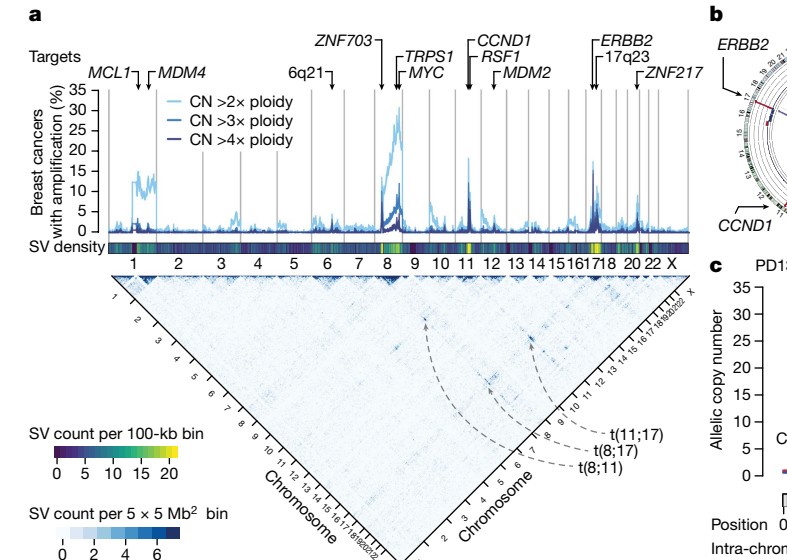

**Fig. 1 | Inter-chromosomal translocations frequently precede focal amplifications in breast cancer. a**, Copy-number profile and structural variations (SVs) in 780 breast cancers. The fraction of tumours containing amplified genomic regions with different copy-number thresholds (top) and frequencies of SVs connecting two genomic regions (bottom) are shown. CN, copy number **b**, Circos plots show the copy number and the SVs in three cases of breast cancer, which are positive for oestrogen receptor (ER) and progesterone receptor (PR) expression and overexpress HER2. **c**, The relationship between the amplified segments and the translocations in a representative breast cancer case harbouring the amplicons in chromosomes (chr.) 6 and 11. The number of supporting read fragments for the rearrangement is shown on the right *y*-axis. Brown lines and arcs indicate the SVs at the borders between the amplified and unamplified segments. Intra-chromosomal SVs are coloured on the basis of the orientation of their breakpoints.

emergence of focal amplifications during treatment has also been reported[18], suggesting their relationship to the on-going mutational processes in cancer cells. Despite the importance of focal amplifications in breast cancer, how the cell of origin acquires the amplicons and whether this process is associated with risk factors for breast cancer have remained unclear.

Here we identify a mechanism initiating focal amplification in breast cancer by analysing whole-genome sequencing (WGS), RNA sequencing and epigenomic data. Through bioinformatic analysis and experimental validation, we show that oestrogen-induced DNA breaks and their repair by inter-chromosomal translocations initiate a cascade of events leading to focal oncogene amplification.

## Translocations prior to focal amplifications

We analysed 780 breast cancer whole genomes (Supplementary Table 1) collected from 5 published studies[1,15,19–21]. This merged cohort represents a heterogeneous group of breast cancers in terms of age, histology and clinical subgroups (Extended Data Fig. 1a–c). All sequencing datasets were uniformly processed in a validated bioinformatic pipeline[22], with the variant calls showing excellent concordance with the consensus calls by the Pan-Cancer Analysis of Whole Genomes (PCAWG) consortium (Supplementary Fig. 1).

We first estimated the frequency of high-level amplifications and low-level copy gains across the genome in the cohort (Fig. 1a, top). The regions showing modest copy gains (2× to 3× ploidy) corresponded to the common arm-scale gains in breast cancer—for example, +8q and +1q. By contrast, the regions amplified to a higher degree (more than 4×) were focal and frequently encompassed oncogenes or master transcription regulators, including *ERBB2*, *CCND1*, *ZNF703*, *MYC* and *ZNF217*. Some of the focal amplicons included genes whose roles in breast cancer have not been clearly established. Integrative analyses of RNA sequencing and CRISPR screen datasets[23] pointed to additional putative oncogenes in these regions (Supplementary Figs. 2 and 3).

The focal amplifications were also rearrangement hotspots that were frequently connected to each other by inter-chromosomal translocations (Fig. 1a, bottom; hereafter, we use 'translocation' only to refer to inter-chromosomal rearrangement). For example, translocations between 11q and 17q were observed in 16 out of 25 breast cancers that have co-amplifications of *CCND1* and *ERBB2*, with some cases showing a remarkably similar pattern of copy number and rearrangements (Fig. 1b). There are two potential explanations for translocations connecting the amplicons. First, they could represent rearrangements that occur as a consequence of the amplification process, such as translocations from aborted BFB cycles or integration of ecDNAs to the chromosomes[4]. Second, these translocations could be causative events that give rise to the amplicons[24]. Notably, we found that some of these amplicon-connecting translocations were at the exact boundaries of the amplicons, demarcating the amplicons from the unamplified neighbourhood. As evidenced by a large number of supporting reads, these 'boundary translocations' were amplified as part of the amplicons (Fig. 1c), indicating that the translocations were formed before amplification[25], strongly favouring the second scenario.

By integrating copy number and structural variants (SVs), we identified 5,502 discrete amplified regions that are flanked by unamplified segments in the 780 genomes (506 had at least one focally amplified region) (Supplementary Table 2). We identified SVs demarcating the boundaries of amplicons in 80% of the cases (8,779 out of 11,004 boundaries), with 25% being translocations. Other boundary SVs were large-scale intra-chromosomal SVs (typically around 10 Mb in size) (Extended Data Fig. 1d). The boundary SVs were supported by larger numbers of reads compared with the other SVs, indicating that they preceded the amplifications (Extended Data Fig. 1e). The breakpoints of the boundary SVs showed less microhomology and were more frequently in the early-replicating segments and genic regions (Extended Data Fig. 1f–h). This suggests that the fragility of early-replicating regions and their rearrangements by non-homologous end-joining[26] have an important role in initiating the focal amplification.

The distribution of the boundary SVs showed the rearrangements of specific oncogene neighbourhoods before amplification (Fig. 2a). Of note, many cases had their early translocations between two regions

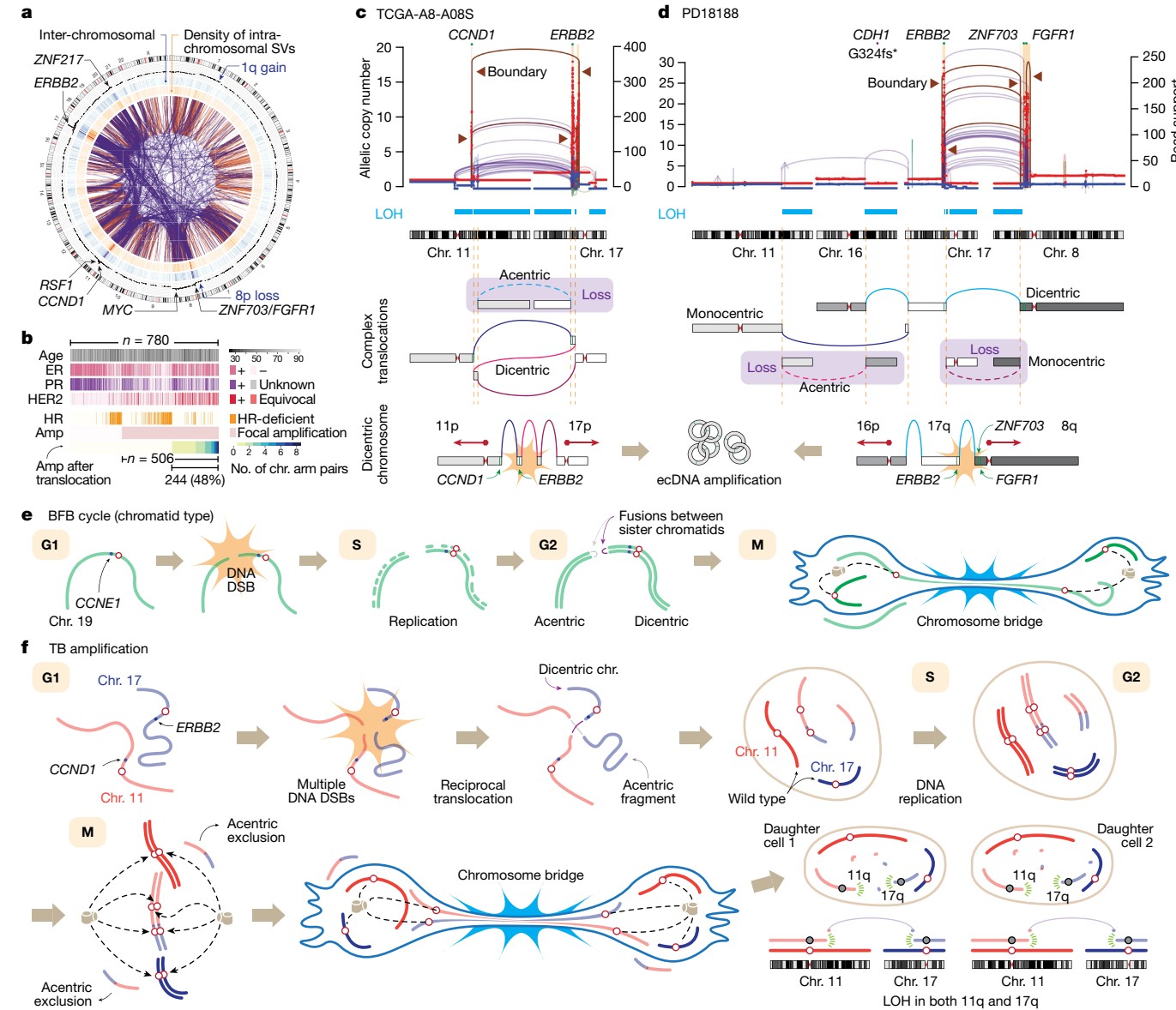

**Fig. 2 | Translocations lead to the amplification of breast cancer oncogenes via chromosome bridge formation. a**, SVs connecting the amplicon boundaries in 479 breast cancer cases with focal amplifications are overlaid in the centre (orange and purple arcs indicate intra- and inter-chromosomal SVs, respectively). The outermost track shows average copy numbers. **b**, Clinicopathologic and genomic features of 780 breast cancer cases and their associations with TB amplifications (Amp). Further details in Extended Data Fig. 2d. **c**, A representative case harbouring focal amplifications in *CCND1* and *ERBB2*. The boundary SVs border the amplified and unamplified segments. We reconstructed the initial rearrangement event on the basis of the boundary SVs or the SVs at the border of the segments showing LOH. Further details in Extended Data Fig. 3. **d**, Another case with focal co-amplifications after complex translocations. **e**, A schematic illustration of the classical BFB cycle (chromatid type). **f**, The TB amplification model. LOH is frequently observed in both arms involved in the TB amplification (**c**,**d**). This dual-LOH pattern indicates that the initial translocation happened in the G1 phase (further details in Extended Data Fig. 5c). During mitosis, the replicated, two 'flipped' sister dicentric chromosomes form the chromosome bridge.

that both contained oncogenes. For example, chromosomes 17q (which contains *ERBB2* and 17q23), 11q (*CCND1*, *RSF1* and *PAK1*), 8p (*ZNF703* and *FGFR1*), 8q (*MYC*) and 20q (*ZNF217*) were frequently involved in such translocations. These cases often displayed co-amplification of the oncogenes from different chromosomes. In addition to the reported translocations causing 11q–8p co-amplicons[27], our analysis identified many other pairs, confirming that translocation before amplification of oncogenes is pervasive in breast cancer. We found that 31% of all breast cancer cases (244 out of 780) and 48% of those with focal amplification (244 out of 506) had their amplicons formed after translocations (Fig. 2b). These tumours were frequently associated with clinical HER2 positivity and HER2-enriched or luminal B expression phenotypes. Notably, these breast cancers rarely showed the genomic footprints of

defective homologous recombination (HR). Among the HR-proficient tumours, the pattern of amplifications and their boundary translocations were similar between the ER⁺ and ER⁻ subgroups (Extended Data Fig. 2a–c). Furthermore, *PTEN* deletions were significantly depleted in the tumours with the amplifications subsequent to translocations (odds ratio, 0.15; 95% confidence interval, 0.0017–0.15; Extended Data Fig. 2d).

## The translocation–bridge amplification model

To understand the cellular consequences of the early translocations, we reconstructed the complex rearrangements for relatively simple cases. TCGA-A8-A08S is a triple-positive (for ER, PR and HER2) invasive

ductal carcinoma from a 71-year-old woman (Fig. 2c, top), and its genome showed multiple focally amplified segments in 11q and 17q, encompassing *CCND1* and *ERBB2*, respectively. We found a massively amplified boundary translocation connecting the telomeric border of *CCND1* amplicon and the centromeric border of the 17q amplicon along with another amplified translocation connecting the other two borders, reflecting an ecDNA structure containing both oncogenes (Extended Data Fig. 3). By integrating the SVs at the major copy-number junctions, we were able to reconstruct the original derivative chromosome with two centromeres. This dicentric chromosome was formed by complex translocations between 11q and 17q, juxtaposing the two telomeric neighbourhoods of *ERBB2* and *CCND1* (Fig. 2c, bottom). The dicentric chromosome then led to a chromosome bridge formation during mitosis, the segment between the centromeres was fragmented[12], and the DNA fragments containing the oncogenes formed ecDNAs. Known fragility and recombination of ecDNAs explain the dense, unamplified internal rearrangements[11]. The segments showing loss of heterozygosity (LOH) distal to the amplicons were acentric and probably lost owing to missegregation during mitosis[13]. Another case with a similar pattern involving 11q and 17q is illustrated in Supplementary Fig. 5.

PD18188 is also a triple-positive, invasive ductal carcinoma from a 75-year-old woman (Fig. 2d, top) with focal co-amplifications involving *ERBB2*, *ZNF703* and *FGFR1*. We found several highly amplified SVs, including the boundary translocation connecting the telomeric borders of the 17q and the 8p amplicons. A sharp copy-number transition with numerous unamplified internal rearrangements indicated an ecDNA formation. In this case, the initial rearrangement event is a set of chain-like translocations involving four chromosomes, reminiscent of chromoplexy[28,29] (Fig. 2d, bottom). The original chain was initially resolved into a dicentric and a monocentric chromosome. Subsequently, the dicentric formed a chromosome bridge, whose breakage and resolution led to the ecDNA amplification. The four large telomere-bound regions were lost, causing LOH.

In contrast to the classic chromatid-type BFB model (Fig. 2e), it is the inter-chromosomal translocation that directly creates the dicentric chromosome in the examples described above. We refer to this mechanism as 'translocation–bridge' (TB) amplification (Fig. 2f). Tumours with TB amplification typically show co-amplifications and adjacent LOH segments (often subtelomeric) on the affected chromosome arms (we refer to these arms as 'bridge arms'; Extended Data Fig. 4a), whereas the arms on the opposite side of the centromere ('non-bridge arms') are generally spared from rearrangements (Extended Data Fig. 4b–d). This asymmetric footprint strongly favours the TB amplification model rather than an alternative explanation by multi-chromosomal chromothripsis, which is more likely to cause symmetric and widespread complex rearrangements affecting both arms of the chromosomes[7] (Extended Data Fig. 4e, f). The breakage of translocation-induced chromosome bridge could initiate BFB cycles (similar to chromosome-type BFB described in corn zygotes[30]) if the breakpoint is repaired by fold-back inversion with the replicated sister chromatid. However, this pattern is less frequent in breast cancer genomes. Instead, the locally fragmented bridge segments are often ligated together and undergo ecDNA formation in TB amplification, leading to co-amplifications of different chromosomes.

The pattern of LOH segments provides critical information about the timing of initial translocations in the cell cycle as well as the mechanism of dicentric resolution in TB amplification. In many cases, both bridge arms exhibited telomeric LOHs (Figs. 1c and 2c,d and Extended Data Fig. 5a). These LOHs are typically large (affecting around 50% of the bridge arm on average) (Extended Data Fig. 4d) and contiguous, consistent with simultaneous generation of a dicentric chromosome and acentric fragments. To generate daughter cells with the observed 'dual-LOH' pattern and retention of heterozygosity in both non-bridge arms, it requires replicated dicentric sisters (Fig. 2f). The pattern is not

consistent with the expected copy-number outcome from the chromosome bridge formation by a single dicentric chromosome (Extended Data Fig. 5b). In our TB amplification model, the two replicated dicentric sisters align in an anti-parallel orientation on the bridge, as previously proposed in cases of acute lymphoblastic leukaemia[31] and an in vitro model of bridge formation[12]. If two kinetochores of a dicentric attach to the microtubules emanating from opposing poles of the mitotic spindle (in *trans*), the two dicentric sisters will be pulled away to the two daughter cells in an anti-parallel orientation within the bridge (alternative scenarios are further discussed in Extended Data Fig. 5c). Through bridge breakage between the centromeres of sister dicentrics, each of the two daughter cells will inherit two broken hemi-dicentrics (one with intact 11p and the other with intact 17p in Fig. 2f), retaining heterozygosity of these segments and causing the dual-LOH pattern in the bridge arms. We also frequently observed large segmental gains or losses affecting the bridge arms (for example, Extended Data Figs. 3 and 5a). This can be explained by asymmetry of the breakpoints of the two sister dicentrics[12]. Notably, the initial translocation has to happen before DNA replication—that is, in G1 phase—in the TB model (Extended Data Fig. 5d). This differs from the chromatid-type BFB model in which fusion between the sister chromatids occurs after DNA replication (Fig. 2e).

Stabilization of broken hemi-dicentrics will be essential for the survival of daughter cells. We found cases where non-bridge arms were coordinately gained to similar copy-number levels (Extended Data Fig. 5e). In these cases, the two hemi-dicentrics may have been stabilized by mutual ligation and subsequent amplification. Although such ligation would produce another dicentric chromosome, some cases showed focal copy-number losses in the peri-centromeric region (Fig. 2c and Extended Data Fig. 5f), which could generate monocentric chromosomes from the original dicentrics[32].

Together, the strong consistency between the TB model and the various features of the data presents compelling evidence that the oncogene amplifications in breast cancer frequently originate from TB amplification.

## ERα-associated breaks and translocations

To understand the origin of the initial breakage events that led to TB amplification, we explored molecular correlates of the recurrent focal amplification boundaries using epigenomic features in breast cancer cell lines[33] (Supplementary Table 3 and Extended Data Fig. 6a–c). In a multivariate LASSO regression model, we found that chromatin binding of oestrogen receptor-α (ERα) after oestradiol (E2) treatment (E2–ERα) was the best predictor for the location of the amplification boundary hotspots, followed by CTCF, topoisomerase 2B and DNase I hypersensitivity regions (Fig. 3a). Our modelling based on mixed-effect linear regression also showed a similar result (Extended Data Fig. 6d,e). E2–ERα was of particular interest because previous studies have shown that E2-induced ERα binding promotes DNA DSBs at adjacent loci[34,35]. Notably, the telomeric neighbourhood of *ERBB2* (around *RARA*), where the amplification boundaries were frequently located, exhibited a E2–ERα binding peak that is an order of magnitude stronger than the other peaks in the genome (Fig. 3b). The presence of amplification boundaries associated with E2–ERα was dependent on the ER status of the tumours, with a strong correlation in the ER+ subgroup but not in the ER− subgroup (Fig. 3a) that included many tumours with HR deficiency (126 of 271 ER− tumours; 46%). In addition, genomic regions that frequently overlap the amplification boundaries showed more intense E2–ERα binding and were more proximal to prominent E2–ERα peaks (Extended Data Fig. 6f,g). We assessed whether this correlation is specific to the amplification boundaries or if E2–ERα is generally associated with SVs genome-wide. Supporting the latter, SV breakpoints, including those in unamplified regions, were generally concentrated around E2–ERα peaks (Fig. 3c and

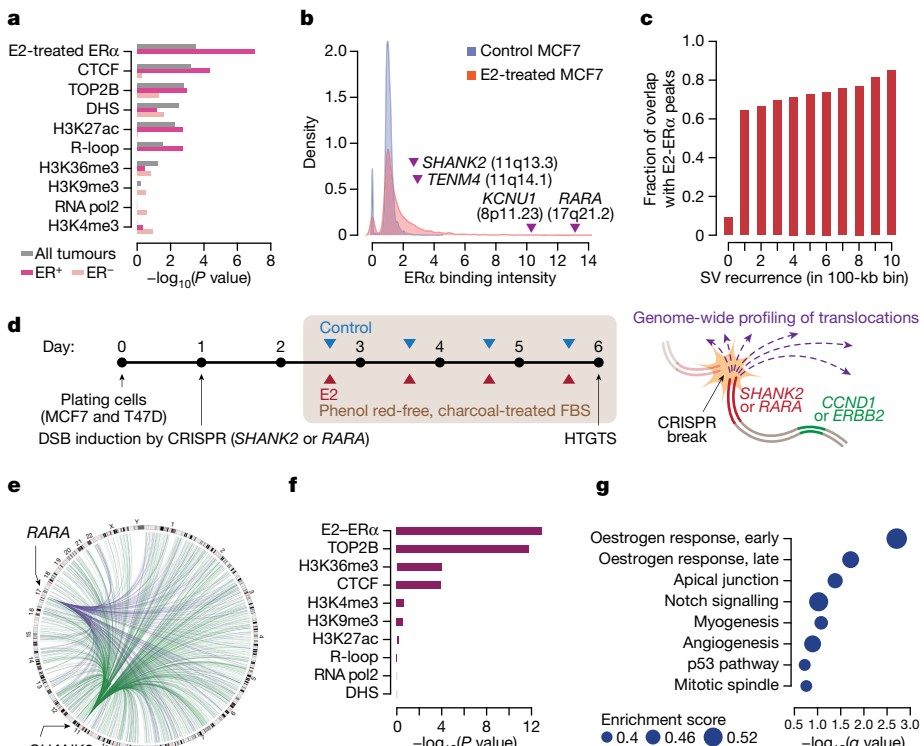

**Fig. 3 | ERα-associated genomic fragility underlies TB amplifications in breast cancer. a**, Association between the amplification boundaries and the epigenetic features from the breast cancer cell lines using multivariate LASSO regression. The analysis is based on 100-kb genome-wide bins. Raw *P* values from a two-sided test are shown. DHS, DNase I hypersensitivity sites. **b**, Density distribution of ERα binding in control versus E2-treated experiments in the MCF7 cells. Genomic loci and related genes are annotated based on their values in the E2-treated experiment. **c**, The degree of overlap between the unamplified SV hotspots and E2–ERα binding. Statistical significance was based on a linear regression among the regions with SVs (*r* = 0.98 by Pearson's correlation; two-sided test, *P* = 2.7 × 10$^{-7}$). **d**, Schematic illustration of the experiments. FBS, fetal bovine serum. **e**, Hotspots of E2-induced translocations (*n* = 1,012 hotspots, more than fourfold increase following E2 treatment) between the induced breaks (the green arcs for those in *SHANK2* intron 10 and the purple arcs for those in *RARA* intron 1) and the prey regions in the MCF7 cells. **f**, Multivariate LASSO regression model for the HTGTS translocation breakpoints. Raw *P* values from two-sided tests. **g**, The gene sets enriched in the E2-induced HTGTS breakpoint hotspots. *q* values provide the estimated false discovery rate (FDR).

Extended Data Fig. 6h,i). This pattern was present in the HR-proficient group but not in the HR-deficient group (Extended Data Fig. 6j). This raises an interesting possibility of E2-induced, ERα-associated fragility in the breast cancer genomes and its role in triggering TB amplification.

To experimentally determine whether the E2 treatment could induce DNA breaks in the ERα binding regions and whether the breaks could be repaired by translocations, we performed high-throughput genome-wide translocation sequencing[36] (HTGTS), a method for genome-wide profiling of DNA DSBs that are translocated to an induced CRISPR–Cas9 break. In two ER$^+$ breast cancer cell lines (MCF7 and T47D), we generated a DSB in intron 10 of *SHANK2* or intron 1 of *RARA* in separate experiments (Fig. 3d and Extended Data Fig. 7a). We selected these loci because their neighbourhoods frequently contained the telomeric boundaries of *CCND1* and *ERBB2* amplicons, respectively, and encompassed prominent E2–ERα peaks. Cells were incubated in oestrogen-depleted media and were treated with E2 or vehicle control for three days before being harvested for HTGTS library preparation. Robust upregulation of the oestrogen-responsive genes by E2 was confirmed by quantitative PCR with reverse transcription (RT–qPCR) (Extended Data Fig. 7b).

We observed an increased number of HTGTS breakpoints after E2 treatment in all experiments (Extended Data Fig. 7c), consistent with the known effect of E2 causing DNA DSBs[34,35]. As expected, the HTGTS breakpoints were enriched in genic regions compared with intergenic regions in both control and E2-treated experiments[36] (odds ratios, 2.37 and 2.41; 95% confidence intervals, 2.34–2.40 and 2.38–2.44,

respectively; Extended Data Fig. 7d). To understand the mechanisms of E2-induced translocations, we modelled the ratio of the HTGTS breakpoints between the E2-treated and the control experiments across the genome (Fig. 3e and Extended Data Fig. 7e) in terms of the epigenomic features using LASSO regression. E2–ERα was indeed the best predictor in this model (Fig. 3f), confirming that the E2 treatment conferred fragility near the E2–ERα peaks. The second-best correlate was topoisomerase 2B binding, suggesting the mechanistic role of topoisomerase 2B in the E2-induced, ERα-associated fragility, as shown in a previous study[34]. Next, we characterized the genes frequently targeted by the E2-induced HTGTS breakpoints. Gene set enrichment analysis (GSEA) showed that the early and late oestrogen-responsive genes were the two most enriched gene sets[37] (Fig. 3g and Supplementary Table 4). For example, we found that the E2 treatment substantially increased DSBs near *RARA* (one of the E2 target genes) and they were frequently repaired by translocations with the CRISPR–Cas9 break in *SHANK2* (Extended Data Fig. 8a), consistent with our model that this translocation initiates TB amplification involving *CCND1* and *ERBB2*. We also observed increased *RARA*–*SHANK2* translocations when we induced the break in *RARA* (Extended Data Fig. 8b). Some recurrent SVs in unamplified regions—for example, rearrangements involving *GATA3* or *FOXA1*—also overlapped with the HTGTS hotspots near the E2–ERα peaks, suggesting a similar mechanism (Extended Data Fig. 8c,d).

Together, the SV hotspots in breast cancer, notably the amplification boundaries, are associated with E2–ERα, which causes DSBs. Repairing these breaks by translocations can initiate TB amplification.

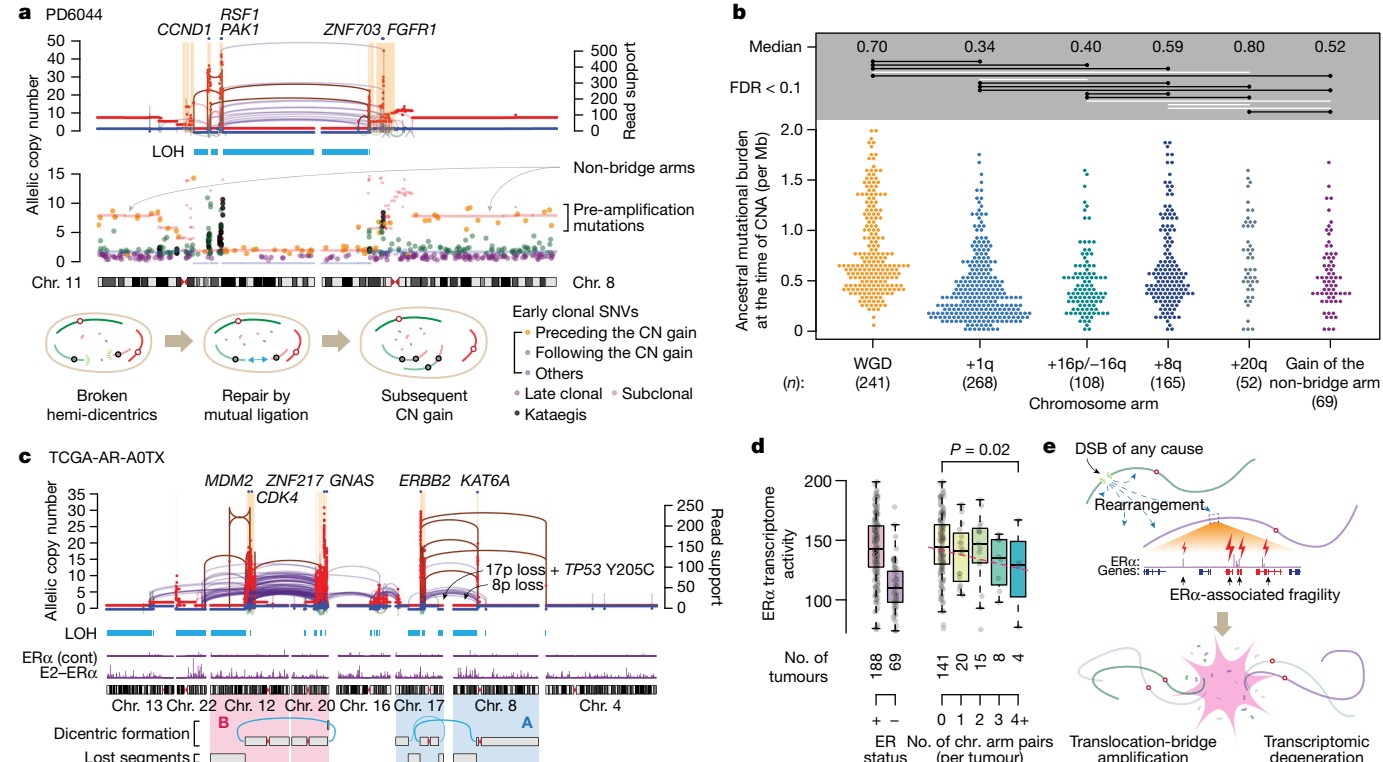

**Fig. 4 | Timing and transcriptional effect of TB amplification. a**, Top, a case with TB amplification showing both non-bridge arms amplified to the same copy-number level. Middle, SNVs were plotted based on their location and copy number. Their colour codes indicate the classification based on timing. Bottom, schematic of the evolution after the bridge breakage in this case. **b**, Timing of frequent copy-number gains in 780 breast cancers, based on the estimated tumour mutation burden (per diploid genome) at the timing of copy-number gains. Two-sided Wilcoxon test. Raw *P* values from top to bottom: $8.2 \times 10^{-27}$, $4.9 \times 10^{-14}$, 0.0026, 0.78, $6.6 \times 10^{-6}$, 0.052, $1.3 \times 10^{-11}$, $6.5 \times 10^{-8}$, 0.00070, $1.1 \times 10^{-5}$, $1.1 \times 10^{-5}$, 0.050, 0.15, 0.053 and 0.0043. The comparisons

with FDR <0.1 are annotated as black horizontal lines. White lines indicate non-significant comparisons. CNA, copy number alteration. **c**, A case with two rounds of TB amplification. The A (involving chromosomes 17, 8, and 4) and B (12, 20, and others) rounds of TB amplifications form two discrete clusters of complex genomic rearrangements without exchanging translocations to each other. **d**, The activity of ERα-medicated transcriptome from RNA sequencing. Box plots indicate median (middle line), first and third quartiles (edges) and 1.5× the interquartile range (whiskers). Statistical significance was assessed by linear regression. **e**, A schematic illustration of the consequence of ERα-associated fragility.

## Timing and effect of TB amplification

Next, we inferred the timing of TB amplification during breast cancer evolution by estimating the burden of mutations preceding the amplifications[29,38]. Because of the small size of the amplicons and the paucity of pre-amplification mutations, direct timing of TB amplification is challenging. We therefore used an indirect approach of estimating the timing of bridge breakage from the cases with global copy gain in the non-bridge arms (Fig. 4a and Extended Data Fig. 9a). These non-bridge arms were probably amplified after bridge breakage when the hemi-dicentrics were separated from the original amplicon (Fig. 2f). Therefore, the timing of non-bridge arm gains enables us to infer the latest possible time of bridge breakage. Analysing 28% (69 out of 244) of cases with TB amplification and gain of non-bridge arms, we found that the gains occurred when the ancestral cell had accumulated a median single-nucleotide variant (SNV) burden of 0.52 per Mb. If we assume a gradual accumulation of clonal mutations at a rate of 29 SNVs per year, which we inferred from a select set of tumours without hypermutation or whole-genome duplication (WGD) (Extended Data Fig. 9b), the median timing of non-bridge arm gain in the 69 cases corresponds to 51 years, which is slightly less than the median age of menopause[39] (approximately 52.5 years). This suggests that a majority of bridge breakage happened in reproductive ages, in line with our experimental evidence of E2-induced DNA DSBs.

We also compared the timing of non-bridge arm gains to the timing of common arm-level copy-number events (Fig. 4b). We found that +1q

and −16q (from the cases with paired +16p, likely creating an isochromosome 16p) were the earliest common events[40], occurring after a median of 0.34 and 0.40 per Mb, respectively, and predating the other three common aneuploidies, +8q (0.59 per Mb), WGD (0.70 per Mb) and +20q (0.80 per Mb). We thus infer that, on average, TB amplification took place later than +1q, but earlier than WGD and +20q (Fig. 4b).

TB amplification events were variable in their complexity, ranging from the simplest two-chromosome cases to those involving multiple chromosomes. Some showed evidence of multiple rounds of TB amplifications. For example, a triple-positive tumour (TCGA-AR-A0TX) showed two separate bundles of translocations, one between 8p and 17q, and another between 12q and 20q (Fig. 4c). The absence of intermingled translocations between these two sets of chromosomes indicates that their fragmentation likely occurred at different time points (Extended Data Fig. 9c). By contrast, cases with such intermingling translocations suggested all-at-once process involving multiple chromosomes (Extended Data Fig. 9d).

Extensive TB amplification had substantial effects on the transcriptome. Notably, the ER⁺ breast cancer cases with more extensive TB amplification events were associated with reduced ERα-transcriptome activity, as measured by our score based on the list of oestrogen-responsive genes[37] (Fig. 4d). Although the expression of ERα was unaffected, several oestrogen-responsive genes showed significantly decreased expression in the tumours with extensive TB amplifications (Extended Data Fig. 9e). Notably, one of the few upregulated genes in the tumours with extensive TB amplifications was *ASCL1*,

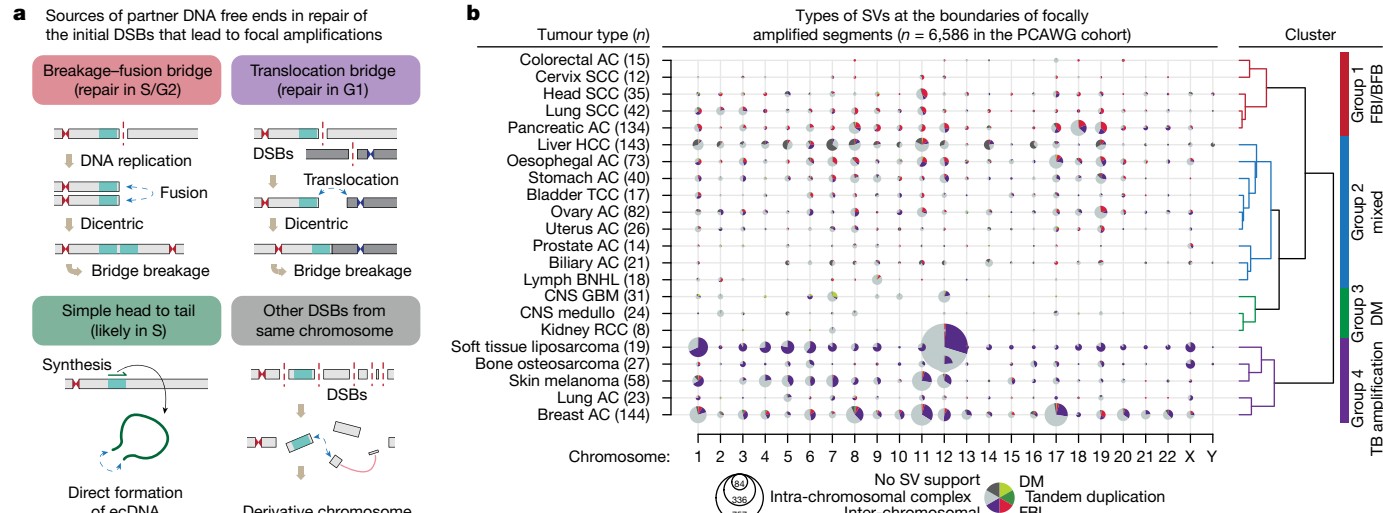

**a** Sources of partner DNA free ends in repair of the initial DSBs that lead to focal amplifications

**b** Types of SVs at the boundaries of focally amplified segments (n = 6,586 in the PCAWG cohort)

**Fig. 5 | Patterns of focal amplifications between cancer types reflect their preferential mode of DNA break repair. a**, Classification of the SVs at the amplification boundaries and their associated mechanisms. **b**, The pattern of SVs at the amplification boundaries in different tumour types. The size of the circle indicates the number of amplified segments, as guided by the concentric circles and numbers below the plot. Tumour types are grouped by hierarchical clustering using the boundary SVs in all chromosomes. AC, adenocarcinoma; BNHL, B cell non-Hodgkin lymphoma; CNS, central nervous system; DM, double minutes; FBI, fold-back inversion; GBM, glioblastoma; HCC, hepatocellular carcinoma; medullo, medulloblastoma; RCC, renal cell carcinoma; SCC, squamous cell carcinoma; TCC, transitional cell cancer.

the transcriptional regulator of neuroendocrine lineage, suggesting an altered differentiation state. Overall, these data suggest that TB amplification disrupts the neighbourhood of oestrogen-responsive genes, leading to degradation of the ERα-driven transcriptional activity (Fig. 4e).

## Different mechanisms across cancer types

Finally, we investigated the mechanisms of focal amplification in other tumour types. We re-analysed more than 2,600 samples across 38 tumour types from PCAWG[1], focusing on SVs located at amplicon boundaries (Supplementary Table 5). The expanded set of oncogenes in this cohort were amplified in more diverse patterns than in the breast-only cohort (Extended Data Fig. 10a,b). We classified the boundary SVs into four categories based on how the initial DNA break was repaired (Fig. 5a): (1) fold-back inversions, which can result from BFBs; (2) TB amplifications; (3) focal amplicons generated by simple self-ligation of an unrearranged DNA segment with one intra-chromosomal SV connecting the head (−) and tail (+) of the segment, forming an ecDNA; and (4) intra-chromosomal SVs that were not included in the other categories, mostly complex rearrangements generated by chromothripsis.

Hierarchical clustering of all tumour types on the basis of the categories of boundary SVs identified four major groups (Fig. 5b). Group 1 was enriched for fold-back inversions and included head and neck, lung squamous cell carcinomas and pancreatic adenocarcinomas. Group 3 often contained simple ecDNA amplicons in high copy numbers and included glioblastoma and medulloblastoma. Group 4 was enriched for inter-chromosomal translocations indicative of TB amplifications and included liposarcomas, melanomas, and adenocarcinomas of the lung and breast. Group 2 showed mixed features of Groups 1 and 4.

The distribution of these classes of events suggested a tissue type-specific, rather than locus-specific, bias for the underlying rearrangement process. For example, *CCND1* amplifications in head and neck squamous cell carcinomas appear to be driven primarily by BFB-mediated amplification (Extended Data Fig. 10b) rather than by TB amplifications as observed in breast cancer. *EGFR* amplifications (7p) in glioblastoma were frequently generated from head-to-tail SVs that

formed ecDNAs[41,42] (Extended Data Fig. 10c), in contrast to the common BFB-type amplification of *EGFR* in oesophageal adenocarcinomas.

Some Group 4 tumour types, especially liposarcomas and acral melanomas, showed highly complex patterns of translocations involving multiple chromosomes[43]. These tumours probably share the mechanistic basis of TB amplification, given that many translocations are highly amplified at their amplicon boundaries. We found a marginal trend towards more frequent TB amplification among female patients compared to male patients (odds ratio, 1.14; 95% confidence interval, 1.01–1.29; Extended Data Fig. 10d) in a merged analysis excluding cancers of female- or male-specific organs. TB amplification was not associated with *ESR1* expression, age or survival in most cancer types, although this analysis had limited power due to the small number of patients (Extended Data Fig. 10e–g).

Collectively, this analysis determined that TB amplification is frequent in multiple tumour types, and there are tissue type-specific patterns of SVs at the initial steps of focal oncogene amplifications.

## Discussion

Exposure to oestrogen—affected by the timing and duration of menstruation, pregnancy and exogenous replacement—is a major risk factor for breast cancer development[44]. Accordingly, pharmacologic inhibition of oestrogen has effectively reduced breast cancer incidence among high-risk individuals[45,46]. These clinical observations have been frequently attributed to the role of oestrogen as the master transcriptional regulator of mammary epithelial tissues, promoting their proliferation and preventing apoptosis. Our study suggests that, additionally, oestrogen has a direct effect on genome structure, contributing to oncogenesis through TB amplification.

The initial rearrangements forming the dicentric chromosome in breast cancer cases were often complex, reminiscent of chromoplexy in prostate cancers[28]. Association between the androgen receptor binding and the DNA breaks causing *TMPRSS2–ERG* fusion, which frequently exists as part of a chromoplexy chain, has been proposed previously[28,47,48]. If prostate cancers share a similar mechanistic background with breast cancers, TB amplification would be expected to be common in prostate cancers, but this is not the case. We speculate

that this reflects differences in the timing of events. *TMPRSS2–ERG* is one of the earliest genomic events in prostate oncogenesis[48]. In this early oncogenesis with nearly intact DNA repair system, extreme copy-number events such as focal amplifications would be subject to negative selection. Accordingly, we found some prostate cancer cases showing a genomic footprint of chromosome bridge formation but without focal amplifications. In these cases, the broken bridge would have been stabilized without causing high-level amplification (Extended Data Fig. 10h). By contrast, TB amplification in breast cancer appears to occur in already-aneuploid breast cells, often harbouring driver point mutations[49]; furthermore, E2–ERα antagonizes the p53-induced apoptosis in breast cancer cells[50]. These events could provide a permissive cellular environment for focal oncogene amplifications.

Our pan-cancer analysis indicates that TB amplification is common in other cancer types with a modestly increased frequency in women. Whether this indicates a pervasive genomic effect of oestrogen beyond breast cancer is unclear. Notably, we found that DNase I hypersensitivity was the best predictor of amplification boundary hotspots in ER− breast cancers, and this may provide clues for the origin of TB amplification in other cancer types. Some oestrogen-responsive genes have vital roles in many other tissue types (for example, *CCND1* and *RARA*), and their chromatin features could confer local fragility.

Some tissues favour DSB repair with sister chromatids, leading to classical BFB, whereas others prefer inter-chromosomal translocation as the initial step for focal amplification. We propose that this tissue type-specific bias could be owing to the differences in the cell cycle phase in the timing of the DNA end-joining events for chromosomal breakages. Fusion with sister chromatid will be favoured if a chromosome is broken in G1 but the break ends are not resolved until G2, where the replicated break ends will typically be ligated to each other because they are held in close proximity by chromatid cohesion. Notably, multiple fold-back inversions are present in the amplicons for the Group 1 tumours (Extended Data Fig. 10b), suggesting iterative cycles of BFB. By contrast, if DSBs occur in a breast cancer cell in G1 owing to an oestrogen-mediated mechanism, these breaks are probably resolved by inter-chromosomal translocation prior to the initiation of DNA replication. After DNA replication, sister dicentric chromosomes can generate ecDNAs, either through the fragmentation of bridge segments and their direct circularization or through chromothripsis of hemi-dicentrics in the next cell cycle[12].

In summary, we identified TB amplification as a mutational process underlying focal oncogene amplification that is particularly important in breast cancer. Although the conventional BFB model involving sister chromatid fusions has been studied extensively, we find that inter-chromosomal translocation is the most frequent source of bridges in multiple cancer types, including breast cancer. Our findings extend a growing body of work implicating oestrogen-induced DNA breaks as an important driver of breast oncogenesis.

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

# Methods

## Patient cohort

We merged five breast cancer WGS datasets, downloaded from public repositories: (1) 208 cases from the PCAWG consortium[1]; (2) 72 cases from the International Cancer Genome Consortium (ICGC) French cohort[19]; (3) 395 cases from the Sanger cohort[15] (among the original 560, 108 and 47 were included in the PCAWG and ICGC French cohort studies, respectively; we were able to download 395 of the remaining 405); (4) 87 cases from the British Columbia cohort[20] (among the original 93, 5 cases that were sequenced from formalin-fixed paraffin-embedded tissues were excluded, 1 could not be downloaded); and (5) 20 cases from the Yale study[21]. Two cases were excluded owing to poor data quality. This established our 780-patient cohort for detailed analysis (Supplementary Table 1). The institutional review board of the Harvard Faculty of Medicine approved this study (IRB18-0151). Individual studies complied required ethical guidelines per published manuscripts.

## Uniform data processing and identification of variants

To remove any potential artefacts that may arise from different data processing and analysis steps for different cohorts, we re-processed all data and applied a uniform set of variant calling methods. We used Bazam (v1.0.1)[51] to extract FASTQ files from the BAM or CRAM files and realigned the reads to hs37d5 (as done in PCAWG) using BWA-MEM (v0.7.15)[52]. We used Samtools (v1.3.1)[53] to merge the realigned bam fragments and Picard (v2.8.0) to add read groups and to mark PCR duplicates.

We applied the Hartwig Medical Foundation (HMF) bioinformatics pipeline[22] for our analysis (https://github.com/hartwigmedical/hmftools), as it provides a streamlined software suite for analysing multiple variant types including SNVs, indels, SVs and allele-specific CNVs. We chose this pipeline because, in their PURPLE algorithm (v2.54), the boundaries of copy-number segments were determined by jointly analysing regional depth of coverage (COBALT v1.11), B-allele frequency (AMBER v3.5), and, most importantly, SVs. This integration resulted in near-complete concordance between the rearrangement breakpoints and the copy-number boundaries, which was pivotal in analysing the SVs at the amplification boundaries. SVs were called primarily by GRIDSS2 (ref. 54) (v2.12.0), annotated with RepeatMasker (v4.1.2-p1) and Kraken2 (ref. 55) (v2.1.2), filtered by GRIPSS (v1.9), and further annotated and analysed with LINX[56] (v1.15). SNVs and indels were primarily called by SAGE (v2.8) with recommended parameters for 30x tumour coverage. Tumours showing genomic features of HR deficiency were identified using CHORD[57] (v2.00).

Based on our benchmark analysis (Supplementary Note), we applied an in-house filter for short, non-reciprocal, and singleton inversions for samples that showed a large number of such probably artefactual patterns. We also filtered out somatic L1 transduction events that originated from the 18 hot source retroelements detected from 279 breast cancers (PCAWG + Ferrari et al. study[19]) using xTea[58] (v0.1.6; Supplementary Note and Supplementary Table 6).

## Defining focal amplifications

We defined an amplicon as a genomic segment for which the absolute copy number was more than three times greater than the baseline copy number of the chromosome arm. The arm-level baseline was defined as the integer copy number supported by the largest of the combined genomic segments sharing the same copy number. For the chromosome arms where the most common copy number was haploid, we considered diploid as the baseline copy number. Contiguous amplicons were merged, and the amplicons less than 1 kb in size were removed. From the 780 breast cancer cases, we identified 11,490 amplicons. Among these, we focused on 5,502 (48%) amplicons that bordered an unamplified region (Supplementary Table 2). The unamplified regions were defined as those where the copy number was no greater than one

above the arm-level baseline. We included regions where the copy number was one copy above the arm-level baseline, because the unequal breakpoints in the two flipped dicentric chromosomes during the TB breakage could result in duplicated genomic segments.

For pan-cancer analysis, we also applied the same steps to the PCAWG consensus calls to define amplicons. Due to the incomplete concordance between copy-number boundaries and SV breakpoints, we created additional copy-number segments on the consensus copy-number calls: we divided a copy-number segment into two when there was an SV breakpoint in the middle, at least 10 bp apart from both boundaries; next, the absolute copy number was re-calculated for each genomic segment from BAM files considering depth, GC content and mappability. Based on this analysis, we focused on 6,586 amplicons adjacent to the unamplified regions.

The SVs at their boundaries were classified into six categories (fold-back inversion, translocation, simple head-to-tail, tandem duplication, intra-chromosomal complex, and no SV support) as described in Fig. 5b. Fold-back inversion were defined as head-to-head or tail-to-tail intra-chromosomal SV with breakpoints less than 5 kb apart. Among the amplicons generated by simple head-to-tail SVs (duplication-like), double minutes (DMs) were defined as amplicons with copy number more than three times greater than that of adjacent segment; while those with a copy number three times or less were classified as tandem duplications. Other amplicons bordered by intra-chromosomal SVs were classified as intra-chromosomal complex rearrangements, which often resulted from chromothripsis. Amplicons without supporting boundary SVs were grouped as 'no SV support'. Hierarchical clustering of tumour types was performed using their fraction of fold-back inversion, translocation, and DMs at the amplicon boundaries. The optimal number of clusters was determined to be $k = 4$ due to the distinct differences observed among the groups.

## Correlative analysis with TB amplification

Among the 244 breast cancer cases with TB amplification, many displayed the genomic footprints of TB amplification and chromothripsis at the same time. Some cases showed a heavier burden of intra-chromosomal rearrangements than of boundary translocations, suggesting a predominant role of chromothripsis in these cases. To conduct a correlative analysis between TB amplification and driver genomic events, we selected 151 (out of 244) cases exhibiting an extensive footprint of TB amplification with 10 or more translocations between the involved chromosomes. We used the potential driver genetic alterations identified by the PURPLE algorithm, which included recurrently altered genes by mutation ($n = 363$), germline alterations ($n = 15$) and deletions ($n = 124$). We excluded gene amplifications ($n = 127$) here because many of them were TB amplification. We selected the top 10% of genes in each class and examined their presence in the tumours with and without extensive footprint of TB amplification. We excluded the genetic alterations present in less than 5% of the samples (39 cases). Primary statistical testing was performed by the two-sided Fisher's exact test with FDR <0.1.

## Reconstruction of complex genomic rearrangements

Complex genomic rearrangements were reconstructed as described[29]. Given the higher structural complexity of the amplicons compared to that of fusion oncogenes, we focused on the SVs at the borders of LOH segments as well as on the amplified SVs, which are more likely to have occurred earlier than the SVs on the already-amplified segments. The amplified SVs were defined based on the abundance of supporting read fragments with respect to a tumour-specific threshold. To determine the threshold, we sorted all SVs in each tumour by the number of supporting read fragments and chose an inflection point, beyond which the increase in the number of supporting reads changed markedly.

To reconstruct the rearrangements, we first connected the chromosomal regions through the amplified SVs. The unamplified SVs within

the amplicons were excluded due to their late timing (probably after the ecDNA formation). Then, the SVs outside of the amplicons were connected to finalize the most likely ancestral karyotype. For visualization (for example, Fig. 1c), we plotted the allelic absolute copy number as well as the SVs (vertical lines and connecting arcs) with their number of supporting read fragments (available in the PURPLE output), which provides the relative timing information within the amplified regions. On these plots, we often displayed chromosomes in their flipped orientations (p arm on the right and q arm on the left) for easier mechanistic interpretation. To avoid overlap between the major- and the minor-allele copy-number segments, we subtracted 0.2 from the minor-allele copy numbers. We shaded amplicon regions with orange colour and annotated key oncogenes on top.

## Clustering structural variations
We used LINX[56] (v1.15) to identify genomic rearrangement clusters. In brief, LINX uses multiple additional criteria to group SVs into clusters other than breakpoint proximity, including: SVs that are phased by a deletion bridge or an LOH segment, translocations connecting two chromosome arms in common, all fold-back inversions on a chromosome arm, and others. In 780 breast cancers, we identified 1,556 complex genomic rearrangement clusters with 10 or more SVs involved. Among these, 295 clusters (found in 245 samples) involved multiple chromosomes and contained boundary translocations, which are the key features of TB amplification (Extended Data Fig. 4b). On average, these clusters contained 137 SVs (range: 10–1,515) and 3.75 boundary translocations (range: 1–33). Fusion genes were analysed as part of this step, and the result was discussed in Supplementary Note (Supplementary Fig. 6).

## Analysis of mutational signatures
We calculated the mutational spectra of SNVs and indels using SigProfilerMatrixGenerator[59] (v0.1.0) with the SBS-96 and ID-83 classification system. We performed de novo extraction of the mutational signatures, matched them to the reference catalogue, and refitted and validated them using MuSiCal[60] (v1.0.0-beta). We used an expanded SBS and ID signatures catalogue described in the MuSiCal manuscript. One ID signature did not match to the ID catalogue but had high similarity to a recently described ID signature commonly observed in people with African ancestry[61]. We also separately analysed the mutational signatures near the SV breakpoints (Supplementary Fig. 4). In this analysis, the observed SBS mutational spectra were linearly decomposed using SBS1, SBS2, SBS5, SBS13 and SBS18.

## Mutational timing analysis
We analysed the timing of copy-number gains for genomic segments larger than 5 Mb. Two approaches were used. First, we used relative timing, the temporal order among the different copy number-gained segments. This could be estimated by the ratio of amplified versus unamplified mutations in each amplified segment, using Mutation-TimeR algorithm[38] (v1.00.2). Synchronous copy-number gain events were determined using the code accompanying a previous publication[38] (available at https://gerstung-lab.github.io/PCAWG-11/). Second, we calculated the absolute mutation burden of the ancestral cell at the moment of copy-number gains. We quantified the number of mutations amplified up to the maximal major copy number of the non-bridge arms from the cases with TB amplification. A mutation was assumed to be a pre-amplification event when the estimated copy number of the mutation was larger than the major copy number of the locus times 0.75. If the estimated copy number was smaller, the mutation was classified as post-amplification or on the minor allele. The probabilities of being pre-amplification, post-amplification or on the minor allele, and subclonal mutation were estimated using the binomial distribution as previously described[29]. We used this approach in estimating the absolute timing of common aneuploidies, including gain of 1q, 8q,

16p (in those cases with a paired 16q loss) and 20q and whole-genome duplication.

Assuming a stable mutation rate in early oncogenesis, we estimated the timing of non-bridge arms and common arm-level copy-number gains. The clonal mutation burden increases with age at a rate of 29.4 mutations per year in our selected cases ($n = 147$; purity ≥0.6, number of SNVs <10,000, fraction of SBS2 + SBS13 <0.5, no whole-genome duplication, microsatellite stable, and HR proficient; Extended Data Fig. 9b). We divided the median ancestral mutation burden by this rate to estimate the typical age when the common aneuploidy events occurred. When we repeated this analysis using total SNV counts (which may overestimate the rate of accumulation due to the inclusion of all subclonal mutations), we found a rate of 33.1 mutations per year (Extended Data Fig. 9b).

## RNA analysis
We used the RNA-sequencing data from ref. 15 to quantify the activity of the ER-driven transcriptome. For the 263 samples for which the data were available, we studied the expression of the frequently amplified genes with respect to their amplification status. The findings were validated in the METABRIC cohort[14], using their diploid samples ($n = 1,904$) to minimize the impact of whole-genome duplication. We defined the ER target genes based on the Hallmark gene sets in MSigDB[37]. The list of 275 genes in the early and late oestrogen-responsive set included well-known ER target genes such as *GREB1*, *TFF1* and *PGR*. We tested if these genes were differentially expressed between the ER$^+$ and the ER$^-$ groups ($n = 188$ and 69, respectively, with 6 ER-unknown cases). Using the 136 genes showing a significantly higher level of RNA expression in the ER$^+$ group, we determined the ER activity of each tumour, calculating the fraction of genes that had an expression level of 50th percentile or higher. We tested different percentile cutoff values, and the score based on the 50th percentile showed the best separation between the ER$^+$ and ER$^-$ cases and a good spread within the ER$^+$ cases (Fig. 4e).

## Integration of the CRISPR screen information
To study the functional importance of the amplified genes, we integrated CRISPR screen data from the DepMap project[23]. Of the 46 breast cancer cell lines studied, ER and HER2 status were available for 41 cell lines. We used the gene effect score as the readout for cellular dependence on a given gene. (0 indicated no viability effect on cells by knockout of the gene; −1 indicated the median cytotoxic effect observed by knockout of common essential genes[23]). We compared gene effect score among the putative target genes in the amplicons (Supplementary Note).

## Integration of the epigenomic data
We used epigenomic profiles from the ENCODE[33] and Roadmap Epigenomics[62] consortia (accession numbers and further details are provided in Supplementary Table 3). When MACS (v2)[63]-processed output was not available, we downloaded FASTQ files from GEO and aligned the reads to hg19 using Bowtie (v1.2.2.)[64] with the unique mapping option. For generating input-normalized ChIP enrichment tracks and detecting significant peaks, we used MACS2 callpeak and bdgcmp functions with the q-value threshold of 0.01.

To quantify the relative enrichment of an epigenetic feature with respect to amplicon boundaries, we first identified all 100-kb bins that overlap the epigenetic feature, and then compared the number of bins that overlap versus not overlap amplicon boundaries. The significance was calculated using the one-sided Fisher's exact test unless otherwise specified.

To find associations between the distribution of amplicon boundaries and epigenomic variables, we used the multivariate LASSO regression model, which is more tolerant to the multicollinearity between the variables compared to other linear regression models. A multivariate linear mixed-effect model also supported the conclusion. We evaluated

multicollinearity among the epigenetic features by calculating the variance inflation factor (VIF), a standard method for quantifying collinearity between the dependent variables. We considered VIF >5 as concerning and >10 as serious collinearity issues[65].

### Analysis of three-dimensional chromatin contact
We explored the relationship between the chromatin contact frequencies and the chromosomal regions frequently involved in TB amplification (Supplementary Fig. 8). For the comparison of chromatin interactions between the E2-treated and untreated conditions, we used chromatin conformation capture-based high-throughput sequencing data in untreated- and E2-treated MCF7 cells[66]. The contact frequencies were combined for each chromosome arm-pair and then compared between the E2-treated and untreated conditions. We also analysed Hi-C data from T47D luminal breast cancer cell line from 4D Nucleome Data Portal (https://data.4dnucleome.org). Contact frequencies were normalized by balance-based method (KR normalization) using Juicer[67] to reduce the effects from possible copy-number variations.

### Cell lines and cultures
MCF7 (ATCC, HTB-22) and T47D (ATCC, HTB-133) cells were maintained in RPMI 1640 medium (Corning, 15-040-CV) supplemented with 10% fetal bovine serum (FBS; Gibco, 10437-028), 100 U ml$^{-1}$ penicillin-streptomycin (Corning, 30-002-CI), and 2 mM L-glutamine (Corning, 25-005-CI). For RT–qPCR, cells were cultured in RPMI 1640 medium without phenol red (Corning, 17-105-CV) supplemented with 10% charcoal- and dextran-treated FBS (R&D System, S11650H) and either (1) 0.01% ethanol or (2) 1 µM β-estradiol (Sigma Aldrich, E2758) for 4 days with fresh β-estradiol every 24 h. For the HTGTS experiment, cells were plated in RPMI 1640 medium (Corning, 15-040-CV) supplemented with 10% FBS (Gibco, 10437-028), 100 U ml$^{-1}$ penicillin-streptomycin, and 2 mM L-glutamine and were transduced with CRISPR–Cas9-containing lentiviral supernatants targeting SHANK intron 10 or RARA intron 1 with 6 µg/ml polybrene. Thirty hours after the lentiviral infection, cells were washed with phosphate-buffered saline (PBS) three times and cultured as described above for RT–qPCR. Cells were collected for the HTGTS library preparation on day 6. 293FT (Invitrogen/ThermoFisher Scientific, R70007) cells were used to produce CRISPR/Cas9-containing lentiviral particles and were maintained in DMEM medium (Corning, 15-017-CV) supplemented with 10% FBS, 100 U ml$^{-1}$ penicillin-streptomycin, and 2 mM L-glutamine. All cell lines were tested negative for mycoplasma contamination and were cultured at 37 °C in 5% CO2 atmosphere.

Genomic DNA (gDNA) was extracted from MCF7 and T47D cells using rapid lysis buffer (100 mM Tris-HCl pH 8.0, 200 mM NaCl, 5 mM EDTA, 0.2% SDS) containing 10 µg ml$^{-1}$ proteinase K (P2308, Sigma Aldrich). After overnight incubation at 56 °C, gDNA was precipitated in one volume isopropanol, and the DNA pellet was resuspended in Tris-EDTA buffer. gDNA was used for preparation of the HTGTS library.

Total RNA was isolated from the cells using Rneasy Plus Mini Kit (Qiagen, 74136). cDNA was synthesized using iScript cDNA synthesis kit (Bio-Rad, 1708891). All RT–qPCR experiments were performed in triplicate on Icycler iQ Real-Time PCR Detection System (Bio-Rad) with iTaq universal SYBR green supermix (Bio-Rad, 1725121). Expression levels for individual transcripts were normalized against ACTB. Primers for RT–qPCR are listed in Supplementary Table 7.

### Lentiviral particle productions
To produce lentiviral particles, $5.5 \times 10^6$ 293FT cells were plated in a 10 cm dish a day before the transfection. On the following day, cells were transfected using Xfect transfection reagent (Takara Bio, 631318) with 20 µg of lentiCRISPR–Cas9 plasmid, 3.6 µg of pMD2.G plasmid (Addgene, 12259), 3.6 µg of pRSV-Rev plasmid (Addgene, 12253) and 3.6 µg of pMDLg/pRRE plasmid (Addgene, 12251). The medium was changed with complete culture medium 6 h after transfection. The viral supernatant was collected 48 h post-transfection, passed through a 0.45-µm syringe filter (PVDF membrane; VWR, 89414-902), pooled, and used either fresh or snap frozen.

### CRISPR–Cas9 sgRNA design and cloning
For SpCas9 expression and generation of single guide RNA (sgRNA), the 20-nt target sequences were selected to precede a 5′-NGG protospacer-adjacent motif (PAM) sequence. The human SHANK2 intron 10-targeting sgRNA and human RARA intron 1-targeting sgRNA were designed with the CRISPR design tool CRISPick (https://portals.broadinstitute.org/gppx/crispick/public). Oligonucleotides synthesized by Integrated DNA technology were annealed and cloned into the BsmbI–BsmbI sites downstream from the human U6 promoter in lentiCRISPR v2 plasmid (Addgene, 52961). sgRNA sequences were confirmed by Sanger sequencing with U6 promoter primer 5′-GAGGGCCTATTTCCCATGAT-3′. Oligonucleotides for sgRNA cloning are listed in Supplementary Table 7.

### High-throughput genome-wide translocation sequencing
HTGTS libraries were generated by the emulsion-mediated PCR (EM-PCR) methods as previously described[36]. In brief, gDNA was digested with HaeIII enzyme (New England Biolabs, R0108) overnight. HaeIII-digested blunt ends were A-tailed with Klenow fragment (3′→5′ exo-; New England Biolabs, M0212). An asymmetric adaptor (composed of an upper liner and a lower 3′-modified linker; Supplementary Table 7) was then ligated to fragmented DNA. To remove the unrearranged endogenous SHANK2 and RARA locus, ligation reactions were digested with XbaI (New England Biolabs, R0145L) for SHANK2 locus and EcoRI (New England Biolabs, R0101L) for RARA locus, respectively. In the first round of PCR, DNA was amplified using an adaptor-specific forward primer and a biotinylated reverse SHANK2 primer oriented to capture the 5′ portion of SHANK2 junction and using a biotinylated forward RARA primer and an adaptor-specific reverse primer with Phusion High-Fidelity DNA polymerase (ThermoFisher Scientific, F530S). Twenty cycles of PCR were performed in the following conditions: 98 °C for 10 s, 58 °C for 30 s, and 72 °C for 30 s. Biotinylated PCR products were enriched using the Dynabeads MyOne streptavidin C1 (ThermoFisher Scientific, 65002), followed by an additional digestion with blocking enzymes for 2 h. Biotinylated PCR products were eluted from the beads by 30-min incubation with 95% formamide/10mM EDTA at 65 °C, and purified using Gel Extraction Kit (Qiagen, 2870). In the second round of PCR, the purified products were amplified with EM-PCR in an oil-surfactant mixture. The emulsion mixture was divided into individual aliquots and PCR was performed using the following conditions: 20 cycles of 94 °C for 30 s, 60 °C for 30 s, and 72 °C for 1 min. The PCR products were pooled and centrifuged for 5 min at 14,000 rpm to separate the PCR product-containing phase and the oil layer. The layer was removed and the PCR products were extracted with diethyl ether three times. EM-PCR amplicons were purified using the Gel Extraction Kit. The third round of PCR (10 cycles) was performed with the same primers as in the second round of PCR, but with the addition of linkers and barcodes for Illumina Mi-seq sequencing. The third round PCR products were size-fractionated for DNA fragments between 300 and 1,000 base pairs on a 1% agarose gel (Bio-Rad, 1613102). The PCR products containing Illumina barcodes were extracted with the Gel Extraction Kit.

The HTGTS libraries were sequenced on Mi-seq (Illumina NS500 PE250) at the Molecular Biology Core Facility of the Dana-Farber Cancer Institute. The libraries were generated from each of the three biological replicate experiments and analysed for each experimental condition. Oligonucleotide primers used for SHANK2 and RARA library preparations are listed in Supplementary Table 7.

The HTGTS data were processed and aligned as previously described[68]. In brief, the reads for each experimental condition were

demultiplexed by designed barcodes. To enhance the specificity and ensure that the analysed sequences contain the bait portion, reads were further filtered by the presence of primer sequence and additional five downstream bases. After the filtering, barcode, primer, and bait portions of the reads were masked for alignment. Then, the processed reads were aligned to GRCh37/hg19 using BLAT. We removed PCR duplicates (reads with same junction position in alignment to the reference genome and a start position in the read less than 3 bp apart), invalid alignments (including alignment scores < 30, reads with multiple alignments having a score difference <4 and alignments having 10-nucleotide gaps), and ligation artefacts (for example, random HaeIII restriction sites ligated to bait break site). The position of HTGTS breakpoints (often referred to as 'junctions' in previous publications[36,68]) were determined based on the genomic position of the 5' end of the aligned read.

Due to the universal increase of the HTGTS breakpoints by the E2 treatment in all biological replicates and in both cell lines (Extended Data Fig. 7c), we primarily analysed breakpoint ratios in genomic bins between the E2-treated group and the control group (for absolute counts, the normalization-based approach used in previous HTGTS experiments[68] negated the effect of the E2 treatment, as discussed in ref. 69). Thus, to investigate the mechanisms underlying the E2-induced translocations, we modelled the ratio of the HTGTS breakpoints (E2-treated/control) in genome-wide bins (250 kb) using multivariate LASSO regression. We used the same epigenomic datasets that were used in the modelling of the amplicon boundaries. In addition, we performed GSEA to study the gene sets enriched in the E2-induced HTGTS breakpoint hotspots. For this analysis, we calculated per-gene HTGTS breakpoint ratios (included ±5 kb upstream and downstream of the gene) and averaged the ratios from four experimental pairs (MCF7/*SHANK2*, T47D/*SHANK2*, MCF7/*RARA* and T47D/*RARA*) after excluding the bait region (±1 Mb from the CRISPR target site). Based on the ordered list of all genes, a pre-ranked GSEA was performed using the GSEA application (v4.2.3).

## Statistics and reproducibility

The statistical tests or methods are described in the figure legends. We used R (v4.1.1) for all data processing and secondary computational analysis. For the HTGTS, we performed three biologically independent experiments per group (defined by cell line and CRISPR targets) as specified in the figure legends.

## Reporting summary

Further information on research design is available in the Nature Portfolio Reporting Summary linked to this article.

## Data availability

WGS datasets generated through ICGC or PCAWG consortium are available at the ICGC Data Portal (download instructions and links available in the downloading PCAWG data section; https://docs.icgc.org/pcawg/data/). The other WGS data are available from European Genome-phenome Archive (EGA; https://www.ebi.ac.uk/ega/) with the following accession numbers: Ferrari et al.[19], EGAS00001001431; Nik-Zainal et al.[15], EGAD00001001334, EGAD00001001335, EGAD00001001336, EGAD00001001338 and EGAD00001001322; Zhao et al.[20], EGAS00001001159; and Li et al.[21], EGAS00001004117. The HTGTS dataset is available in Gene Expression Omnibus (GEO) under the accession number GSE227369. Epigenomic datasets are available at Gene Expression Omnibus (http://www.ncbi.nlm.nih.gov/geo), 4D Nucleome Data Portal (http://data.4dnucleome.org), and other repositories under the accession numbers provided in Supplementary Table 3. MSigDB gene set collections are available from GSEA (http://www.gsea-msigdb.org/gsea/downloads.jsp). Somatic variant calls, including SNVs, indels, SVs and allelic copy-number information for 780 breast cancer cases are available from the Park laboratory website (http://compbio.med.harvard.edu/TBAmplification/).

## Code availability

The codes used in this study are publicly available at https://github.com/parklab/focal-amplification. Source code for genomic event analysis tool (GEAT; v0.1), which was developed in the Chiarle laboratory to perform the HTGTS analysis is available at https://github.com/geatools/GEAT.

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

**Acknowledgements** This work was funded by grants from Ludwig Center at Harvard (J.J.-K.L., S.J.E., D.P. and P.J.P.), Cancer Research UK Grand Challenge and the Mark Foundation to the SPECIFICANCER team (J.J.-K.L., E.V.W., S.J.E. and P.J.P.), National Institutes of Health R01CA222598 (R.C.), T15LM007092 (V.V.V.), F31CA264958 (V.V.V.), R01CA213404 (D.P.), Office of Faculty Development/CTREC/BTREC Career Development Fellowship (T.-C.C), EMBL (J.E.V.-I. and I.C.-C.), and Howard Hughes Medical Institute (S.J.E. and D.P.). We thank A. Viari, E. Pleasance and L. Pusztai for guiding data access, M. Brown and Z. Sandusky for comments, and Q. Wang and A. Tran for bioinformatic assistance. We gratefully acknowledge the contributions of the many clinical networks across ICGC, TCGA and other consortia, who provided samples and data. We thank the patients and their families for their participation in the individual projects.

**Author contributions** J.J.-K.L. conceived the study and led all analysis. P.J.P. supervised overall study design and analysis. J.J.-K.L., D.P. and P.J.P. wrote the manuscript with input from Y.L.J., T.-C.C. and S.J.E. J.J.-K.L., J.E.V.-I., V.V.V. and I.C.-C. performed structural variation analysis. J.J.-K.L., Y.L.J. and C.C. performed copy-number analysis. Y.L.J. performed epigenome analysis. T.-C.C. performed HTGTS experiments. D.C.G. and H.J. performed mutational signature analysis. J.J.-K.L. and V.L. performed timing analysis. E.V.W. performed gene expression analysis. J.J.-K.L., Y.L.J., T.-C.C., E.V.W., S.J.E., R.C., D.P. and P.J.P. interpreted the data. All authors read and approved the final version of this manuscript.

**Competing interests** D.P. is a member of the Volastra Therapeutics scientific advisory board.

**Additional information**
**Correspondence and requests for materials** should be addressed to Jake June-Koo Lee or Peter J. Park.

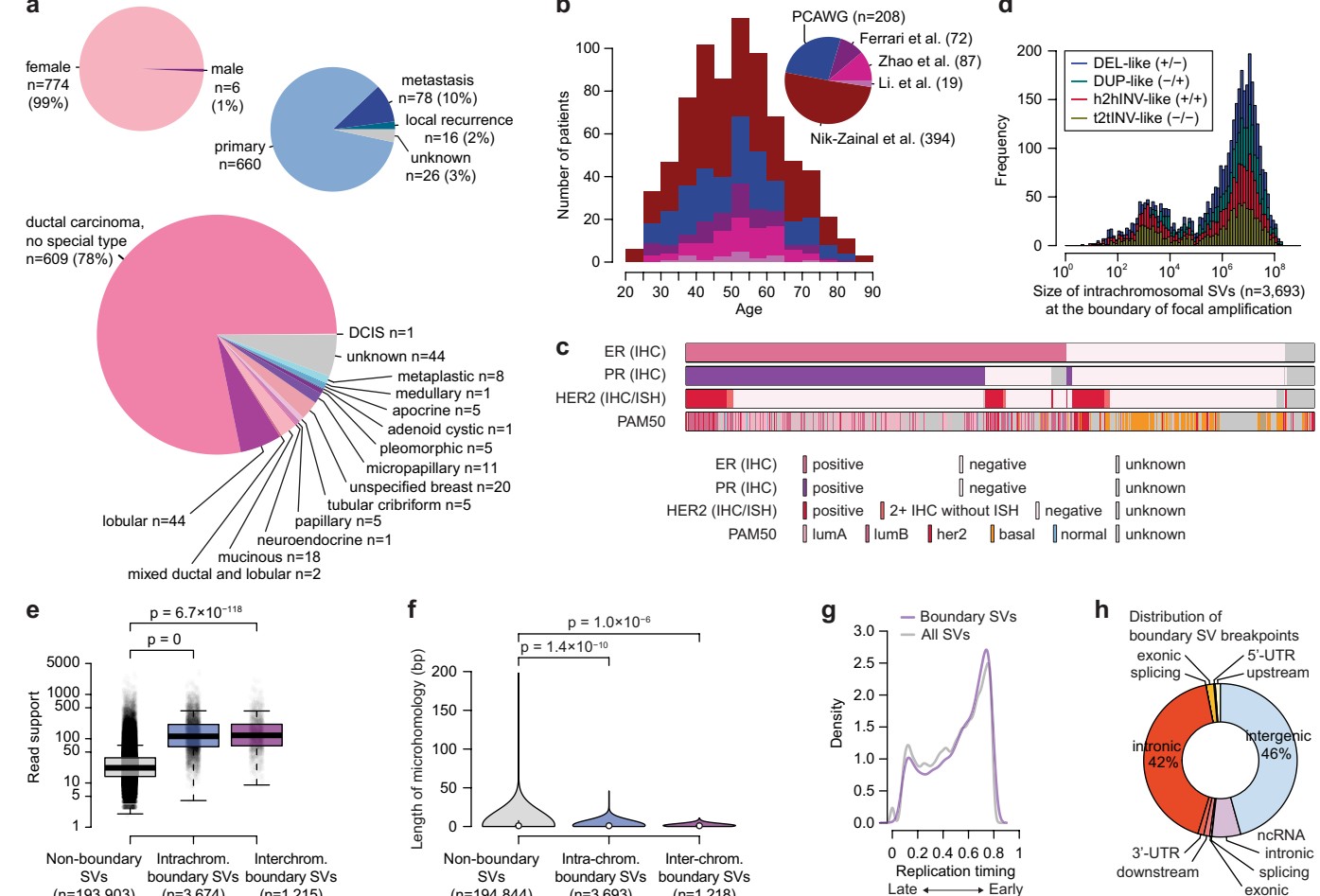

**Extended Data Fig. 1 | Clinicopathologic characteristics of 780 breast cancers and genomic features of structural variations at the amplification boundaries. a**. Patients' sex, disease site, and primary diagnosis in pathology. **b**. Distribution of patients' age at diagnosis. Number of patients in parenthesis. **c**. Estrogen receptor, progesterone receptor, and HER2 status by pathology, and gene expression (PAM50)-based subtype. IHC, immunohistochemical staining; ISH, in situ hybridization; PAM50, prediction analysis of microarray 50 (Methods). **d**. A stacked histogram of intra-chromosomal boundary SVs based on their size. A large fraction of them are large-sized (~10 Mbp) intra-chromosomal SVs with all four different types contributed equally. A peak around the size of ~1 Kbp comprises fold-back inversions, the genomic

signature of chromatid-type breakage-fusion-bridge cycles. **e**. SVs at the boundaries of amplified segments are supported by a significantly larger number of reads than those not at the boundaries. Comparisons were by a two-sided, two-sample *t* test. Box plots indicate the median (thick line), the first and third quartiles (edges), and 1.5x of the interquartile range (whiskers). **f**. Length of microhomology sequences at the breakpoint of the SVs. Statistical comparisons were made by two-sided Wilcoxon's test. Mutational features around the SV breakpoints are further discussed in Supplementary Note. **g**. A density histogram of SVs in relation to their replication timing zone. **h**. Genomic annotation of the boundary SV breakpoints.

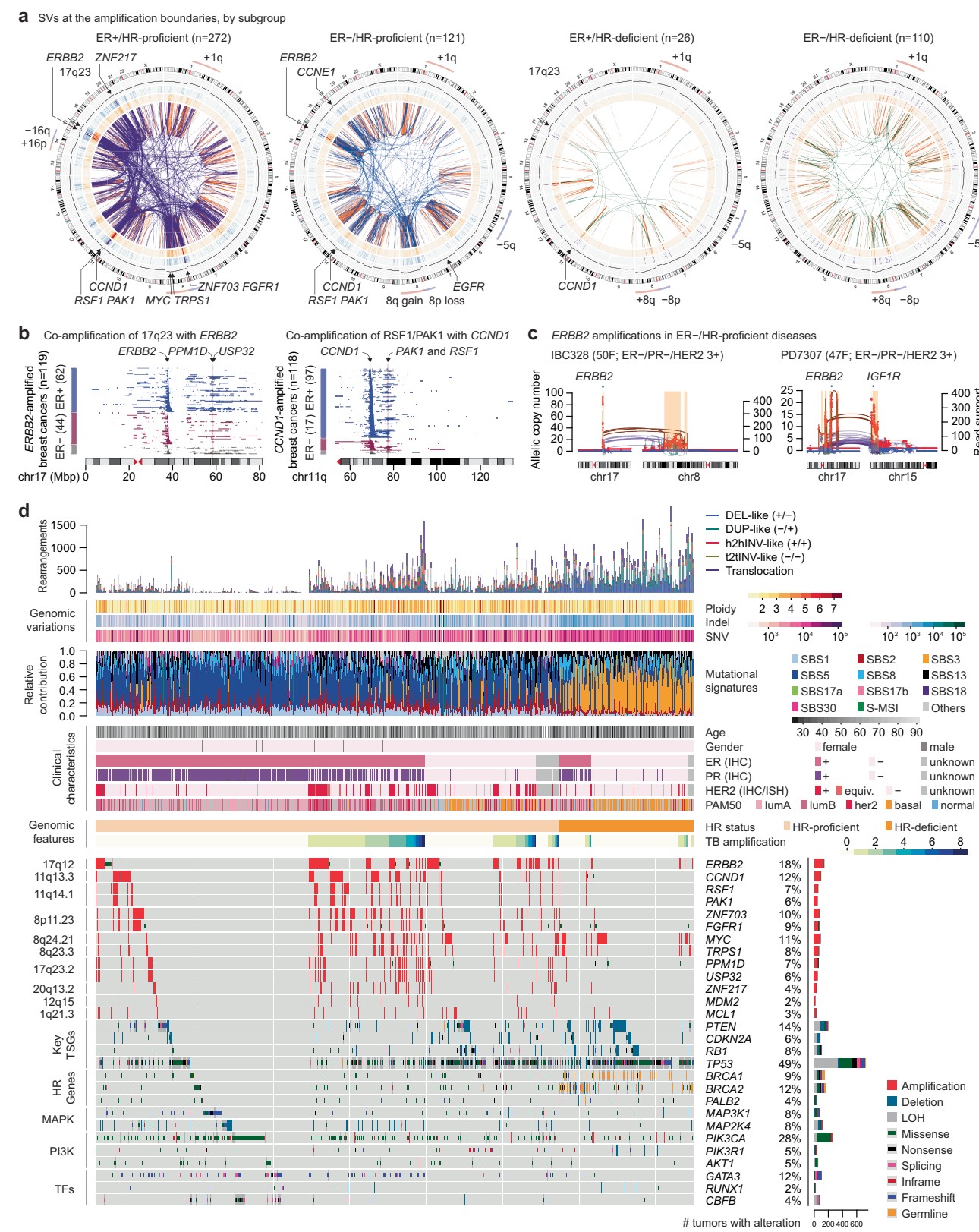

**Extended Data Fig. 2** | See next page for caption.

**Extended Data Fig. 2 | Genomic alterations in 780 breast cancers and their relationship with translocation-bridge amplification. a**. Landscape of boundary SVs by the ER and homologous recombination (HR) status. Frequently amplified genes and common aneuploidies are annotated. **b**. Amplified genomic segments in *ERBB2*-amplified tumors by the ER status (left panel). *ERBB2* and two oncogenes in 17q23 are annotated. Co-amplification of the 17q23 region (harboring *PPM1D*, *MIR21*, *USP32*, etc.) was numerically more frequent in the HER2+/ER+ group compared to the HER2+/ER− but without statistical significance (37% vs 18%; odds ratio = 2.63; 95% CI = 0.98−7.71 by two-sided Fisher's exact test). Amplified genomic regions in *CCND1*-amplified tumors by the ER status (right panel). *CCND1*, *PAK1*, and *RSF1* are shown as shaded areas. *CCND1* amplification was more common in the ER+ group compared to the ER− (21% vs 6%; odds ratio=3.85; 95% CI=2.22−7.05 by two-sided Fisher's exact test). **c**. Representative cases of *ERBB2* amplification in ER−/HR-proficient breast cancers (ER−/HER2+). The pattern of copy-number amplification with frequent translocations at their boundaries is similar to what was observed in the ER+/HER2+ cases. **d**. Whole-genome alteration landscape of 780 breast cancer cases shows the relationship between the translocation-bridge amplifications and driver genetic alterations. IHC, immunohistochemical stain; ISH, in situ hybridization; HR, homologous recombination; TSG, tumor suppressor gene; MAPK, mitogen-activated protein kinase; PI3K, phosphatidylinositol-3-kinase; TF, transcription factor.

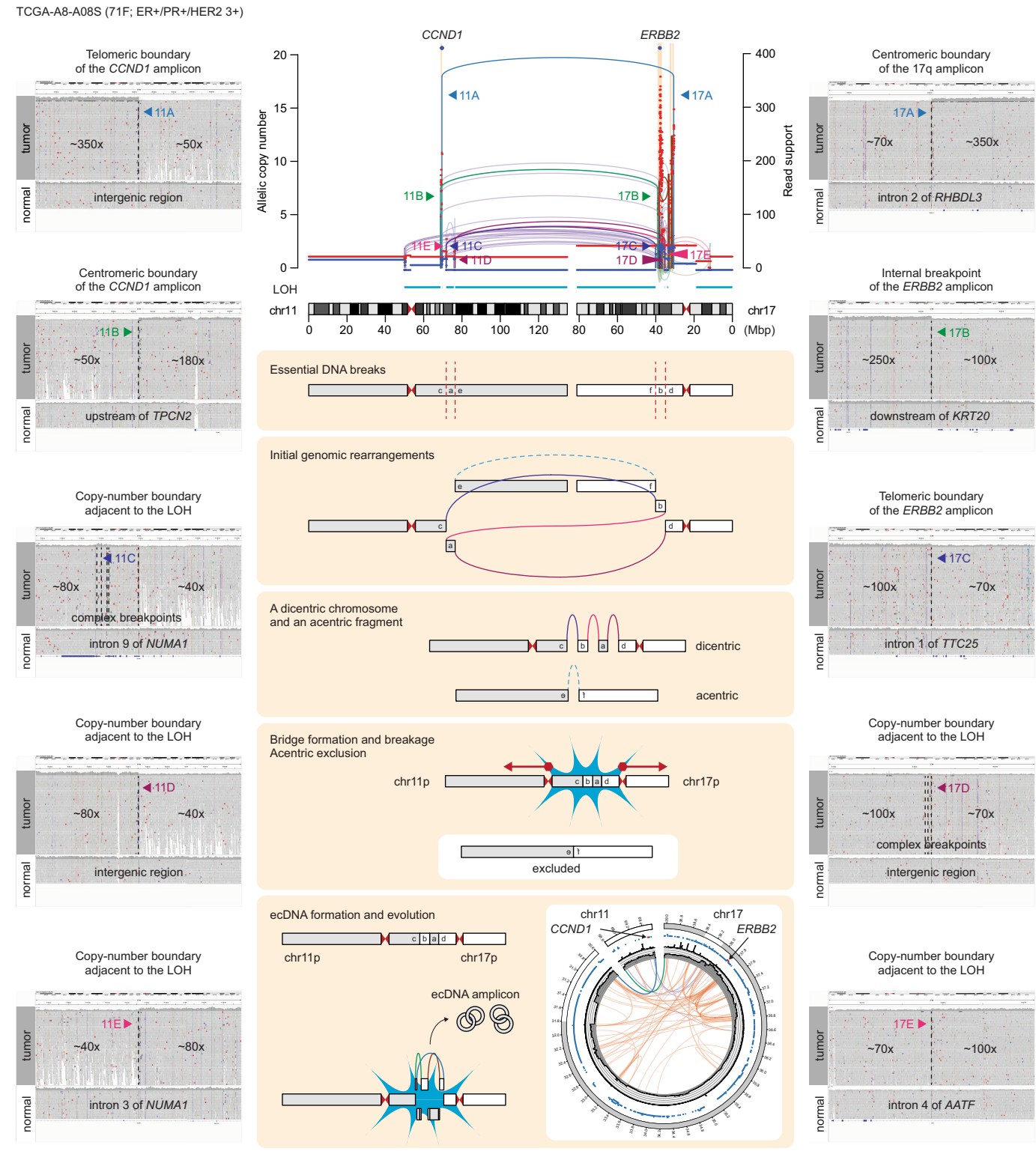

**Extended Data Fig. 3 | Reconstruction of translocation-bridge amplification.** Copy number and SV information from TCGA-A8-A08S case, harboring 11q and 17q focal amplifications connected by translocations at their boundaries (TB amplification). Five informative SVs are highlighted on the SV plot and their detailed structures are visualized by the Integrative Genomics Viewer.

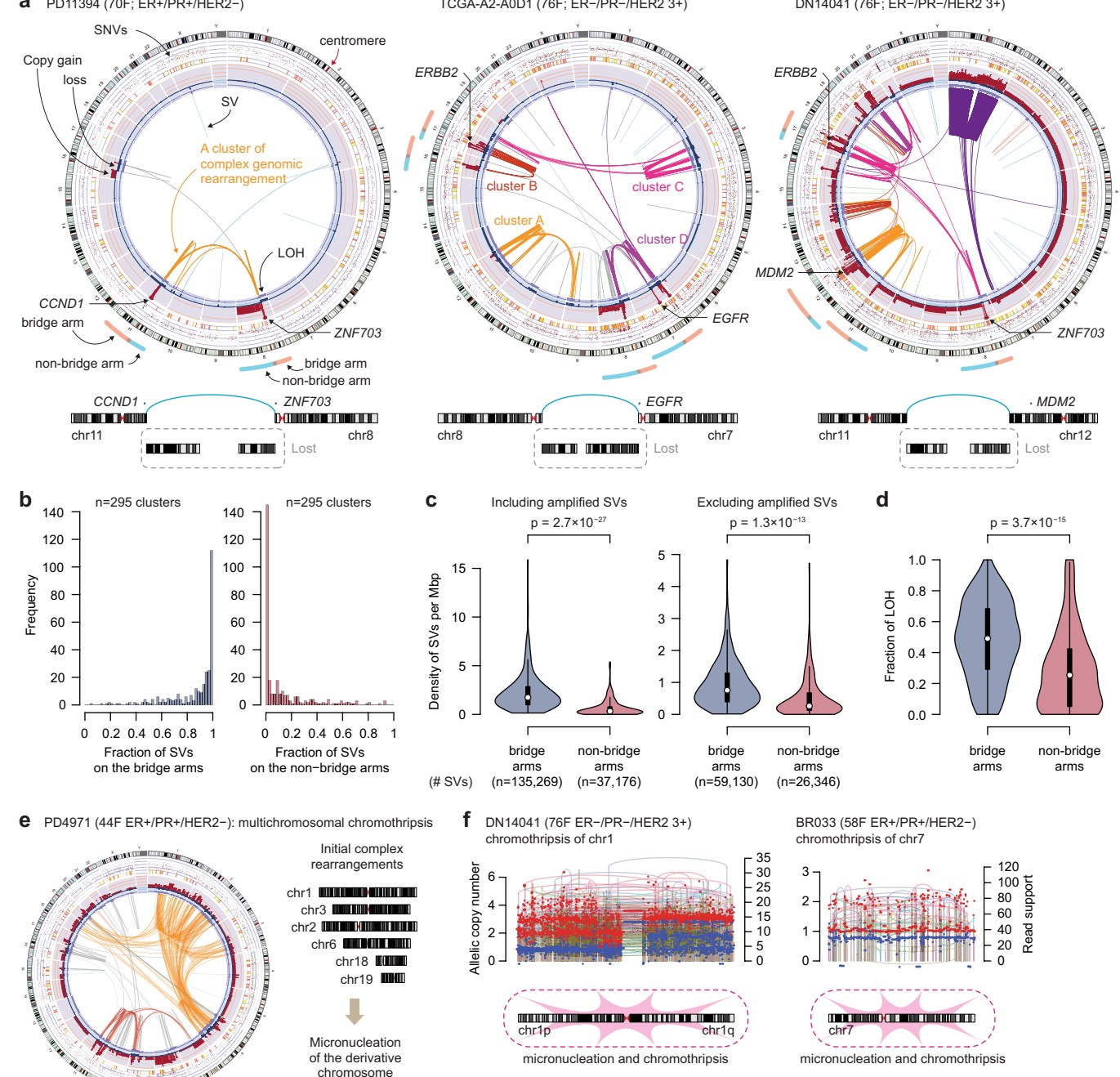

**Extended Data Fig. 4 | Genomic evidence supporting the translocation-bridge amplification model. a.** Evidence of chromosome bridge formation and breakage in three cases with TB amplifications. Complex genomic rearrangements were clustered based on the proximity of SV breakpoints (Methods). Bridge arms typically show massive amplification with heavy rearrangements and adjacent segments with LOH. In contrast, non-bridge arms are relatively spared from rearrangements and often amplified moderately. **b.** The fraction of SVs on the bridge arms (left panel) and non-bridge arms (right panel) among the complex genomic rearrangement clusters showing TB amplifications. Most clusters have their SVs concentrated on the bridge arms, whereas their non-bridge arms are often spared from the SVs. **c.** The density of SVs on the bridge and the non-bridge arms, including all SVs (left panel) and among the non-amplified SVs (right panel). The bridge arms showed a higher density of SVs compared to the non-bridge arms (median 1.74 vs. 0.33 SVs/Mbp; p = 2.7 × 10⁻²⁷; by two-sided, two-sample *t* test), and this trend

was still significant after excluding all the SVs in the amplified regions (median 0.75 vs. 0.26 SVs/Mbp; p = 1.3 × 10⁻¹³). Violin plots in **c** and **d** show the distribution of SV density, the box plot in the center of the violin indicates median (white dot), first and third quartiles (edges), and 1.5× of interquartile range (vertical line). **d.** The fraction of genomic segments affected by LOH on the bridge arms and non-bridge arms. The bridge arms showed a more extensive LOH compared to the non-bridge arms (median 49% vs 25% of the arm; p = 3.7 × 10⁻¹⁵; by two-sided, two-sample *t* test). **e.** An exemplary case of multi-chromosomal chromothripsis (yellow cluster). The SVs are distributed more-or-less evenly in the involved chromosomes without sparing one arm. **f.** Chromosomes showing typical features of chromothripsis. The SVs are distributed evenly on two chromosome arms, and the rearrangements are more-or-less random, in contrast to the asymmetric footprint of TB amplification.

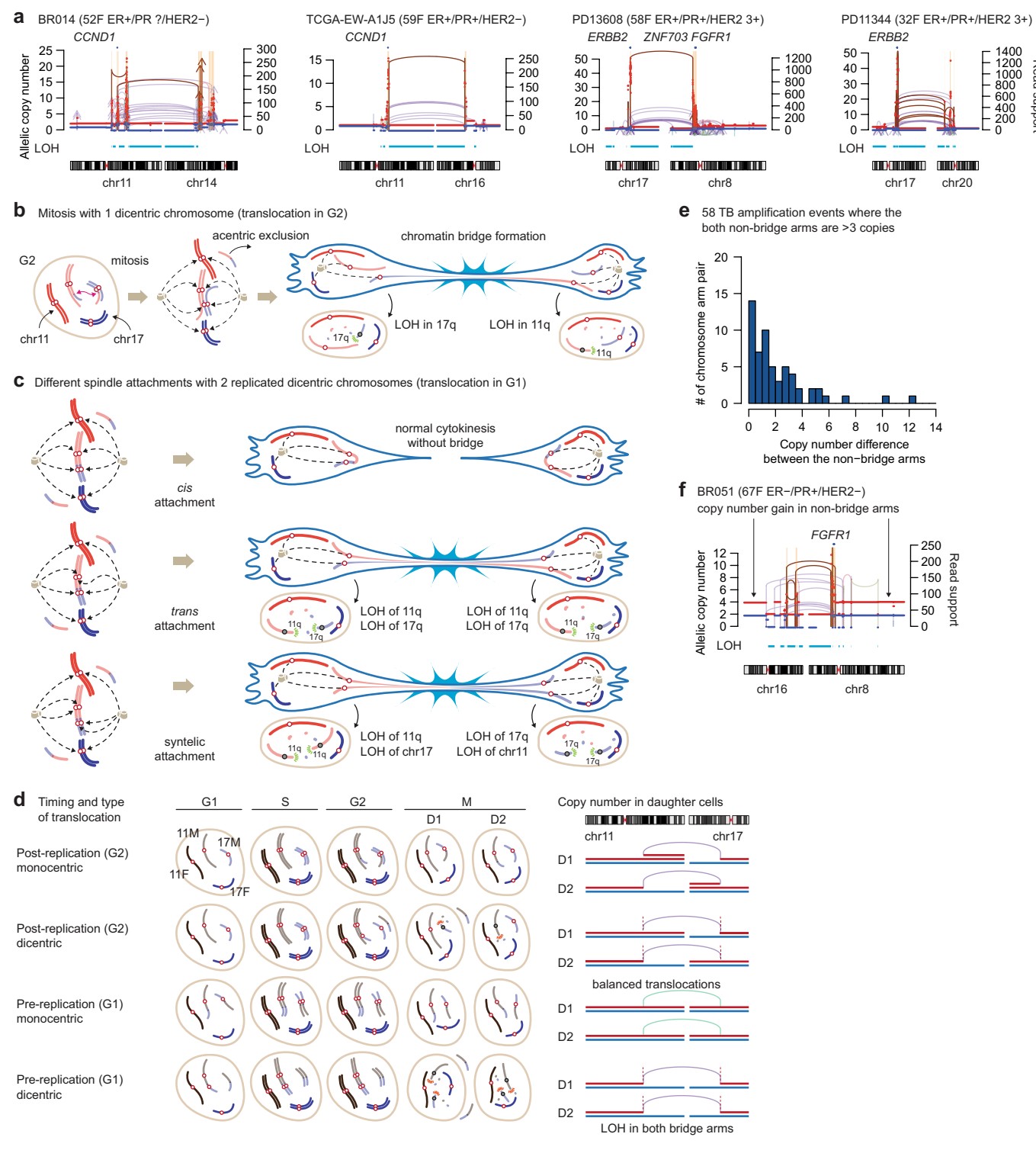

**Extended Data Fig. 5 |** See next page for caption.

**Extended Data Fig. 5 | Evidence of inter-chromosomal translocation in G1 before the DNA replication. a**. 'Dual-LOH' pattern of the bridge arms in the tumors with TB amplification. Both bridge arms in each case show substantial loss of heterozygosity (LOH). Notably, the bridge arm segments proximal to the TB amplification (the inter-centromeric segment on the bridge) often show complex copy-number pattern with segmental loss (first panel; in BR014) or gain (second panel; in TCGA-EW-A1J5) by one copy. This is likely due to the unequal breakpoints between the sister dicentrics. Depending on the location of the break, some segments can be duplicated (exhibiting one-copy gain) or lost (one-copy loss) in a daughter cell after the bridge resolution. **b**. Copy number outcome after the translocation forming a single dicentric chromosome, chromosome bridge, and its resolution. The two daughter cells will inherit one broken arm after the bridge breakage, leading to their 'single-LOH' copy-number pattern. **c**. Three different scenarios of mitotic spindle attachments in the setting of replicated dicentric sisters (initial translocation in G1). If the microtubules are attached in *cis*, normal mitosis will be secured (upper panel). If in *trans*, chromosome bridge will be formed by two 'flipped' dicentrics in anti-parallel orientation and the resultant copy number profile matches with the dual-LOH pattern frequently observed in breast cancer cases with TB amplifications (middle panel). If the microtubules from one pole are attached to the same centromeres (syntelic attachment), each daughter cell will have LOH affecting one arm of a chromosome and whole length of another chromosome (lower panel). **d**. Four different copy number outcomes depending on the timing and orientation of the translocation. We expect post-replication inter-chromosomal translocation to be less frequent due to the active homologous recombination in the S/G2 cells. **e**. Copy-number difference between the two non-bridge arms in the 58 TB amplification events where the two non-bridge arms are globally amplified more than three copies. 36 (62%) out of 58 events showed a copy number difference of less than 2. **f**. A case indicating possible repair by mutual ligation between the non-bridge arms (8q and 16p) after TB amplification. The two non-bridge arms (8q and 16p) are connected to each other by multiple translocations at the copy-number boundaries. A whole-genome duplication took place after the translocations between non-bridge arms and led to the coordinated copy-number gain.

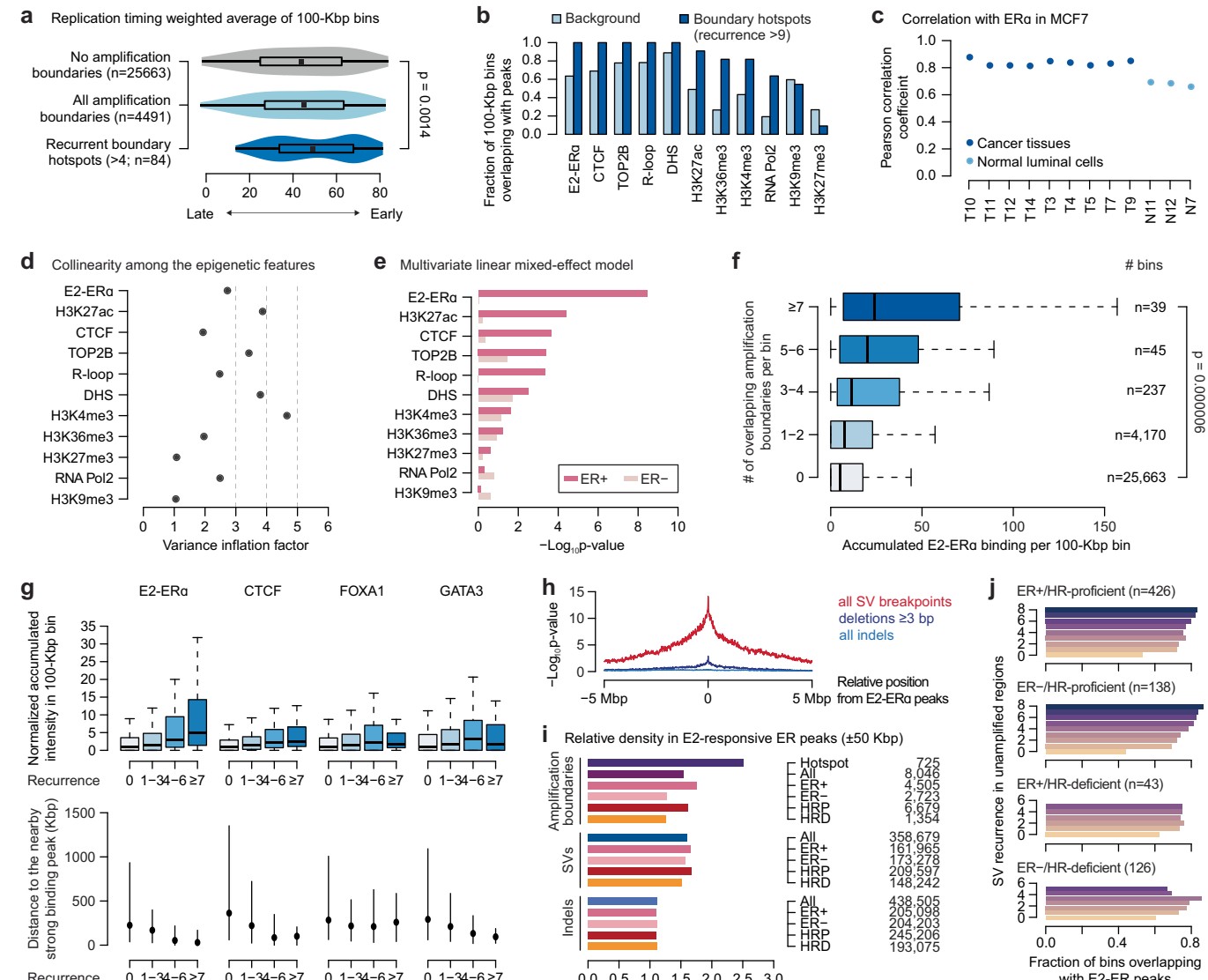

**Extended Data Fig. 6 | Associations between the epigenomic features and the amplification boundaries. a**. Replication timing and amplification boundaries. Violin plots show the replication timing weighted average values in 100-Kbp bins. The box plot in the center of the violin indicates median (black dot), first and third quartiles (edges), and 1.5× of interquartile range (horizontal line). Comparisons were made by one-sided Wilcoxon's rank sum test. **b**. Fraction of 100-Kbp bins overlapping with various epigenomic features in the background and in the recurrent hotspots of amplification boundaries (recurrence >9). The background represents the bins that do not contain amplification boundaries. All epigenomic features were from MCF7 cells, except for TOP2B (from MCF10A cells). DHS, DNase I hypersensitivity sites. **c**. Pearson's correlation coefficients between the ERα binding in MCF7 cells and in tissues, including multiple breast cancer samples and normal luminal breast epithelial cells from a previous study (Supplementary Table 3). The analysis is based on 1 Mbp-sized genome-wide bins. **d**. Assessment of multicollinearity between the epigenomic features by variant inflation factor (VIF; Methods). Some variables showed moderate degree of multicollinearity (VIF 3–5) although others including E2-ERα showed low degree of multicollinearity (VIF < 3). Given these reassuring features, we performed multivariate linear mixed-effect model (panel **e**), as an adjunct analysis. **e**. Predictors of amplification boundaries in the multivariate linear mixed-effect model, by the ER status. 100 Kbp-sized genome-wide bins were used in this analysis. Raw p-values from two-sided test are shown. **f**. A greater cumulative E2-ERα binding is observed in the 100-Kbp bins with more frequent overlaps with the focal amplification boundaries. E2-ERα ChIP-seq data from MCF7 cell line was used in this analysis. Box plots indicate median (thick line), first and third quartiles (edges), and 1.5× of interquartile range (whiskers). Statistical significance was

determined by the one-sided rank sum test. **g**. Binding intensity (fold enrichment) of E2-treated ERα, CTCF, FOXA1, and GATA3 in MCF7 cells based on the 1 Mbp-sized genomic bins with different levels of overlap with the focal amplification boundaries (upper panel). The numbers of genomic bins used in each category are as follows: n = 25663 (recurrence = 0), 4334 (1–3), 118 (4–6), and 39 (≥7). Box plots indicate median (thick line), first and third quartiles (edges), and 1.5x of interquartile range (whiskers). A statistically significant increase in the binding intensity of E2-treated ERα was observed with increasing recurrence of amplification boundaries (p = 2.8 × 10⁻⁶, one-sided Wilcoxon's rank sum test). Genomic distances from the bins containing the amplification boundaries to the strong binding peaks (top 10%) of E2-ERα, CTCF, FOXA1, and GATA3 in MCF7 cells (lower panel). Black dots indicate median, and the vertical lines indicate the range between first and third quartiles. **h**. Enrichment of different classes of variants with respect to the expected values under the assumption of uniform distribution in 100-Kbp genomic bins within 5-Mbp window for each E2-induced ERα peak locus. Statistical significance was assessed by one-sided Fisher's exact test. **i**. Relative density of amplification boundaries, SVs, and indels by their subgroups around the E2-ERα peaks (±50-Kbp window from the center of the peak). Here, the amplification boundary hotspots were defined as 100-Kbp bins supported by >4 tumors. Number of variants used in the analysis was marked on the right. HRP, HR-proficient tumors; HRD, HR-deficient tumors. **j**. Subgroup analyses of the relationship between the SV hotspots of the unamplified regions and the frequency of E2-ERα binding peaks in the regions (related to Fig. 3c). A positive correlation between the E2-ERα peaks and the unamplified SV hotspots is observed among the HR-proficient tumors. In contrast, the trend is not found in HR-deficient tumors.

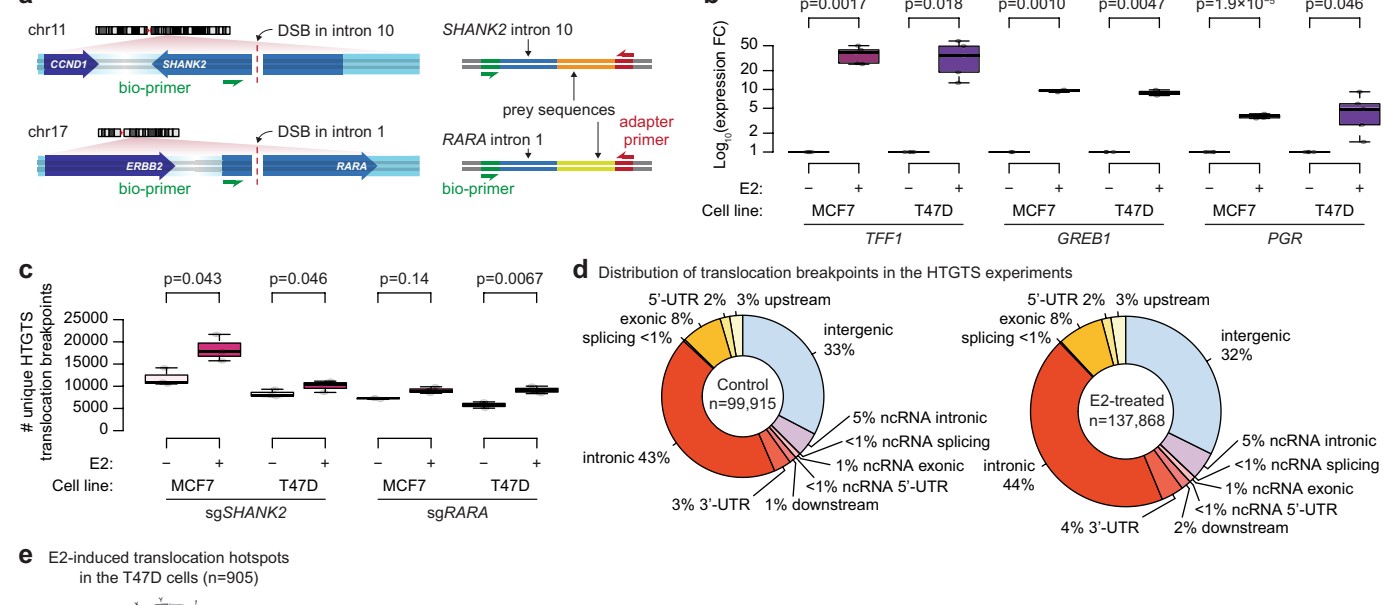

**e** E2-induced translocation hotspots
in the T47D cells (n=905)

*RARA*

*SHANK2*

**Extended Data Fig. 7 | Estradiol induces transcription of its target genes and increases HTGTS translocations. a**. Design of the HTGTS experiment. Using CRISPR/Cas9 system, we induced the DNA double strand breaks (DSBs) in the intronic regions near the prominent E2-ERα binding peaks (intron 10 of *SHANK2* and intron 1 of *RARA*). These sites are also located at the downstream neighborhood of the oncogenes of interest (*ERBB2* and *CCND1*). We designed the library to amplify the translocated sequences to the centromeric end of the CRISPR breaks, which is in the orientation potentially forming a dicentric chromosome. **b**. An increased mRNA expression of canonical target genes of ERα by the E2 treatment. All three genes showed robust upregulation of their expression in both MCF7 and T47D cells. n = 5 biologically independent experiments were performed for *TFF1* and *PGR*, and n = 3 for *GREB1* in two different cell lines. Box plots in **b** and **c** indicate median (thick line), first and third quartiles (edges), and 1.5x of interquartile range (whiskers). In both, statistical comparisons were made by two-sided, two-sample *t* test. **c**. An increased number of unique HTGTS translocation breakpoints by the E2 treatment in all four experimental pairs. n = 3 biologically independent experiments were performed in each group. **d**. Genomic annotation of the HTGTS translocation breakpoints in the control and E2-treated experiments. **e**. A circos plot visualizing the hotspots E2-induced translocations (> 4-fold change by the E2 treatment) between the induced breaks and the prey regions in the T47D cells.

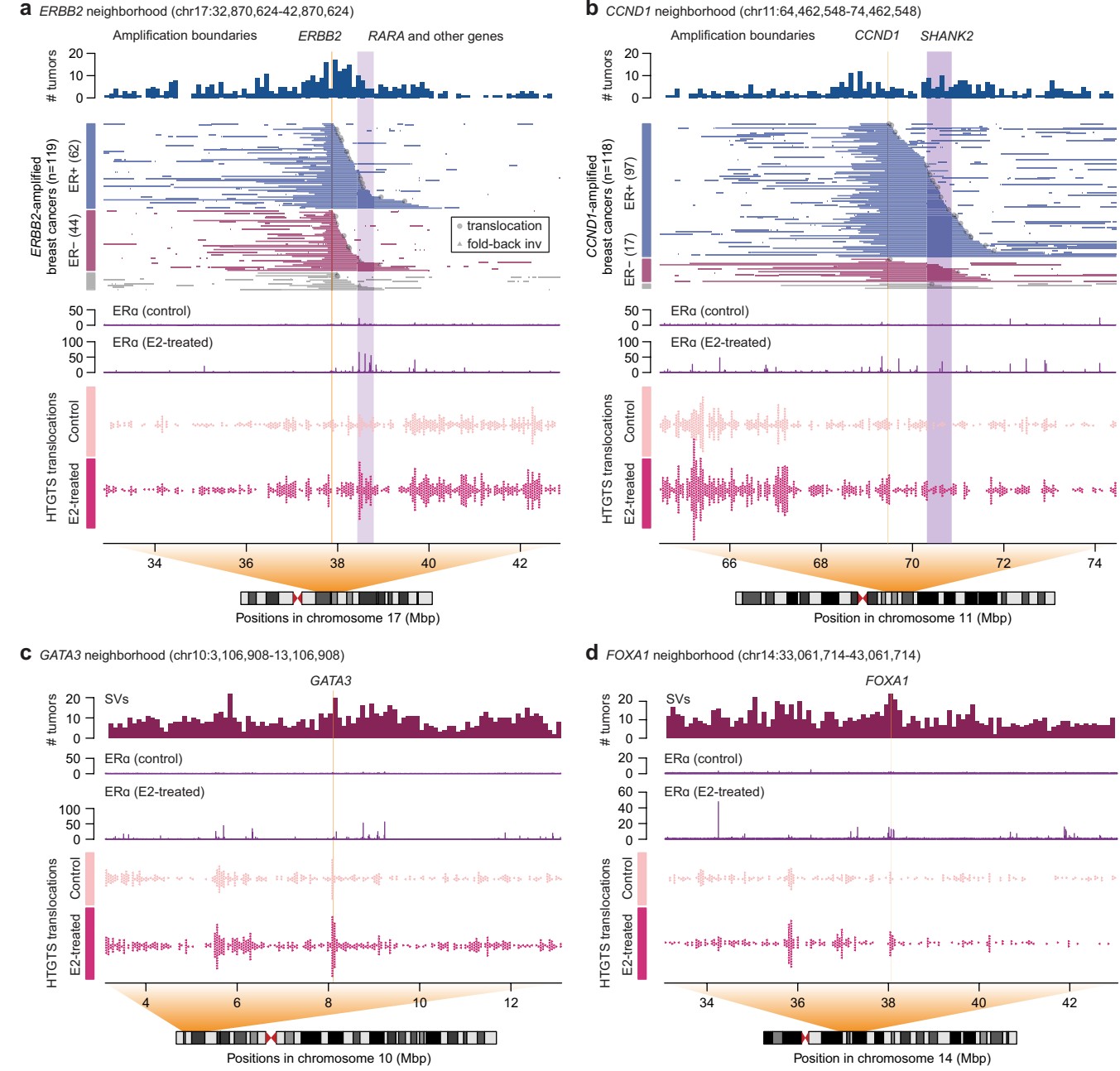

**Extended Data Fig. 8 | Association of E2-induced HTGTS translocations and the driver events in breast cancer genomes. a.** Association between the E2-induced HTGTS translocations and *ERBB2* amplicons. The 10-Mbp genomic region around *ERBB2* (orange shadow) was visualized. Frequency of amplification boundary per bin (uppermost panel), amplified segments in each tumor, ordered by the location of telomeric boundary and the ER status (mid-upper panel), ERα ChIP-seq profile in MCF7 cells (mid-lower panel), and the HTGTS translocation breakpoints from the experiments using sg*SHANK2* (lowermost panel). Extensive E2-ERα binding was observed in several hotspots in the neighborhood of *CDC6*, *RARA*, *IGFBP4*, and *CCR7* (purple shadow), and this region overlaps with the telomeric border of the *ERBB2* amplicon. E2 treatment increased the frequency of translocations >3 fold (from 22 to 69; odds ratio: 1.87 compared to the background; p = 0.009 by Fisher's exact test, two-sided) in the purple shadow region (chr17:38,450,000-38,750,000). E2-induced HTGTS translocation hotspots, prominent E2-ERα binding peaks,

and the telomeric boundary of the *ERBB2* amplicons well overlapped each other, suggesting E2-induced, ERα-mediated fragility as the mechanism of initial translocation that led to the TB amplification. Local DNA fragmentation and segmental loss after the chromosome bridge breakage could explain the tumors with more proximal boundaries. **b.** A similar example of *CCND1* (orange shadow) amplification and its neighborhood. The HTGTS translocation breakpoints were from the experiments using sg*RARA*. E2 treatment also increased the frequency of translocations from 14 to 40 in the region around *SHANK2* (chr11:70,300,000-71,000,000; purple shadow), but this was not significantly larger than the background increase (odds ratio: 1.70; p = 0.09 by two-sided Fisher's exact test). Further discussions in Supplementary Fig. 7. **c.** An example of unamplified SV hotspot in *GATA3*, which overlaps with the E2-induced HTGTS translocation hotspot. **d.** Another example of unamplified SV hotspot in *FOXA1*.

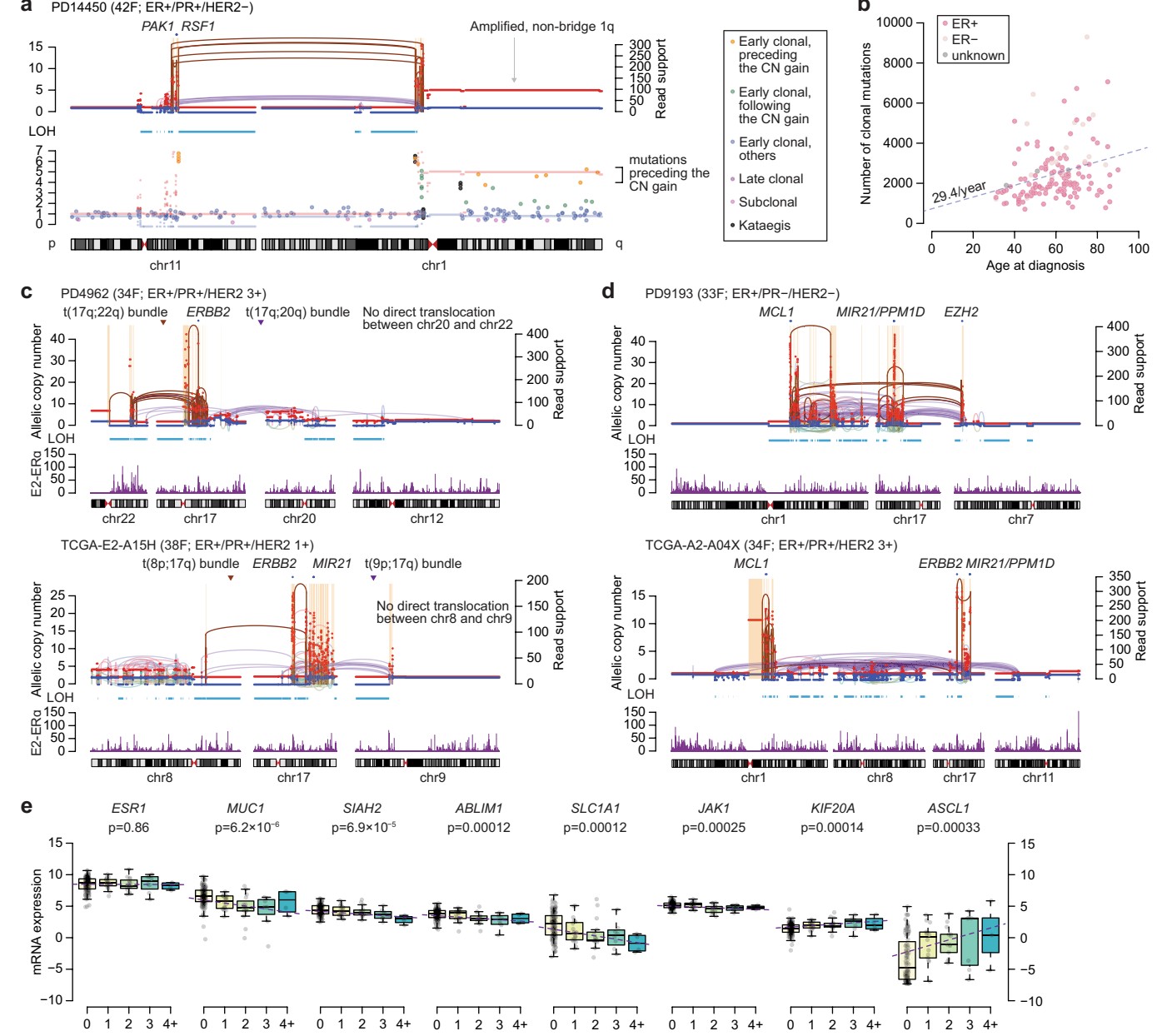

**Extended Data Fig. 9** | See next page for caption.

**Extended Data Fig. 9 | Timing, complexity, and transcriptional impact of translocation-bridge amplification. a**. A TB amplification case from PD14450 showing an amplified non-bridge arm, 1q. The paucity of the SNVs amplified up to the allelic copy-number of 1q indicates that the TB amplification and the subsequent 1q amplification are early events. **b**. Relationship between the age of diagnosis and the burden of clonal single-nucleotide variations (SNVs) in non-hypermutated, diploid, and HR-proficient breast cancers (Methods). The dashed line is based on linear regression (r = 0.29 by Pearson's correlation; two-sided p = 0.0004), indicating our cohort's approximate baseline clonal mutation rate of 29.4/years. A similar analysis using all SNVs in the same group of patients showed a mutation rate of 33.1/years (r = 0.26 by Pearson's correlation; two-sided p = 0.002; corresponding to a median of 45.3 years for the latest possible timing of bridge breakage). **c**. Cases suggesting multiple rounds of dicentric chromosome bridge formation and breakage. Translocation bundles are spatially separated on chromosomes. In PD4962 tumor (upper panel), two bundles of translocations were observed — t(17q;20q) and t(17q;22q). However, there is no direct translocation between chromosome 22q and 20q, indicating the two translocation bundles were formed at different time points. In the same way, TCGA-E2-A15H tumor (lower panel) shows t(8p;17q) and t(9p;17q), but no direct connection was observed between 8p and 9p. **d**. Cases indicating one round of multi-chromosomal TB amplifications, likely involving multiple dicentric chromosomes or a more complex structure, broken and ligated at one event. In contrast to (**c**), these cases show dense bundles of translocations between all pairs of chromosomes involved in the event. Individual chromosome arms have translocations connected to all other involved chromosome arms, indicating that these arms were in the bridge simultaneously, and the broken DNA fragments were ligated in a mixed manner. In PD9193 tumor (upper panel), chromosomes 1q, 17q, and 7q were rearranged first and underwent massive rearrangements. The LOH selectively affecting one of the two arms of each chromosome is consistent with typical TB amplification. After the catastrophic breaks, the DNA fragments were ligated to form an amplicon containing multiple oncogenes. In TCGA-A2-A04X tumor (lower panel), the initial event might be a chromoplexy involving chromosomes 1, 7, 8, 11, 17. Four telomeric LOH segments were observed in the involved chromosomes, so two dicentric chromosomes would likely have existed and were fragmented simultaneously. **e**. Associations between the extent of TB amplifications and the expression of estrogen-responsive genes. Numbers of the ER+ tumors used in the right panel are as follows: n = 141 (0 chromosome arm pairs indicating a TB amplification event), 20 (1), 15 (2), 8 (3), and 4 (4). Box plots indicate median (thick line), first and third quartiles (edges), and 1.5× of interquartile range (whiskers). Statistical tests by linear regression, and the gray dashed line is the regression line. Nine out of 273 genes showed statistically significant trend (FDR < 0.1 after correcting multiple testing). Among these genes, *CDC6* and *TOP2A* showed significant upregulation among the tumors with extensive TB amplification likely due to their presence in the amplicon. We excluded these two genes from this plot.

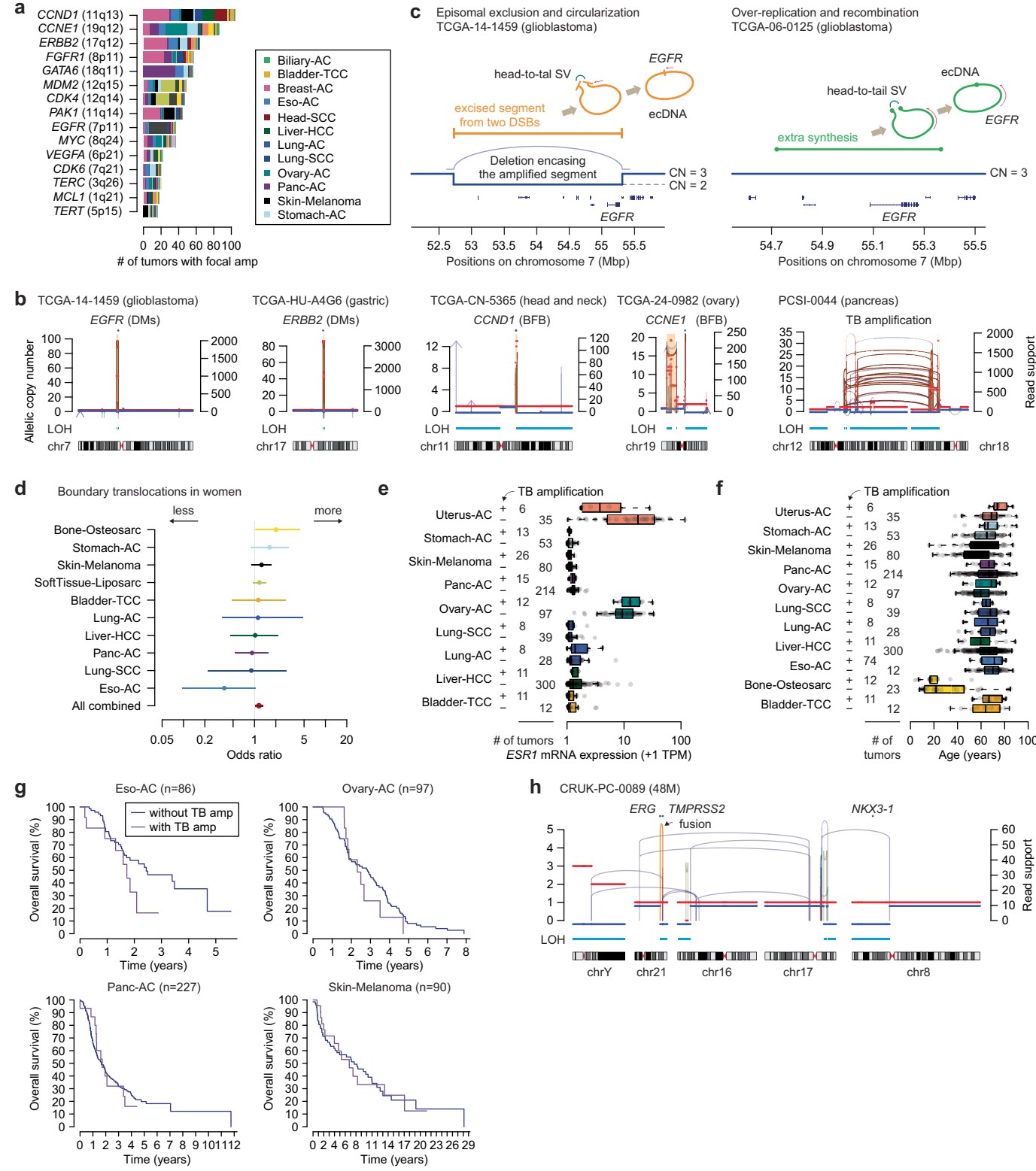

**Extended Data Fig. 10 | See next page for caption.**

**Extended Data Fig. 10 | Diverse mechanisms of focal amplification in pan-cancer. a**. Common target genes of focal amplification in the PCAWG cohort. Colors in the stacked bar plot indicates the tumor type. **b**. Representative cases of focal amplifications from diverse mechanisms in different tumor types. DMs, double minutes; BFB, breakage-fusion-bridge **c**. Two different modes of simple ecDNA formation in glioblastoma samples. Two glioblastoma cases showed an ecDNA formation from cut-out piece of DNA fragment containing *EGFR* (left panel). In these cases, a large deletion totally encompassing the cut-out fragments were observed. These cases demonstrate episomal exclusion mechanism in generating ecDNA. In contrast, many cases showed an ecDNA formation from an extra copy of short DNA patch harboring the *EGFR* (right panel), likely explained by over-replication and recombination. **d**. A meta-analysis excluding cancers of female or male organs showed a marginal trend of more translocations at the amplicon boundaries in female patients. **e**. TB amplification was associated with low *ESR1* mRNA expression in endometrial and liver cancers, although not reaching to our level of statistical significance after correcting multiple testing (FDR <0.1 with two-sided, two-sample *t* test). In these two tumor types, the tumors with TB amplification were rare (6/41 for endometrial and 11/300 for liver) and showed a lower *ESR1* expression

compared to those without TB amplification (7.1 vs. 22.8; p = 0.02 for endometrial, 0.37 vs. 0.82; p = 0.02 for liver, by two-sided, two-sample *t* test; FDR = 0.20 for both). Box plots in **e** and **f** indicate median (solid vertical line in the middle), first and third quartiles (edges), and 1.5x of interquartile range (whiskers). **f**. TB amplification status was associated with old ages in uterine cancer (76 vs. 67 years; p = 0.03 by two-sided, two-sample *t* test). However, the number of tumors with TB amplification was small (n = 6), and the statistical comparison was insignificant after correcting multiple testing (FDR = 0.69). No age difference was noted in other cancer types depending on the TB amplification status. **g**. We performed survival analysis by tumor type when there are 50 or more patients with available survival information. Survival information of the four tumor types with 10 or more cases with TB amplification are plotted. Statistical test was made by two-sided log-rank test, and none of the four tumor types showed significantly different overall survival depending on the TB amplification status. **h**. A representative case of *TMPRSS2-ERG* fusion from complex genomic rearrangements suggesting bridge breakage. In this case, 16q and 21q would have been translocated to each other, forming a dicentric chromosome. Bridge formation and resolution left a typical footprint of dual LOH on both bridge arms without causing focal amplification.

# Reporting Summary

## Statistics

For all statistical analyses, confirm that the following items are present in the figure legend, table legend, main text, or Methods section.

| n/a | Confirmed | |
|---|---|---|
| ☐ | ☒ | The exact sample size (*n*) for each experimental group/condition, given as a discrete number and unit of measurement |
| ☐ | ☒ | A statement on whether measurements were taken from distinct samples or whether the same sample was measured repeatedly |
| ☐ | ☒ | The statistical test(s) used AND whether they are one- or two-sided<br>*Only common tests should be described solely by name; describe more complex techniques in the Methods section.* |
| ☐ | ☒ | A description of all covariates tested |
| ☐ | ☒ | A description of any assumptions or corrections, such as tests of normality and adjustment for multiple comparisons |
| ☐ | ☒ | A full description of the statistical parameters including central tendency (e.g. means) or other basic estimates (e.g. regression coefficient) AND variation (e.g. standard deviation) or associated estimates of uncertainty (e.g. confidence intervals) |
| ☐ | ☒ | For null hypothesis testing, the test statistic (e.g. $F$, $t$, $r$) with confidence intervals, effect sizes, degrees of freedom and $P$ value noted<br>*Give P values as exact values whenever suitable.* |
| ☒ | ☐ | For Bayesian analysis, information on the choice of priors and Markov chain Monte Carlo settings |
| ☒ | ☐ | For hierarchical and complex designs, identification of the appropriate level for tests and full reporting of outcomes |
| ☐ | ☒ | Estimates of effect sizes (e.g. Cohen's *d*, Pearson's *r*), indicating how they were calculated |

*Our web collection on statistics for biologists contains articles on many of the points above.*

## Software and code

Policy information about availability of computer code

| | |
|---|---|
| Data collection | Data was downloaded and extracted with EGA Download Client V3 (available at https://github.com/EGA-archive/ega-download-client). |
| Data analysis | List of softwares:<br>- R (version 4.1.1): https://www.r-project.org<br>- BWA MEM (version 0.7.15): https://github.com/lh3/bwa<br>- Samtools (version 1.3.1): https://github.com/samtools/samtools<br>- Bazam (version 1.0.1): https://github.com/ssadedin/bazam<br>- Picard (version 2.8.0): http://broadinstitute.github.io/picard<br>- PURPLE (version 2.54): https://github.com/hartwigmedical/hmftools/blob/master/purple<br>- COBALT (version 1.11): https://github.com/hartwigmedical/hmftools/blob/master/cobalt<br>- AMBER (version 3.5): https://github.com/hartwigmedical/hmftools/blob/master/amber<br>- GRIDSS2 (version 2.12.0): https://github.com/PapenfussLab/gridss<br>- RepeatMasker (version 4.1.2-p1): https://github.com/rmhubley/RepeatMasker<br>- Kraken2 (version 2.1.2): https://github.com/DerrickWood/kraken2<br>- GRIPSS (version 1.9): https://github.com/hartwigmedical/hmftools/blob/master/gripss<br>- LINX (version 1.15): https://github.com/hartwigmedical/hmftools/blob/master/linx<br>- SAGE (version 2.8): https://github.com/hartwigmedical/hmftools/blob/master/sage<br>- CHORD (version 2.00): https://github.com/UMCUGenetics/CHORD<br>- xTea (version 0.1.6): https://github.com/parklab/xTea<br>- SigProfilerMatrixGenerator (version 0.1.0): https://github.com/AlexandrovLab/SigProfilerMatrixGenerator<br>- MuSiCal (version 1.0.0-beta): https://github.com/parklab/MuSiCal |

- MutationTimeR (version 1.00.2): https://github.com/gerstung-lab/MutationTimeR
- MACS (version 2): https://hbctraining.github.io/Intro-to-ChIPseq/lessons/05_peak_calling_macs.html
- Bowtie (version 1.2.2): https://github.com/BenLangmead/bowtie
- GSEA (version 4.2.3): https://www.gsea-msigdb.org/gsea
- GEAT (version 0.1): https://github.com/geatools/GEAT
Custom code is available (also stated in the manuscript): https://github.com/parklab/focal-amplification
No commercial software is used.

For manuscripts utilizing custom algorithms or software that are central to the research but not yet described in published literature, software must be made available to editors and reviewers. We strongly encourage code deposition in a community repository (e.g. GitHub). See the Nature Portfolio guidelines for submitting code & software for further information.

## Data

Policy information about availability of data

All manuscripts must include a data availability statement. This statement should provide the following information, where applicable:
- Accession codes, unique identifiers, or web links for publicly available datasets
- A description of any restrictions on data availability
- For clinical datasets or third party data, please ensure that the statement adheres to our policy

WGS datasets generated through ICGC or PCAWG consortium are available at the ICGC Data Portal (http://dcc.icgc.org) with download instructions and links available in the downloading PCAWG data section (https://docs.icgc.org/pcawg/data/). For 72 tumors in French study by Ferrari et al., we downloaded the BAM files from European Genome-phenome Archive (EGA; https://www.ebi.ac.uk/ega/) under the accession number of EGAS00001001431. BAM or CRAM files from the Sanger 560 breast cancer project were downloaded from EGA under the accession number of EGAD00001001334, EGAD00001001335, EGAD00001001336, EGAD00001001338, and EGAD00001001322, those from the British Columbia study were under EGAS00001001159 (more detailed sample-by-sample accession numbers are available in Table S4 in the published paper), and those from the Yale inflammatory breast cancer project were under EGAS00001004117. HTGTS dataset is available in Gene Expression Omnibus (GEO) under the accession number of GSE227369. MSigDB gene set collections are available at GSEA website (http://www.gsea-msigdb.org/gsea/downloads.jsp). Epigenomic datasets are also publicly available at Gene Expression Omnibus (GEO; http://www.ncbi.nlm.nih.gov/geo), 4D Nucleome Data Portal (http://data.4dnucleome.org), and other repositories under accession numbers provided in Supplementary Table 3. Somatic variant calls, including SNVs, indels, SVs, and allelic copy number information for 780 breast cancer cases are available at the Park lab website (http://compbio.med.harvard.edu/TBAmplification/).

## Human research participants

Policy information about studies involving human research participants and Sex and Gender in Research.

| | |
|---|---|
| Reporting on sex and gender | This is a meta-analysis study that we do not directly collect information from participants. Instead, we collected sex information from five published studies. Because breast cancer predominantly affects female, and the biological mechanism that we describe in the paper is associated with female endocrine physiology, we focused on biological sex in our study. |
| Population characteristics | This study primarily describes 780 patients with breast cancer. This is a meta-analysis of five published studies based on whole-genome sequencing, and the details of each study is available in "Patient cohort" section of the Methods. Clinicopathologic characteristics of the patients are described in Extended Data Fig. 1. |
| Recruitment | This study is a meta-analysis without direct recruitment of participants. Whole-genome sequencing datasets were obtained from public repositories (accession numbers available in the data availability section of the Methods). |
| Ethics oversight | The institutional review board of the Harvard Faculty of Medicine approved this study (IRB18-0151). Individual studies complied required ethical guidelines per published manuscripts. |

Note that full information on the approval of the study protocol must also be provided in the manuscript.

## Field-specific reporting

Please select the one below that is the best fit for your research. If you are not sure, read the appropriate sections before making your selection.

☒ Life sciences ☐ Behavioural & social sciences ☐ Ecological, evolutionary & environmental sciences

For a reference copy of the document with all sections, see nature.com/documents/nr-reporting-summary-flat.pdf

## Life sciences study design

All studies must disclose on these points even when the disclosure is negative.

| | |
|---|---|
| Sample size | For genome analysis, no sample size calculations were performed. Sample size was determined by the number of sequencing datasets available in each individual study. In total, breast cancer genomes from 780 patients were analyzed in this study. For in vitro experiments, n=3 biological replicate experiments were performed for reliability and feasibility for statistical analysis. Each biological replicate is defined as an independent cell cultures. |

| Data exclusions | We originally downloaded 787 cases. Five cases were excluded because they were sequenced from formalin-fixed paraffin-embedded tissues, and two cases were excluded due to the failure in quality assessment step in our bioinformatic analysis. |
| Replication | We performed high-throughput genome-wide translocation sequencing with three biological replicates per experiment. Result was concordant among the biological replicates. Details are available in Extended Data Fig. 7. |
| Randomization | No randomization was performed, given the descriptive nature of the study. |
| Blinding | No blinding was performed, given the descriptive nature of the study. |

# Reporting for specific materials, systems and methods

We require information from authors about some types of materials, experimental systems and methods used in many studies. Here, indicate whether each material, system or method listed is relevant to your study. If you are not sure if a list item applies to your research, read the appropriate section before selecting a response.

### Materials & experimental systems

| n/a | Involved in the study |
|-----|------------------------|
| ☒ ☐ | Antibodies |
| ☐ ☒ | Eukaryotic cell lines |
| ☒ ☐ | Palaeontology and archaeology |
| ☒ ☐ | Animals and other organisms |
| ☒ ☐ | Clinical data |
| ☒ ☐ | Dual use research of concern |

### Methods

| n/a | Involved in the study |
|-----|------------------------|
| ☒ ☐ | ChIP-seq |
| ☒ ☐ | Flow cytometry |
| ☒ ☐ | MRI-based neuroimaging |

## Eukaryotic cell lines

Policy information about cell lines and Sex and Gender in Research

| Cell line source(s) | MCF7 (ATCC), T47D (ATCC), and 293FT (Invitrogen/ThermoFisher) |
| Authentication | The suppliers of these cell lines provide information on the generation, characteristics, and authentication of the cell line in its website (MCF7: https://www.atcc.org/products/htb-22; T47D: https://www.atcc.org/products/htb-133; 293FT: https://www.thermofisher.com/order/catalog/product/R70007). Cell lines authentication was performed using short tandem repeat (STR) by the supplier. |
| Mycoplasma contamination | The cells were tested negative for Mycoplasma contamination |
| Commonly misidentified lines (See ICLAC register) | None listed in the ICLAC register. |

