## [Peer Review File · Nature]

Manuscript Title: ER α -associated translocations underlie oncogene amplifications in breast cancer

Reviewer Comments & Author Rebuttals

Reviewer Reports on the Initial Version:

Referees' comments:

Referee #1 (Remarks to the Author):

This manuscript describes a computational study of a proposed genomic mechanism by which cancer genomes acquire recurrent genomic amplifications of oncogenes through combined translocation and amplification. They term this mechanism translocation-bridge amplification and associate its occurrence with estrogen receptor binding of DNA at breakpoint loci. They study published datasets of cancer genomes from the ICGC project to describe such genomic events, first in breast cancer and then in the ICGC pan-cancer dataset of multiple cancer types. Overall this is an interesting manuscript, but in its given shape it remains largely speculative and correlative and in-depth computational and experimental work would strengthen the study. some comments and questions below:

1. Several key oncogenes are listed to involve such amplification events such as ERBB2, CCND1, and others. Can the authors demonstrate novel findings of some new oncogenes and acquire evidence of their role in cancer? Amplified genes should be expected to be upregulated in corresponding cancer samples, so mRNA analysis might support such driver genes. Functional validation of novel genes in cell lines or other models would strengthen the study.
2. The link with estrogen binding and DNA breakage is only associative and is based on DNA-binding profiles a few regulatory proteins in common cell lines. Breakpoint sites could be bound by any number of other factors, the epigenomes of primary human cancers may differ from those of cell lines and normal tissues, and DNA binding does not imply causation (although causation is implied repeatedly in the manuscript). Functional and mechanistic experiments to explore causal effects of DNA breaks and estrogen exposure would substantially strengthen the manuscript, since this part is currently only suggestive.
3. Estrogen exposure is likely to have sex specific effects in mutagenesis. While these are effects likely clearer in breast cancer that is predominantly a cancer of female patients, they are less obvious in other cancer types and remain unexplored in the pan-cancer analysis in the latter part of the manuscript. Are there significant sex differences in the occurrences of these genomic events in the various cancer types they consider? are numbers of chromosomal events associated with patient age which may also contribute to estrogen levels? How about overall or progression-free survival or any molecular features of cancers, such as the expression levels or mutations of ER? As a minor note, the use of a male cancer sample as a prominent example was somewhat surprising due to the relative infrequency of those compared to female cancers.

4. The authors have combined ICGC data of breast cancer and PCAWG data of breast cancer and other cancer types into a joint analysis. Although this helps increase power of the analysis, the inconsistent processing of the datasets may cause technical artefacts to become more prominent. Also, detection of structural variations is challenging and in especially for complex events and translocations. the ICGC PCAWG consortium has produced consensus structural variation calls and the approach used by the authors could be validated by examining those PCAWG calls summarised from multiple variant callers. Complementary validation using an independent cohort of whole cancer genomes would also help strengthen the manuscript, since the breast cancer data analysed in the first part of the study is likely partially represented in the second part of the study of PCAWG genomes.

5. Is there evidence of other somatic mutations occurring at recurrent translocation/amplification hotspots, such as SNVs or indels that would indicate additional details of the mutagenic process if they were significantly co-occurring with structural variants, or perhaps additional driver mutations if significantly mutually exclusive with structural variants across cancer patients?

Referee #2 (Remarks to the Author):

The authors report an intriguing analysis and model of the evolution of high-level amplification in cancers – predominantly breast cancer, but with pan-cancer extension. The mechanistic basis of spiky high-level amplifications involving 2+ chromosomes in cancer genomes has remained unexplained since they were first described in the days of copy number arrays. The major insight in the paper is the proposal that these events are initiated by a translocation between two chromosomes that generates a dicentric chromosome – then, with aberrant spindle attachment, a chromosome bridge is generated at mitosis, leading to chromothripsis and ecDNA formation. Subsequently, under selective pressure, the ecDNA (carrying several oncogenes) amplifies and further rearranges, generating the final structures observed in the cancer genome. The model is certainly credible and interesting – I have the following (rather verbose) comments:

1. Other potential models for the evolution of such amplifications – The critical test for the model is whether it is more credible or fits the data better than other possible explanations. The major alternative hypothesis is that these events are initiated by chromothripsis involving two or more whole chromosomes (ie, no initiating translocation). As shown in previous in vitro work from the authors, multiple lagging chromosomes in the same mitosis can wind up captured in the same micronucleus and undergo co-rearrangement. It was not clear to me what features of the data meant that a single translocation had to have initiated the crises observed here. The authors provide strong evidence that interchromosomal rearrangements are heavily amplified and demarcate regions of major amplification – as they say, this means that these rearrangements preceded the amplification. However, in nearly all of the examples shown in Extended Figure 3, there are multiple heavily amplified interchromosomal rearrangements. There may indeed be one event that is more heavily amplified than others but this does not necessarily mean that it occurred earlier – it could just be that it's closer to the target oncogene and therefore under more amplification pressure within the ecDNA. Indeed, probably the critical determinant of timing such rearrangements is the

copy number on either side of breakpoint junctions. That is, a breakpoint junction that demarcates moderate-amplification from non-amplification would likely have happened earlier than one that demarcates minor-amplification from massive-amplification even though the latter rearrangement is present in more copies per cell. It is hard to tell from the low-resolution Ext Fig 3 that I could access, but it seemed like multiple interchromosomal events demarcated amplified from non-amplified regions in most patients. This would arguably be more consistent with the simpler model of chromothripsis as the initiating event.

2. Timing from point mutations – It would be interesting to include an analysis of somatic point mutations within the amplified region, and their ploidy. This will give a sense of the mutation burden the ancestral cell would have had when the amplification event occurred. For example, if there are no / very few point mutations present at high levels of ploidy, then it suggests that the amplification is a critical early event in the evolution of the cancer.

3. Correlation of breakpoint locations with ER binding – These analyses are notoriously difficult because many features of the genome are co-correlated. For example, replication timing, gene density, active histone marks (and presumably oestrogen receptor binding?) are all positively correlated with one another across the genome. This makes it difficult to be certain which factor best explains density of somatic mutations / breakpoints, especially with univariate analyses as performed here. A multivariate analysis should be performed (although is non-trivial due to spatial auto-correlation effects).

4. Effects of selection – The ER binding correlation with SV density here is complicated even further by the fact that the nearby genes are under positive selection in an ER+ breast cancer. That is, it could be that only genes with strong ER-mediated transcriptional control could be positively selected for in an ER-expressing breast cell and therefore SV breakpoints would be closer to ER-binding sites than random genome positions. An additional analysis that would help here is to explore the correlation between passenger translocation events (ie not under selection) and ER binding – similar to Ext Fig 5g/h, but with ‘all translocations’, not ‘all SVs’. If the authors are correct that the ER-binding association is mechanistic rather than selective/functional, then the ‘all translocations’ track should be superimposed on the ‘SVs at amplification boundaries’.

Thank you for the opportunity to comment on this interesting paper!

Signed,
Peter Campbell

Referee #3 (Remarks to the Author):

It is long known that estrogens are carcinogenic (e.g. the higher E2 levels, the higher the chance of getting breast cancer; also the higher the E2 levels, the faster breast cancers progress) but experimental evidence for a mechanistic model has not been clearly provided. Also the sharp demarcation of the ERBB2 and CNND1 amplicons have been noted before but also no explanation

for the occurrence of these sharp demarkations have been put forward. By repurposing whole genome sequencing data from the public domain data, this study puts forward an interesting novel hypothesis that in ER positive breast cancer chromosomal breaks at ER binding sites is an early event targeting several commonly and focally amplified breast cancer genes and frequently involves one or multiple genes to amplify along with each other. And the fact that focal amplification subsequently involves multiple rounds of chromosomal translocation bridge fusion (TBF) formations which are selected for during breast cancer evolution has not been suggested to this extent before. The evidence is provided only by statistical associations using various sorts of genomic (WGS of ICGC cases) and epigenomics data (ChIP-seq; HiC, 3C, etc) from the public domain and careful examination of individual cases showing evidence for the proposed phenotypes. This manuscript puts forward a novel and interesting hypothesis with potential clinical implications for hormonal treatment of breast cancer being driven by this phenotype.

However, various issues remain or do not seem resolved. They are ordered in a linear fashion.

In line 105, “We found co-occurrence of focal hot spots” is mentioned. Since focal hotspots are frequently observed in a subtype-specific manner, it is not surprising to find these to co-occur if an entire breast cancer cohort comprised of multiple subtypes is analyzed. Do the authors still observe co-occurrence of focal hotspots, if the analysis is done by ER subtype. And for this paper it is of concern which of co-occurrences are observed in ER positive disease.

Line 124. “These translocations defining the amplicon boundaries therefore appear to have preceded the generation of the amplicons, strongly favoring the second scenario”. However, the fact that translocation bridge fusions are near clonal is no conclusive evidence for the fact that they are involving the initial event. It may very well be they have been heavily selected for (e.g. through a clonal sweep). Is there evidence against this latter possibility? In any case the observation does support a strong selection pressure on the event.

Line 140 states. “showing a significantly larger fraction of segments affected by loss-of-heterozygosity (LOH) compared to other chromosomes”. It is unclear how this comparison was performed because it may not be surprising to have more LOH in chromosomal unstable regions compared to regions that are stable. A more fair comparison would be to compare amplicons known to be driven by TBF versus those that are not (e.g. those driven by breakage-fusion-bridge (BFB) cycles or those in chromothrypsic regions). In any case it is unclear how the comparison was done and therefore it remains unclear whether this observation is in support of their conclusion. Later the author correctly state that, the observation of the existence of TBF is not entirely novel. In fact this observation was observed by others in this same dataset already. Thus, the authors reinvent the wheel, however, they do show the magnitude of the phenotype which has in my view not been recognized before. And they also that TBF occurs in various other cancers but whether this also driven by dominant transcription is not studied.

One argument against the model: We and also others have observed that the SHANK2 locus and the adjacent TMEM4 locus is indeed frequently rearranged but that the translocations are reciprocal or at least the SHANK2 (and TMEM4) locus can be donor (5' prime fusion partner) and acceptor (3' prime fusion partner) of a translocation.

(<https://www.biorxiv.org/content/10.1101/2021.05.17.441778v1>)

Eventhough chromosomal breakage after ER binding may probably be indifferent whether the fusion occurs with the locus being donor or acceptor of the oncogenic event, I am not sure these observations fit the TBF model presented in figure 2f. At least I wonder whether the current analysis also revealed what was observed earlier and if not why not. And second if it was observed whether telemetric LOH observed is dependent on whether its fusion involved a locus as acceptor or as donor. One of the options seems to me less likely fitting the current model so I would like to understand how the previous findings fit the current model.

One other point which is not clear: Mostly inter-chromosomal bridges are discussed. I do not see why the model would favors inter-chromosomal bridge over intra-chromosomal bridges. Or are inter-chromosomal brigde also observed. In any case in prostate cancer the most frequent translocation (TMPRSS1-ERG) is intra-chromosomal and is considered to occur in an androgen dependent manner. Of course whether the TMPRSS1-ERG fusion or any other fusion in prostate cancer (strongly driven by AR) concerns the same mechanism is as far as I know unknown but I could not find a reason for the phenotype discussed in the paper not to occur intra-chromosomally. Can the authors comment?

In line 242, 2 alternative hypotheses are proposed to explain rearrangements (topoisomerase II-mediated breakage (REF6) and the formation of unstable DNA/RNA hybrids (R-loops) (REF7). SHANK2 and TMEM4 generate long transcripts and expressed in breast cancer and may be involved in R-loop, I wonder whether the authors can provide evidence against these 2 earlier proposed models are can these co-exist with the current model?

Line 260 states "These findings suggest a widespread ER-associated fragility in breast cancer genomes. The DNA segments harboring breast oncogenes are likely selected for focal amplifications. Even though the work indeed suggest that estrogen induced activation is required evidence for this in this dataset is not provided.

However, the HER2 amplicon is present in breast cancer with ER positive and in ER negative disease. Assuming both types of cancer are equally well drive by HER2 and ER negative-HER2 positive disease being derived from a none-luminal progenitor, comparison of ER negative versus ER positive disease is a perfect case-control study for the observations in the HER2 locus. Thus, if sufficient WGS data are available in current or other data sources, and if the hypothesis that ER binding to ER response elements is right, in ER negative disease the boundaries of the HER2 amplicon would not be confined to the regions identified here. Has this comparison been done or is sufficient data lacking?

Another thing related to ER activity and the TBF phenotype: ER presence and ER activity are not fully correlated in ER-positive breast cancer. In the proposed model in the paper, ER activity rather than mere presence of ER would be driving the event. Within ER-positive disease further support of the model would be that tumors with an active ER would favor the phenotype to occur over ER positive BC in which ER is inactive. Since gene expression data for part of this cohort are available and ER activity can be inferred from those data, would it not allow the authors to find additional support for this hypothesi in the studied tumors. E.g. are these translocation fusion bridges involving an ER binding site more frequently seen in ER-positive tumors in which ER is (more) active?

Line 286. Hi-C data detects chromosomal contact sites, however, it also detect rearrangements. T47D has a highly rearranged genome. Thus, whether HiC in T47D is most suitable to determine if regions are in close proximity is questionable (Extended Data Fig. 7a) unless results were corrected for rearrangement present in T47D. Alternatively Hi-C data from another breast cancer cell line model with a (more) stable or at least less rearranged genome is preferably used for the proposed comparison.

In line 294, it is stated that the 20 most recurrent chromosome arm pairs exhibit significantly higher interaction frequencies after E2 stimulation. I wonder whether the original data (REF 39) were corrected for copy number gain as these severe over-estimated interactions over regions not having rearrangements in the studied cell line model. The question thus remains whether after this correction the significance of this observation remains.

One final question out of curiosity. Is ER just involved in the causing the breaks or does it subsequently facilitate selection of the event. Do the authors have data in support of the latter are can they eliminate the latter based on specific observations. This difference may have clinical impact with regard of the treatment of these cancers.

John Martens

Author Rebuttals to Initial Comments:

Responses to the Reviews

We are genuinely grateful to the reviewers for taking time from their busy schedule to provide extensive and thoughtful comments. This manuscript has benefitted tremendously from the questions and insights by the reviewers.

Referee #1 (Remarks to the Author):

This manuscript describes a computational study of a proposed genomic mechanism by which cancer genomes acquire recurrent genomic amplifications of oncogenes through combined translocation and amplification. They term this mechanism translocation-bridge amplification and associate its occurrence with estrogen receptor binding of DNA at breakpoint loci. They study published datasets of cancer genomes from the ICGC project to describe such genomic events, first in breast cancer and then in the ICGC pan-cancer dataset of multiple cancer types. Overall this is an interesting manuscript, but in its given shape it remains largely speculative and correlative and in-depth computational and experimental work would strengthen the study. some comments and questions below:

We appreciate this constructive comment. In response, we have performed experimental validation and more in-depth bioinformatic analysis that have considerably strengthened our model.

1. Several key oncogenes are listed to involve such amplification events such as ERBB2, CCND1, and others. Can the authors demonstrate novel findings of some new oncogenes and acquire evidence of their role in cancer? Amplified genes should be expected to be upregulated in corresponding cancer samples, so mRNA analysis might support such driver genes. Functional validation of novel genes in cell lines or other models would strengthen the study.

The main purpose of our study is to provide a new mechanism of how oncogene amplifications occur, and we have used the well-known genes to illustrate the significance and pervasiveness of the model. As such, finding new oncogenes was not one of our aims.

Nonetheless, to address the reviewer's question, we have carried out integrative analysis of WGS, mRNA expression profile, and CRISPR screen datasets to identify several previously-uncharacterized genes that may play oncogenic roles in breast cancer. Given that we have amassed a large collection of 780 breast cancer genomes from multiple studies for the revision, we had more power to detect new oncogenes. Although this is an important question in cancer genomics, we had to prioritize the findings directly supporting the main goal in our main text. We thus placed these findings in the **Supplementary Note**.

Briefly, we performed a more detailed analysis of the WGS datasets focusing on the frequently amplified regions. We integrated the mRNA expression profile (available in 263 breast cancer cases) from the Sanger study by Nik-Zainal *et al.* (*Nature* 2016; PMID 27135926) with their paired WGS. Findings were validated using the gene expression information from the METABRIC cohort (*Nature* 2012; PMID 22522925), of which copy number and mRNA expression datasets are available in 1,904 diploid tumors. Finally, we integrated CRISPR screen data from the DepMap project (*Cell* 2017; PMID 28753430) to study the impact of genes on cellular proliferation.

Identification of significantly amplified genes: We used GISTIC 2.0 (v2.0.23) to call significantly amplified genomic segments in 780 breast cancer whole genomes. GISTIC identified 29 amplification peaks and 24 deletion valleys (**Supplementary Note Fig. 2a**). Oncogenes and tumor suppressors are frequently located in the peaks and valleys. After filtering and merging the adjacent peaks, we focused on the nine most recurrent amplification peaks (**Supplementary Note Fig. 2b**). We analyzed these peaks by piling up the amplified regions ($\geq 4x$ of the estimated ploidy for each tumor) to minimize the impact of the tumors showing extreme copy number amplitude of the focal amplicons. In this way, we were able to find the genes at the very top of the amplification peaks. Key findings are summarized below.

Supplementary Note Fig. 2

- Selection of multiple genes in the focal amplicons:** We found that some peak regions have only one peak at the top (e.g., *ERBB2*), but others have multiple peaks. For example, the prominent peak region on 11q13.3 cytoband has ‘double peaks’, of which the higher peak encompasses *CCND1*, and the lower peak targets a less-known gene in breast cancer, *FADD*. The 17q23 amplicon, which has been frequently observed in

hormone receptor-positive breast cancers, has a broad copy-number elevation stretching through a megabase region (57.9-58.9 Mbp) with triple peaks. The highest peak encompasses an uncharacterized gene, *APPBP2*. This gene is in the neighborhood of *PPM1D*, an oncogenic protein phosphatase frequently altered in myeloid tumors and several other types. The other two peaks include *MIR21* and surrounding two long genes (*VMP1* and *TUBD1*) and *USP32*. The 8p11.23 peak region also showed the double-peak pattern. The higher peak has *ZNF703*, and the lower peak harbors the other two putative oncogenes, *NSD3* and *FGFR1*.

- Uncharacterized genes:** Some target genes of the focal amplicons were largely unknown in terms of their role in breast cancer. *FADD* in 11q13.3 is a good example, and there are several genes, including *QRSL1*, *MTRES1*, and *BEND3*, targeted by the 6q21 amplicon. Their mRNA expression and the functional impact of their disruption are addressed in the following sections. We also confirm that the genes suggested as putative breast cancer oncogenes based on microarray analysis, including *RSF1*, *PAK1*, and *TRPS1*, are indeed the targets of focal amplifications in breast cancer.

Supplementary Note Fig. 3

mRNA expression of the significantly amplified genes: For a total of 263 cases with paired WGS and RNA seq datasets from the Sanger study (PMID 27135926), we studied the mRNA expression of the frequently amplified genes in association with their amplification status. We validated the findings in the METABRIC cohort (PMID 22522925), using their diploid samples (n=1,904) to minimize the transcriptional impact of whole-genome duplication. Most of the genes showed robust mRNA overexpression when they are focally amplified (**Supplementary Note Fig. 3a**). This tendency was concordant between the Sanger and METABRIC cohorts. Notable exceptions are *TRPS1* and *MYC*, whose mRNA expression ranges largely overlapped

between the amplified and unamplified tumors in both cohorts. However, both genes demonstrated their essentiality in the CRISPR screen (**Supplementary Note Fig. 3b**; the results are discussed in the following section).

Knockout phenotype of the significantly amplified genes: To study the functional importance of the amplified genes, we integrated CRISPR screen data from the DepMap project (**Supplementary Note Fig. 3b**). A total of 46 breast cancer cell lines were profiled in the project, and 41 were available for ER/HER2-based classification. We used the gene effect score as the readout for cellular dependence on the given gene. This score was calculated by the Chronos algorithm (*Genome Biol* 2021; PMID 34930405) as part of the DepMap project. Value 0 indicates no viability effect on cells by knockout of the gene, whereas -1 indicates the median cytotoxic effect observed by knockout of common essential genes. This dataset reproduced the functional dependence of known oncogenes in the relevant molecular subsets. For example, knockout of the *ERBB2* showed selective cytotoxicity in HER2+ cell lines, in contrast to the modest effects in HER2- cell lines. Similarly, knockout of the *CCND1* showed more profound cytotoxicity in ER+ cells than ER- cells. CRISPR knockout of the *FADD* showed moderate cytotoxicity in ER+ cells, the subtype where *FADD* was frequently amplified. *QRS1* knockout also showed moderate cytotoxicity in breast cancer cells, in contrast to the other co-amplified genes, including *MTRES1* and *BEND3*. As previously proposed as a transcriptional regulator in luminal breast cancers (*Cell Rep* 2018; PMID 30380416), *TRPS1* knockout showed selective cytotoxicity in ER+/HER2- cells.

We do not think the lack of cytotoxicity in the CRISPR knockout screen challenges the functional importance of the amplified genes. DepMap is designed to study the gene's impact on cellular proliferation and survival, but not other hallmarks of cancer. For example, it is not surprising that the chromatin modifier genes (*RSF1* and *NSD3*) or the genes regulating cytoskeleton (*PAK1*) showed minimal cytotoxicity. These genes have been previously validated for their function and are frequent targets of focal amplification in our analysis.

- *RSF1* is at the top of the 11q14.1 peak. Overexpression of *RSF1* in the transformed mammary epithelial cells and associated xenograft models showed proliferative and invasive phenotype (PMID 25433701).
- *NSD3* is necessary for activating oncogenic NOTCH signaling in breast cancer through its role in H3K36 methylation (PMID 32967925).
- *PAK1* is essential in transforming breast epithelial cells by *ERBB2* via the beta-catenin pathway (PMID 23576562). 11 out of 45 *PAK1*-amplified breast cancers in our analysis also have *ERBB2* amplification.

With these results, we have proposed *FADD* and *QRSL1* as potential new oncogenes for breast cancer. In addition, we confirmed that *RSF1*, *NSD3*, *PAK1*, and others were frequent targets of focal amplification in breast cancer, further supporting the previous studies proposing them as potential oncogenes in breast cancer.

2. The link with estrogen binding and DNA breakage is only associative and is based on DNA-binding profiles a few regulatory proteins in common cell lines. Breakpoint sites could be bound by any number of other factors, the epigenomes of primary human cancers may differ from those of cell lines and normal tissues, and DNA binding does not imply causation (although causation is implied repeatedly in the manuscript). Functional and mechanistic experiments to explore causal effects of DNA breaks and estrogen exposure would substantially strengthen the manuscript, since this part is currently only suggestive.

The reviewer is correct to note that we have implied causation between ER α binding and DNA breakage in our previous version of the manuscript. However, it was not simply based on correlation, and we correlated the finding to the previous experimental observations of estrogen-induced DNA double-strand breaks (DSBs). For example, Stork *et al.* showed that E2 stimulation causes R-loop formation and subsequent DNA double-strand breaks (DSBs) dependent on DNA replication (*eLife* 2016; PMID 27552054). Ju *et al.* reported DSBs in early-responsive estrogen target genes in a topoisomerase II beta-dependent manner (*Science* 2006; PMID 16794079).

We now provide direct experimental evidence of estrogen-induced DNA double-strand breaks (DSBs) in the genome-wide ER α binding hotspots and their repair through chromosomal translocations. We performed high-throughput genome-wide translocation sequencing (HTGTS), a method that allows us an unbiased detection of the DNA double-strand breaks (DSBs) and their repair by the chromosomal translocations (*Cell* 2011; PMID 21962511). This work was performed by the laboratory of Roberto Chiarle, who developed this method while he was in Fred Alt's lab.

We are delighted to report that the experiments worked well, giving the results that we'd expected given our inference on estrogen binding as the initial event that induces DNA DSBs and their repair by translocations. We confirmed **1) a global increase of translocation frequencies by the estradiol (E2) treatment, 2) strong correlations between the HTGTS breakpoints and the ER α binding after the E2 treatment (E2-ER α) and/or topoisomerase 2B binding, and 3) the significant enrichment of the estrogen-responsive genes among the hotspots of HTGTS breakpoints.** These translocations include those that could form a dicentric chromosome between the two oncogene neighborhoods. They could provide the initial step for the TB amplification. The details are below; these findings are described in the **Main Text, Fig. 3, and Extended Data Figs. 7 and 8.**

Experimental design: Two ER⁺ breast cancer cell lines, MCF7 and T47D, were used in our experiments. These two cell lines were selected because they are the most extensively characterized ER⁺ cell lines in the literature, mount robust transcriptional responses to E2 treatment, and have a relatively short doubling time (24–36 hours). We also considered profiling normal-like mammary epithelial cells. However, the existing models, including MCF10A and HMEC, do not express the ER α , and the primary mammary epithelial cells are known to rapidly lose their ER expression *in vitro* (PNAS 2019; PMID 31110002), which precludes their use. Since the telomeric boundaries of *ERBB2* and *CCND1* amplifications have been frequently observed in *RARA* and *SHANK2* loci and their neighborhoods, we induced programmed DNA DSBs in intron 10 of *SHANK2* or intron 1 of *RARA* using the lentiCRISPR/Cas9 system in separate experiments (**Fig. 3d**). Thirty hours after the transduction, we strictly depleted estrogen in the culture system using phenol red-free and charcoal-stripped FBS (the culture protocol was adopted from PMID 31110002). In this setting, the control group was cultured with daily 0.01% ethanol treatment for four days (–E2), and the experimental group was treated with daily estradiol for four days (+E2). We confirmed an increased expression of well-known estrogen-responsive genes (*TFF1*, *GREB1*, and *PGR*) in response to the E2 treatment by quantitative RT-PCR (**Extended Data Fig. 7b**). On day 6, cells were collected and subjected to HTGTS (three biological replicates per experiment; 12 libraries for each target). Identical amounts of genomic DNA were used to prepare each HTGTS library. The primary analysis of the HTGTS dataset was performed as previously described (*Cell* 2011; PMID 21962511 and *Nature* 2017; PMID 28199309) to detect the breakpoints (referred to as ‘junctions’ in the previous publications) to which the programmed DNA DSBs were translocated.

E2-induced DNA breaks in the ER α binding hotspots and their repair by translocations: We found increased translocation breakpoints by the E2 treatment in all four experimental pairs (MCF7/*SHANK2*, T47D/*SHANK2*, MCF7/*RARA*, and T47D/*RARA*; **Extended Data Fig. 7c**). This finding indicates that the E2 treatment increased the frequency of DNA double-strand breaks (DSBs), which were repaired by the translocations. In both control and E2-treated experiments, the translocation breakpoints were enriched in the genic regions compared to the intergenic regions (**Extended Data Fig. 7d**; density ratio between the genic vs. intergenic regions = 2.37 in the control and 2.41 in the E2-treated). To investigate the mechanisms underlying the E2-induced translocations, we modeled the genome-wide ratio of the HTGTS breakpoints (E2-treated/control) using multivariate LASSO regression. Various epigenomic features that have been associated with DNA breaks and rearrangements were used as independent variables (**Fig. 3f**). Notably, the best predictor of the E2/control ratio of HTGTS breakpoint was ER α binding after the E2 treatment (E2-ER α), confirming that the E2-induced DNA breaks occurred at the vicinity of E2-ER α binding regions, and likely mediated by the binding of ER α to the chromatin. This finding is in line with the previous experimental observations of estrogen-induced fragility of its transcriptional targets (PMID 16794079 and

27552054). These results confirm the causal relationship between the E2 treatment, DNA breaks adjacent to the E2-ER α binding, and their repair by the translocations. Last, we performed a gene set enrichment analysis (GSEA) to study the gene sets enriched in the E2-induced HTGTS hotspots. A merged analysis of the four experimental pairs indicated significant enrichment of the early and late estrogen-responsive genes among the top genes (**Fig. 3g**; enrichment score of 0.55 and 0.50, FDR q-values of 0.001 and 0.022).

Fig 3.

Extended Data Fig. 7

Genome-wide fragility by the E2-ER α and subsequent selection: Many of the experimentally validated estrogen target genes were among the genes most often translocated. For example, *GREB1* (ranked at 1st) and *ITPK1* (4th) showed massive E2-induced HTGTS translocations (**Supplementary Note Fig. 5c, d**). These translocations were concentrated near the E2-ER α peaks, indicating the role of E2-ER α in inducing the DNA DSBs. Similarly, *RARA*, another estrogen target gene, showed >3-fold increase of HTGTS translocations with the induced breaks in *SHANK2* by the E2 treatment (394th; **Extended Data Fig. 8a**). *RARA* neighborhood is one of the most prominent SV hotspots in the breast cancer genome and frequently provides the initial translocation starting the *ERBB2* amplification. Although *GREB1*, *ITPK1*, and *RARA* are all frequently translocated in the experiment due to the E2-induced, ER α -mediated fragility, only *RARA* translocations were frequently found in the cancer genomes. We speculate that this is primarily because of functional selection. Translocations in *RARA* could lead to the TB

amplification cascade. Although the bridge breakage and massive rearrangement may come as a cost, amplification of the oncogenes would confer a selective advantage. In contrast, translocation of *GREB1* or *ITPK1* may not be beneficial to the cancer cells due to their loss of function or disruption of the region. Functional selection may explain some of the non-amplified SV hotspots as well. *GATA3* and *FOXA1* are important transcriptional regulators in ER-associated breast oncogenesis, and their mutations have been reported as driver events in multiple cancer types (*Nature* 2019; PMID 31243372 and *Nature* 2012; 22722193; **Extended Data Fig. 8c, d**). Although they are not canonical estrogen-responsive genes, they have E2-ER α peaks nearby. Our experiment showed a modest increase of HTGTS translocations in their neighborhood by the E2 treatment, and they precisely overlapped with the SV hotspots (unamplified) in breast cancer, suggesting E2-induced, ER α -mediated fragility as the potential mechanism of *GATA3* or *FOXA1* rearrangements.

3. Estrogen exposure is likely to have sex specific effects in mutagenesis. While these are effects likely clearer in breast cancer that is predominantly a cancer of female patients, they are less obvious in other cancer types and remain unexplored in the pan-cancer analysis in the latter part of the manuscript. Are there significant sex differences in the occurrences of these genomic events in the various cancer types they consider? are numbers of chromosomal events associated with patient age which may also contribute to estrogen levels? How about overall or progression-free survival or any molecular features of cancers, such as the expression levels or mutations of ER?

To address this, we performed subgroup analyses by patient's sex, integrated *ESR1* mRNA expression, and performed a pan-cancer meta-analysis while excluding the tumor types in sexual organs. The results are described below and in the **Main Text**, **Extended Data Fig. 10d-g**, and **Supplementary Note**.

More frequent boundary translocations in the female patients: We examined if the pattern of the boundary SVs was associated with the patient's sex in our pan-cancer analysis. In a merged analysis excluding cancers in the breast, ovary, endometrium, uterine cervix, and prostate, the focally amplified regions in female patients tend to have more frequent translocations at their boundaries compared to those in male patients, but the difference was marginal (odds ratio: 1.14, 95% confidence interval: 1.01–1.29; **Extended Data Fig. 10d**). Subgroup analyses by cancer type were primarily limited by the small number of patients in each type, but we observed a tendency for more frequent boundary translocations among the female patients in the tumor types with frequent boundary translocations (Group 4 tumors in **Fig. 5b**; odds ratios: 1.18 for liposarcoma, 2.01 for osteosarcoma, 1.26 for melanoma, and 1.12 for lung adenocarcinoma) and a couple more tumor types (1.63 for stomach and 1.13 for urinary bladder), although osteosarcoma was the only one reaching to the level of statistical significance ($p=0.04$).

Extended Data Fig. 10

Factors correlated with TB amplification: In our breast cancer cohort, the TB amplifications were not associated with younger or older ages (mean age of the group with TB amplifications vs. without; 56 vs. 55, $p=0.306$ by two-sample t test). In the PCAWG cohort, we also found that the presence or magnitude of TB amplification was not associated with patient age in any cancer type (**Extended Data Fig. 10f**), and also not associated with the expression level of *ESR1* (**Extended Data Fig. 10d**) after adjusting multiple testing (cut off of FDR <0.1). We performed survival analyses for those tumor types with ≥ 50 patients with available survival information. In four tumor types (esophageal adenocarcinoma, ovarian cancer, pancreatic adenocarcinoma, and cutaneous melanoma), we had ≥ 10 tumors showing TB amplification. No difference in overall survival was observed in these tumor types, between the cases with and without TB amplifications. In gastric cancers, it appeared that the TB amplification was associated with a worse survival outcome (median, 366 vs. 2197 days; $p=0.03$), but this analysis was very much limited by the small number of patients in each group (total $n=32$, 5 of them showed TB amplification).

Mechanistic interpretation of the pan-cancer landscape: Our analysis of breast cancer genomes indicated estrogen-induced fragility of the ER binding regions, and we confirmed it in our HTGTS experiments. Estrogen receptor binding-induced fragility explains the landscape of genomic rearrangements in ER+ breast cancers. The genomic rearrangement landscape of ER- breast cancer follows a more usual rule previously described in other cancer types – the fragility of the early-replicating, transcribed, and open chromatin regions (*Nature* 2020; PMID 32025012 and *Genome Res* 2013; PMID 23124520). Many ER target genes and oncogenes play vital roles

in normal physiology (e.g., *CCND1*, *RARA*, etc.) in other cell types and would likely be actively transcribed upon various cellular stimuli, which may confer their fragility (*Cell* 2017; PMID 28187286). Previous experimental studies also suggested that other nuclear receptors (androgen receptor, retinoid X receptor, AP-1, etc.) could also confer local fragilities, probably through similar mechanisms (*Science* 2006; PMID 16794079 and *Nat Genet* 2010; PMID 20601956). In many tissue types, the neighborhood of *CCND1* may be fragile regardless of estrogen or androgen, and their rearrangements can initiate focal amplification. In ER+ breast cancers, this feature is more pronounced due to its unique physiology dependent on estrogen.

Interestingly, we find slightly different usage of the *CCND1* neighborhood between cancer types in developing focal amplifications (**Extended Data Fig. 10i**). In many cancer types, including breast, head and neck, and esophageal cancers, *SHANK2* frequently has the initial break that starts the amplification process. A majority of breast cancer cases follow the TB amplification pathway, while head and neck cancers frequently follow the chromatid-type BFB pathway for amplifying *CCND1* (**Extended Data Fig. 10b**). In contrast, in hepatocellular carcinomas, the immediate downstream region of the *CCND1* is more frequently used as the boundary of the amplicon, implying different genomic fragility of the oncogene neighborhood depending on the tissue types.

As a minor note, the use of a male cancer sample as a prominent example was somewhat surprising due to the relative infrequency of those compared to female cancers.

We believe that the male breast cancer case also well supports our estrogen-induced fragility model because there has been a large body of literature supporting the critical role of estrogen in the development of male breast cancer (most notably, *J Clin Oncol* 2015; PMID 25964249). However, we appreciate this reviewer's comment and reduced emphasis on this case. This case is no longer presented in the main figures (moved to **Extended Data Fig. 3b**).

4. The authors have combined ICGC data of breast cancer and PCAWG data of breast cancer and other cancer types into a joint analysis. Although this helps increase power of the analysis, the inconsistent processing of the datasets may cause technical artefacts to become more prominent. Also, detection of structural variations is challenging and in especially for complex events and translocations. the ICGC PCAWG consortium has produced consensus structural variation calls and the approach used by the authors could be validated by examining those PCAWG calls summarised from multiple variant callers. Complementary validation using an independent cohort of whole cancer genomes would also help strengthen the manuscript, since the breast cancer data analysed in the first part

of the study is likely partially represented in the second part of the study of PCAWG genomes.

As the Reviewer points out, accurate detection of SVs forming complex genomic rearrangement events, especially in association with copy number variations (CNVs), is a key task in this study. In the previous version of the manuscript, our SV analysis was either based on the PCAWG consensus calls (n=206) or our in-house ensemble calls (n=72, non-PCAWG samples). Then, we performed copy-number re-segmentation by integrating the SV information to better calibrate the copy-number boundaries. Although our approach showed good performance in most cases, we found cases showing incomplete matching between the boundary SVs and the amplified segments in visual inspection. To improve this, we implemented Hartwig Medical Foundation (HMF) bioinformatic pipeline, which jointly analyzes SVs and CNVs. As described below, we found that the variant calls from our HMF-based pipeline were mostly comparable to the PCAWG consensus calls, but the HMF calls provided better consistency between the SVs and the border of amplicons, which was instrumental in the mechanistic analysis of amplicon formation. Our group has considerable expertise in development of CNV and SV algorithms (e.g., BIC-seq for CNV detection described in Xi *et al.*, *PNAS* 2011; Meerkat for SV detection described in Yang *et al.*, *Cell* 2013) and were part of the PCAWG SV working group, but we were impressed by the HMF results. We increased the size of our whole-genome analysis from 278 to 780 cases and uniformly processed them in the new pipeline.

Integrative calling of copy number and structural variants: In our initial manuscript, we had utilized an ensemble calling strategy based on three different SV callers (SvABA, Delly, and BRASS; selected SVs supported by two or more callers) followed by a copy number junction analysis (Jabba assisted by in-house copy number quantification) in French ICGC tumors (n=72). However, our manual inspection of the sequencing data showed some cases where the copy number junction was not well supported by the SVs. Some cases were simply due to the limited sensitivity of the SV detection. The ensemble calling strategy often missed true positive SV if it was called by only one caller. In addition, some copy-number boundaries were found bordered by an SV of which another breakpoint cannot be mapped uniquely due to their location in the repeat elements (often called ‘loose end’ or ‘single breakend’ in the recent publications, e.g., Cameron *et al.* *Genome Biol* 2021; PMID 34253237). GRIDSS2 is a new SV calling algorithm, which showed excellent performance in a recent study comparing 10 SV callers (*Nat Commun* 2019; PMID 31324872) and has strength in calling the SVs with single breakend. This tool is recently packaged as part of the Hartwig Medical Foundation (HMF) bioinformatics pipeline, along with an SV-aware copy number caller (PURPLE) and a complex SV annotator (LINX), which drew our attention.

After a rigorous benchmark analysis (see below) against PCAWG consensus SV, copy number, SNV, and indel calls, we felt that the HMF pipeline was at least as good as the PCAWG pipeline

in terms of the variant calls and superior for concordant analysis of SVs and CNVs. Therefore, we uniformly processed the expanded cohort through this pipeline. Thanks to this feature, we were able to clearly distinguish the SVs demarcating the amplified region, as this step is critical in understanding the mechanisms of complex rearrangements.

Benchmark analysis vs. PCAWG consensus SVs: To evaluate the performance of our HMF-based pipeline compared to the PCAWG consensus calls, we first analyzed the 208 PCAWG breast cancer genomes. Our new HMF-based pipeline recovered 39,442 out of 49,404 PCAWG consensus SVs in these samples (recovery rate = 79.8%; **Supplementary Note Fig. 1a**). A median SV recovery rate per sample was 84.8%. When we look into the SVs that were missed by our pipeline but were called by the PCAWG consensus pipeline, they include a large number of short-length non-reciprocal inversion concentrated on several samples (e.g., TCGA-A2-A04P, TCGA-AO-A0J4, and TCGA-A8-A07I), that are likely to be false positives. Our pipeline also identified 14,640 SVs not supported by PCAWG consensus calls, but a large fraction of them (n=5,947; 40.6%) were small SVs (≤ 100 bp), which are not called as SVs in the PCAWG pipeline (where the SV was defined as >100 bp). Some of the key SVs that were missed in the PCAWG consensus were called by our pipeline. For example, the translocation connecting the left borders of the *ERBB2* and *RSF1/PAK1* amplicons in the TCGA-A1-A0SM tumor was discovered by the new pipeline (in the initial submission, we found this translocation by manual review) but missed by PCAWG. We also compared the SV calls by our new pipeline and our previous ensemble calls on the French breast cancer cohort (n=72). Our new pipeline recovered 17,925 SVs from 19,759 SVs used in our initial submission (recovery rate = 90.7%). Regarding other classes of genomic variants, ploidy and purity estimates by PURPLE were also largely concordant with PCAWG consensus estimates (**Supplementary Note Fig. 1b, c**), with 177 (86%) out of 206 tumors showing minimal difference (<0.5) between the two pipelines. For SNV and indel analyses, we applied SAGE, the HMF software for small variant analysis, with adjusted detection parameters for 30X cancer whole-genomes. In 208 samples, our pipeline recovered 1,237,872 of 1,447,162 SNVs (recovery rate = 85.5%; **Supplementary Note Fig. 1d**) and 82,391 of 92,784 indels (88.8%; **Supplementary Note Fig. 1e**). In summary, our new pipeline demonstrated recovery of 80% SVs, 86% SNVs, and 89% indels in the PCAWG cohort and we applied this pipeline to the newly expanded cohort.

Expansion of the cohort: As we waited for the experimental work to be performed, we decided to take the opportunity to nearly triple the sample size for our paper. In addition to responding to the reviewer's request for validation in another cohort, we felt that analysis of the large cohort of breast cancer would give more accurate estimates of all inferences and make the manuscript stronger.

In addition to the 278 genomes in the initial submission (from the PCAWG cohort and French ICGC study by Ferrari et al. – *Nat Commun* 2016; PMID 27406316), we downloaded 403 cases

with matched tumor-normal genomes uniquely used in the Sanger study by Nik-Zainal *et al.* (*Nature* 2016; PMID 27135926), 87 cases from the British Columbia Personalized OncoGenomics study by Zhao *et al.* (*Clin Cancer Res* 2017; PMID 29246904), and 20 cases from the Yale inflammatory breast cancer study by Li *et al.* (*Genome Med* 2021; PMID 33902690). All of these sequence files have been deposited to the European Genome-Phenome Archive (EGA), and we downloaded the files through the EGA download client. Along with the 278 genomes that were used in the initial submission, we processed a total of 790 tumor-normal pairs in our new pipeline and finally included 780 breast cancer cases that were successfully processed and passed the quality control step (8 cases were excluded due to the issues with the sequence file, and 2 cases were excluded because of failure in QC step, more details in **Methods**). For the British Columbia study, we reached out to the authors to obtain more detailed clinicopathologic information and received hormone receptor and HER2 status that were not available in the published manuscript. The clinical characteristics (**Extended Data Fig. 1a-c**), summarized genomic alteration profile, and major drivers (**Extended Data Fig. 2d**) are illustrated in detail.

5. Is there evidence of other somatic mutations occurring at recurrent translocation/amplification hotspots, such as SNVs or indels that would indicate additional details of the mutagenic process if they were significantly co-occurring with structural variants, or perhaps additional driver mutations if significantly mutually exclusive with structural variants across cancer patients?

In summary, we identify no notable differences in the patterns of the SNVs and indels in the vicinity of amplicon boundaries compared to those around the internal SVs within the amplicons. The boundary SVs showed minimal or no microhomologies, indicating their formation by non-homologous end-joining. In a correlative analysis with molecular features or driver genetic alterations, TB amplification was enriched in homologous recombination-proficient tumors and rarely overlapped with *PTEN* deletion. These findings were implemented in the **Extended Data Figs. 1, 2, and Supplementary Note**. More details are available below.

SNVs and indels at the vicinity of boundary hotspots: We studied the SNVs in the vicinity of boundary translocations (we only used the SNVs proximal to the translocation breakpoint, considering their orientation) compared to those near the SVs inside of the amplicons (“internal SVs”; **Supplementary Note Fig. 4a-d**). Only 18% (73 out of 245) of the tumors with TB amplifications showed SNVs within the 100-bp window from boundary translocations, but this fraction goes up to 40% and 60% in 1-Kbp and 10-Kbp windows, respectively. Mutational signature analysis showed largely similar spectra of mutations in both regions, with a significant contribution of SBS2 and SBS13, two APOBEC3-associated mutational signatures (**Supplementary Note Fig. 4b, c**). Their contribution reaches the highest fraction at the 1-Kbp

window, then is down-trended in larger windows due to the contribution of other mutational signatures, including SBS5. We also analyzed indels near the boundary and internal SVs (**Supplementary Note Fig. 4d**), but the number of indels in the vicinity of the boundary translocations was insufficient to perform a reliable analysis. For example, only 16 indels were found in the 1-Kbp window, 48 in 10-Kbp, and 187 in 100-Kbp. The 100-Kbp window analysis shows no noticeable differences between the spectra of indels close to the boundary SVs and the internal SVs. Last, the boundary SVs tend to have shorter microhomology at their break ends compared to the internal SVs (1.39 vs 1.70, $p < 1 \times 10^{-7}$ by two-sample t test). This is consistent with our TB amplification model starting from the joining of two DNA DSBs by non-homologous end-joining.

Supplementary Note Fig. 4

Driver correlation: To study the correlation between the TB amplifications and other driver genetic alterations, we selected 151 tumors well-representing the features of TB amplification based on stringent criteria (ten or more translocations between the bridge arms). We also collected driver genetic events through the PURPLE pipeline, including 6140 small-scale mutations, 1630 gene amplifications, 506 gene deletions, and 74 germline mutations. We tested if any driver events are significantly more frequent or depleted among the 151 breast cancers with strong features of TB amplification. Notably, we found a significant depletion of *PTEN* deletion among the tumors with TB amplifications (OR=0.15, $p=0.0017$ by Fisher's test, FDR=0.048). Especially, the inactivation of *PTEN* (either by mutation or deletion) is mutually exclusive with *ERBB2* amplification (OR=0.24, $p=0.0004$; **Extended Data Fig. 2d**) and marginally with *CCND1* amplification (OR=0.41, $p=0.035$). Similar trend of mutual exclusiveness between the *ERBB2* amplification and *PTEN* inactivation is also observed in the METABRIC ($n=2173$; OR=0.35, $p=0.0003$) and the MSK-IMPACT datasets (*Cancer Cell* 2018; PMID 30205045; $n=1918$; OR=0.41, $p=0.0024$), suggesting its biological relevance.

Referee #2 (Remarks to the Author):

The authors report an intriguing analysis and model of the evolution of high-level amplification in cancers – predominantly breast cancer, but with pan-cancer extension. The mechanistic basis of spiky high-level amplifications involving 2+ chromosomes in cancer genomes has remained unexplained since they were first described in the days of copy number arrays. The major insight in the paper is the proposal that these events are initiated by a translocation between two chromosomes that generates a dicentric chromosome – then, with aberrant spindle attachment, a chromosome bridge is generated at mitosis, leading to chromothripsis and ecDNA formation. Subsequently, under selective pressure, the ecDNA (carrying several oncogenes) amplifies and further rearranges, generating the final structures observed in the cancer genome. The model is certainly credible and interesting – I have the following (rather verbose) comments:

We appreciate Dr. Campbell's constructive and insightful comments, which we believe have led to a much-improved version of our manuscript.

1. Other potential models for the evolution of such amplifications – The critical test for the model is whether it is more credible or fits the data better than other possible explanations. The major alternative hypothesis is that these events are initiated by chromothripsis involving two or more whole chromosomes (ie, no initiating translocation). As shown in previous in vitro work from the authors, multiple lagging chromosomes in the same mitosis can wind up captured in the same micronucleus and undergo co-rearrangement. It was not clear to me what features of the data meant that a single translocation had to have initiated the crises observed here. The authors provide strong evidence that interchromosomal rearrangements are heavily amplified and demarcate regions of major amplification – as they say, this means that these rearrangements preceded the amplification. However, in nearly all of the examples shown in Extended Figure 3, there are multiple heavily amplified interchromosomal rearrangements. There may indeed be one event that is more heavily amplified than others but this does not necessarily mean that it occurred earlier – it could just be that it's closer to the target oncogene and therefore under more amplification pressure within the ecDNA. Indeed, probably the critical determinant of timing such rearrangements is the copy number on either side of breakpoint junctions. That is, a breakpoint junction that demarcates moderate-amplification from non-amplification would likely have happened earlier than one that demarcates minor-amplification from massive-amplification even though the latter rearrangement is present in more copies per cell. It is hard to tell from the low-resolution Ext Fig 3 that I could access, but it seemed like multiple interchromosomal events demarcated amplified from non-amplified regions in most patients.

This would arguably be more consistent with the simpler model of chromothripsis as the initiating event.

We appreciate this insightful comment. We agree with the comment that it is important to consider alternative explanations and determine whether they could be eliminated given the data. First, we described reasons why the observed patterns of complex genomic rearrangement support the model starting from a translocation and bridge breakage rather than the scenario of multi-chromosomal micronucleation (**Main Text** and **Extended Data Fig. 4**). Then, we would like to address the question of whether the amplification boundary was bordering a moderately-amplified region (rather than a non-amplified one). In short, yes, there were some that border a moderately-amplified region, but the vast majority did not. Now we modified our algorithm and excluded those bordering moderately-amplified regions.

Evidence favoring bridge model over multi-chromosomal micronucleation: The best evidence supporting the chromosome bridge-based model is the ‘asymmetric’ involvement of the chromosome arms. The footprint (multiple SVs) of TB amplification is heavily concentrated on the two arms on the chromosome bridge (we refer to them as ‘bridge arms’ in our revised manuscript), and the other two arms (‘non-bridge arms’) are largely spared. This pattern is different from the typical genomic footprint that we expect for micronucleated chromosomes (*Nature* 2015; PMID 26017310 – an example in the figure below), where there are multiple DNA double-strand breaks in the entire length of the chromosome followed by their subsequent rearrangements in random order. In our newly expanded cohort, we identified 1,556 complex genomic rearrangement clusters (10 or more SVs involved) in 780 breast cancers. Among these, 295 clusters (from 245 breast cancer cases) involve multiple chromosomes and contain boundary translocations, suggesting a potential TB amplification process. On average, these clusters contain 137 SVs (range = 10–1515), including 3.75 boundary translocations (range 1–33). In most of these clusters (242 out of 295; 82%), bridge arms harbor $\geq 70\%$ of their SVs (**Extended Data Fig. 4b**). In contrast, rearrangements of the non-bridge arms are relatively rare – the non-bridge arms have no SVs in 123 (42%) out of 295 clusters, and contain $< 20\%$ of the SVs in $\sim 79\%$ of the clusters (232 of 295), highlighting the asymmetric distribution of the SVs. Translocations between the two non-bridge arms, which we would expect from multi-chromosomal micronucleation and chromothripsis, were rarely observed (only in 7% of the clusters; 21 out of 295). Accordingly, the bridge arms show a higher density of rearrangements compared to non-bridge arms (median 1.74 vs. 0.33 SVs per Mbp; $p < 1 \times 10^{-15}$; **Extended Data Fig. 4c**), and this trend is still significant after excluding all the SVs in the amplified regions (median 0.75 vs. 0.26 SVs per Mbp; $p < 1 \times 10^{-12}$). The bridge arms are also subject to a more extensive loss-of-heterozygosity compared to non-bridge arms (median 49% vs 25% of the arm; $p < 1 \times 10^{-14}$; **Extended Data Fig. 4d**). All these features indicate the selective damage of the bridge arms and their subsequent repair between the broken segments, consistent with the TB

amplification model. We also included example clusters that are more likely explained by whole-chromosomal micronucleation (multi-chromosomal chromothripsis in **Extended Data Fig. 4e**).

Extended Data Fig. 4

Multiple amplified genomic segments in the bridge arms: We previously reported a fragmentation of the chromosomal bridge after its breakage, of which the resultant copy number oscillated between two states due to the asymmetric segregation of the broken DNA fragments into the daughter cells, similar to chromothripsis (*Science* 2020; PMID 32299917). Because of this local fragmentation, discrete segments on the bridge arms can be ligated together and subsequently amplified. Furthermore, the broken hemi-dicentrics and DNA fragments could be micronucleated in the subsequent cell cycle, as observed in our prior experimental work, providing another chance of fragmentation. These explain the multiple amplified segments found in the typical cases of TB amplification. In addition to these, we oftentimes see cases whose initial rearrangements forming the dicentric chromosome would have been complex translocations reminiscent of chromoplexy, adding further complexity to the footprint of TB amplification (e.g., **Fig. 4d**, **Extended Data Fig. 3a**, **4f**, and **9**). So, the reviewer is correct to note that in some instances the TB amplification contains the features expected in a chromothripsis event. But we hope that the clear asymmetry between the bridge and non-bridge arms presented above demonstrates clear support for our model.

Boundary SVs bordering between the amplified and unamplified regions: To answer the question systematically, we modified our algorithm to detect only the SVs at the boundaries demarcating an amplified region from unamplified regions (reflected throughout the **Main Text**

and **Figs. 1, 2**) while excluding the SVs at the border between extremely amplified and mildly amplified segments. We defined unamplified regions as those segments at or below the baseline copy number of the residing chromosome arm. In addition, we also included the segments whose copy number is greater by one copy from the baseline because the bridge resolution process often causes retention of the same genomic segments from sister chromatids (**Fig. 2f** and associated examples in **Extended Data Fig. 5a**). This new algorithm identified 11,490 discrete, amplified genomic regions in 780 breast cancer genomes (median: six regions per tumor). From these amplified regions, we identified 4,889 boundary SVs at the junction of amplified and unamplified segments. We described the detailed features of the boundary SVs in **Extended Data Fig. 1d-h**. Briefly, the boundary SVs are supported by a large number of reads, 25% of them are inter-chromosomal, and the intra-chromosomal ones are usually large (~10 Mbp) SVs or fold-back inversions (~1 Kbp). They tend to have shorter microhomologies at their break ends, usually located in the early-replicating segments, and rarely overlapped with the common fragile sites (0.256% vs. 0.889%, $p < 1 \times 10^{-15}$ by Fisher's exact test).

2. Timing from point mutations – It would be interesting to include an analysis of somatic point mutations within the amplified region, and their ploidy. This will give a sense of the mutation burden the ancestral cell would have had when the amplification event occurred. For example, if there are no / very few point mutations present at high levels of ploidy, then it suggests that the amplification is a critical early event in the evolution of the cancer.

We analyzed the timing of copy number gains based on the burden of pre-amplified mutations. This approach was used in our previous paper describing the timing of fusion oncogene copy-number gain in lung cancer (*Cell* 2019; PMID 31155235). In that paper, we estimated the latest timing of fusion oncogene formation, by analyzing the timing of copy-number gain of the chromosome harboring the fusion oncogene. Because the fusion oncogene was formed before the copy-number gain, the timing of copy-number gain provided the latest possible timing of fusion oncogene formation. In a similar way, here we studied the latest timing of chromosome bridge breakage from the cases where the non-bridge arms were amplified. In summary, the timing of bridge breakage appeared to be variable among the cases available for this analysis, but generally happened earlier than whole-genome duplication or gain of 20q. Some common aneuploidy events, e.g., gain of 1q or isochromosome 16p formation, often preceded the gain of non-bridge arms. Assuming that somatic mutations accumulate linearly in early oncogenesis (especially when they are in pre-transformed cells), the timing of non-bridge arm gains roughly matches with the median age of menopause, suggesting that the formation of chromosome bridge would be pre-menopausal events in many cases. Now, the result is described as a new paragraph in the **Main Text**, associated **Fig. 4**, and **Extended Data Fig. 9**. More details are available below.

Fig. 4

Extended Data Fig. 9

Timing the TB amplifications from the copy-gain of non-bridge arms: Because of the small size of the focal amplicons and the challenge of estimating mutational copies in heavily rearranged, highly amplified segments, direct timing of focal amplifications was impossible. Therefore, we took an indirect approach to measure the timing of the neighboring large segments that were amplified after the bridge breakage. In PD6044, for example (Fig. 4a), the large parts of 8q and 11p were amplified likely after the bridge breakage. These segments were the non-bridge arms when the two sister dicentrics formed chromosome bridge, were separated from the adjacent focal amplicon after the bridge breakage, likely ligated to each other, stabilized, and subsequently amplified (here, mutual ligation of the two hemi-dicentrics is likely given their coordinate amplification to the same copy number). Since the bridge breakage happened earlier than the copy-gain of the 8q and 11p, the timing of 8q-11p co-amplification could give us the clue of latest possible timing of the bridge breakage. Similarly, in PD14450 (Extended Data Fig. 9a), the non-bridge arm 1q was amplified after the bridge breakage, when it was separated from the focal amplicons. Therefore, the timing of 1q gain provides us the latest possible timing of bridge breakage.

In this way, we analyzed the timing of hemi-dicentric amplification as the latest possible timing of bridge breakage. We directly quantified the number of mutations amplified up to the maximal major copy number of the hemi-dicentric region in a similar approach that we took in our previous work (Cell 2019; PMID 31155235; Fig. 4a and Methods). Among the 295 complex genomic rearrangement clusters harboring boundary translocations (used in Extended Data Fig

4b), 69 cases showed copy-number gain in the non-bridge arm bordered by the focal amplicon. These cases allowed us to probe the latest possible timing of the bridge breakage (**Fig. 4b**). Furthermore, we estimated mutational burden of the ancestral cells when they had acquired common aneuploidies, including WGD (estimated from the regions with bi-allelic gains; 241 tumors were available for this analysis), the gain of 1q (n=268), 8q (165), and 20q (52). We also identified that the 16q loss, one of the most common arm-level losses in breast cancers, was paired with an arm-level gain of 16p in many cases (n=108), suggesting isochromosome 16p formation as the primary mechanism for 16q loss. In these cases, we were able to estimate the timing of 16q loss using 16p gain as the surrogate.

In this analysis, we found that 1q gain and 16p gain paired with 16q loss are usually earlier events than others (confirmed by Wilcoxon's test; **Fig. 4b**), consistent with previous cytogenetic observation (*Genes Chromosomes Cancer* 2015; PMID 25546585). These happened when the ancestral cell had a mutational burden of 0.34, and 0.40 per Mbp per genome, respectively. Gain of 8q (0.59/Mbp), WGD (0.70), and gain of 20q (0.80) were generally later than the gains of 1q or 16p. The median timing of non-bridge arm gain in select tumors with TB amplification (n=69) was 0.52 per Mbp per genome, which is significantly later than 1q gain but earlier than WGD or 20q gain (**Fig. 4b**). Consistent with this result, most cases with TB amplification as well as WGD showed dual-LOH pattern in their bridge arms, indicating that the WGD happened later than the bridge breakage in these cases (e.g., PD6044 in **Fig 4a** and BR014 in **Extended Data Fig. 5a** are whole genome-duplicated; ploidy: 4.05 and 3.20, respectively). Taken together, our analysis showed the order between the common aneuploidies and chromosome bridge formation and breakage, which often happened earlier than WGD.

TB amplification in woman's life: Based on the assumption of gradual accumulation of somatic mutations in the early oncogenesis, we correlated the timing of major aneuploidy events and non-bridge arm gains to the time period of human life. The clonal mutation burden increased with age at a rate of 29.4 mutations per year in our select cases (n=147; defined by high purity, no hypermutation, no whole-genome duplication, and HR proficient; **Extended Data Fig. 9b** and **Methods**). Based on this rate, latest median timing of chromosome bridge breakage corresponds to 50.8 years of age, which is about five years earlier than the median age of breast cancer diagnosis in our cohort (56 years). The latest timing of bridge breakage roughly matches with the median menopausal age of 52.5 years reported from the large observational cohort study in the United States by Gold et al. (*Am J Epidemiol* 2013; PMID 23788671), suggesting that many bridge breakage likely happened during reproductive ages. This finding is in agreement with our experimental findings of E2-induced, ER α -mediated inter-chromosomal translocations and their contribution to TB amplifications.

3. *Correlation of breakpoint locations with ER binding – These analyses are notoriously difficult because many features of the genome are co-correlated. For example, replication timing, gene density, active histone marks (and presumably oestrogen receptor binding?) are all positively correlated with one another across the genome. This makes it difficult to be certain which factor best explains density of somatic mutations / breakpoints, especially with univariate analyses as performed here. A multivariate analysis should be performed (although is non-trivial due to spatial auto-correlation effects).*

We already described above experimental evidence of estrogen receptor-associated DNA breaks and translocations that trigger the TB amplification cascade (Response to reviewer 1’s comment 2). Here, as suggested, we analyzed the amplification boundary hotspots using a multivariate statistical model with several other DNA-binding proteins and epigenetic factors. We primarily used the multivariate LASSO regression model, which is more tolerant to the multicollinearity between the variables. **As detailed below, this analysis demonstrated that the E2-ER α was the dominant predictor of the amplification boundaries.** We supplemented our conclusion with a multivariate linear mixed-effect model while acknowledging the moderate degree of multicollinearity. Again, E2-ER α stood out as the dominant contributor. The new statistical models are described in the **Main Text, Fig. 3, and Extended Data Fig. 6.**

Fig. 3

Extended Data Fig. 6

Multivariate LASSO regression model: Analysis of all breast cancer cases (n=780) showed that the E2-ER α was the most significantly associated variable with the amplification boundaries (**Fig. 3a**). Notably, this association was only significant in the ER+ breast cancers but not in the ER- cases (the first row of Fig. 3a shows p-value of $\sim 10^{-7}$ for ER+ cases [red bar] and close 1 for ER- [pink bar is almost absent]). We found that the lack of association in the latter subgroup was largely driven by the ER-, HR-deficient tumors (e.g., *BRCA1* mutated tumors), which consisted 46% of the tumors (126 out of 271) in the ER- subgroup.

Collinearity between epigenetic variables: We assessed multicollinearity among the epigenetic variables by calculating the variance inflation factor (VIF), a standard method to quantify the collinearity between the dependent variables. Genomic features associated with open chromatin

regions, notably H3K4me3 (active promoter mark), H3K27ac (active enhancer mark), and DNase I hypersensitivity sites, showed high VIF values of 4.66, 3.87, and 3.79. The VIF of E2-treated ER α was 2.73, lower than the other markers. Given that VIF >5 is generally perceived as concerning and VIF >10 a serious collinearity problem (Menard 2001 Applied Logistic Regression Analysis, 2nd edition), the epigenetic variables in our analysis did not reach the level of serious concern (**Extended Data Fig. 6d**), although some recent literature suggested a more conservative cutoff of 2.5 (PMID 29937587). Our multivariate linear mixed-effect model confirmed the most significant contribution from E2-ER α , followed by H3K27ac, CTCF, and others (**Extended Data Fig. 6e**).

4. Effects of selection – The ER binding correlation with SV density here is complicated even further by the fact that the nearby genes are under positive selection in an ER+ breast cancer. That is, it could be that only genes with strong ER-mediated transcriptional control could be positively selected for in an ER-expressing breast cell and therefore SV breakpoints would be closer to ER-binding sites than random genome positions. An additional analysis that would help here is to explore the correlation between passenger translocation events (ie not under selection) and ER binding – similar to Ext Fig 5g/h, but with ‘all translocations’, not ‘all SVs’. If the authors are correct that the ER-binding association is mechanistic rather than selective/functional, then the ‘all translocations’ track should be superimposed on the ‘SVs at amplification boundaries’.

Thank you for the opportunity to comment on this interesting paper!

*Signed,
Peter Campbell*

Since we experimentally proved that the association between the rearrangement and the E2-ER α is indeed mechanistic/causal (Response to reviewer 1’s comment 2), we think that the fundamental question behind this suggestion is resolved.

We totally agree that the functional selection has major influence on the genomic footprint of the focal amplification. However, it appears that the source of selection would be the oncogenes, such as *ERBB2*, *CCND1*, or *ZNF703/FGFR1*, rather than the ER target genes in their neighborhood. In a number of examples, we found that the well-known estrogen-responsive genes were transected by the complex translocations that led to the *ERBB2* amplifications, indicating functional disruption of those ER target genes rather than their preservation (e.g., *TOP2A* is transected by the boundary SV in **Extended Data Fig. 3b**). In line with this, we also found that the overall expression of estrogen-responsive genes was lower rather than higher in the tumors with more extensive TB amplification, likely reflecting the disruption of the ER-

driven transcriptional program (Fig. 4d; more details are described in response to Reviewer 3's comment 9).

The latter analysis suggested by the reviewer, separating the passenger inter-chromosomal translocations from the other SVs in previous **Extended Data Figs. 5g, h** (current **Extended Data Figs. 6h, i**), was very challenging to perform. There was no clear way to filter out functionally selected events. Even among the SVs in the unamplified regions, we found multiple SV hotspots that could indicate their functional selection (**Extended Data Figs. 8c, d**). In addition, we think that intra-chromosomal SVs can also arise from the same mechanism of E2-associated, ER α -mediated fragility. Notably, many initial rearrangement events causing the dicentric formation were complex rearrangements involving both inter- and intra-chromosomal rearrangements (e.g., **Fig. 2c**). Although limited, we compared all inter-chromosomal SVs vs. intra-chromosomal SVs in terms of their overlap with the E2-ER α peaks, and this did not show any meaningful difference (relative density around the ER peaks = 1.67 for inter-chromosomal, 1.65 for intra-chromosomal SVs among the HR-proficient tumors).

More importantly, our updated analysis based on 780 breast cancer cases again confirmed the higher enrichment of the recurrent amplification boundaries in the E2-ER α peaks (**Extended Data Fig. 6i**; here, the hotspots were defined as 100-Kbp bins with >4 tumors). Compared to the genome-wide SVs that have originated from multiple different mutational processes, the early SVs, either inter- or intra-chromosomal ones, were more closely associated with ER-associated fragility.

Referee #3 (Remarks to the Author):

It is long known that estrogens are carcinogenic (e.g. the higher E2 levels, the higher the chance of getting breast cancer; also the higher the E2 levels, the faster breast cancers progress) but experimental evidence for a mechanistic model has not been clearly provided. Also the sharp demarcation of the ERBB2 and CNND1 amplicons have been noted before but also no explanation for the occurrence of these sharp demarcations have been put forward. By repurposing whole genome sequencing data from the public domain data, this study puts forward an interesting novel hypothesis that in ER positive breast cancer chromosomal breaks at ER binding sites is an early event targeting several commonly and focally amplified breast cancer genes and frequently involves one or multiple genes to amplify along with each other. And the fact that focal amplification subsequently involves multiple rounds of chromosomal translocation bridge fusion (TBF) formations which are selected for during breast cancer evolution has not been suggested to this extent before. The evidence is provided only by statistical associations using various sorts of genomic (WGS of ICGC cases) and epigenomics data (ChIP-seq; HiC, 3C, etc) from the public domain and careful examination of individual cases showing evidence for the proposed phenotypes. This manuscript puts forward a novel and interesting hypothesis with potential clinical implications for hormonal treatment of breast cancer being driven by this phenotype.

However, various issues remain or do not seem resolved. They are ordered in a linear fashion.

We are delighted that Dr. Martens appreciates the importance of providing a mechanistic model for puzzling observation in the field and for his many comments and questions below.

1. In line 105, “We found co-occurrence of focal hot spots” is mentioned. Since focal hotspots are frequently observed in a subtype-specific manner, it is not surprising to find these to co-occur if an entire breast cancer cohort comprised of multiple subtypes is analyzed. Do the authors still observe co-occurrence of focal hotspots, if the analysis is done by ER subtype. And for this paper it is of concern which of co-occurrences are observed in ER positive disease.

Co-amplification of two oncogenes (from different chromosomes) that are connected by translocations is not only observed in the subgroup level but also, more importantly, in the individual case level. Below we describe the findings from the subgroup analysis (described in **Extended Data Fig. 2a-c**).

Extended Data Fig. 2

Hotspots by the ER and the homologous recombination status: We compared ER⁺ vs. ER⁻ and homologous recombination (HR)-proficient vs. HR-deficient tumors to further characterize the TB amplification process. Co-occurrence of focal amplifications through TB amplification was frequent in homologous recombination-proficient breast cancers (222 out of 593 HR-proficient tumors). In contrast, TB amplification was rare in the HR-deficient subgroup (37% vs. 13%, $p < 1 \times 10^{-10}$ by Fisher's exact test). Among the HR-proficient tumors, TB amplification is found in both ER⁺ and ER⁻ tumors at a similar frequency (36% and 41%, respectively). We do not find any notable differences in the pattern of rearrangements or copy number alterations, such as the presence of LOH in both bridge arms, indicating common mechanistic background. Instead, some oncogenes were amplified in subgroup-specific ways. *CCND1* was more frequently amplified in ER⁺/HR-proficient subgroup, whereas *CCNE1* was more frequent in ER⁻/HR-proficient subgroup. We also compared the pattern of *HER2* amplification between the ER⁺ vs. ER⁻ subsets (more details will be discussed in response to comment 8).

2. Line 124. "These translocations defining the amplicon boundaries therefore appear to have preceded the generation of the amplicons, strongly favoring the second scenario". However, the fact that translocation bridge fusions are near clonal is no conclusive evidence for the fact that they are involving the initial event. It may very well be they have been heavily selected for (e.g. through a clonal sweep). Is there evidence against this latter possibility? In any case the observation does support a strong selection pressure on the event.

We are not making the early event claim based on their clonality. The amplified boundary SVs are ‘early clonal’ or even ‘very early clonal’ in the sense that they are amplified up to the maximal copy number of the amplicon. This indicates that the translocations were already present when the segment started to gain its copies. In our revised manuscript, we also made our criteria calling the TB amplifications more stringent, in which we selectively call the amplified boundary SVs only when they are flanked by non-amplified regions (following Reviewer 2’s comment 1). Through this approach, we filtered out some of the early clonal boundary SVs that were formed after the global copy number gains, so that we can capture the pattern of SVs that initiates the amplification process. We apologize if our previous description was confusing. In our revised main figure, we tried to explain the general concept of TB amplification in a schematic illustration, as well as to provide examples (**Fig. 2**).

3. Line 140 states. “showing a significantly larger fraction of segments affected by loss-of-heterozygosity (LOH) compared to other chromosomes”. It is unclear how this comparison was performed because it may not be surprising to have more LOH in chromosomal unstable regions compared to regions that are stable. A more fair comparison would be to compare amplicons known to be driven by TBF versus those that are not (e.g. those driven by breakage-fusion-bridge (BFB) cycles or those in chromothrypsic regions). In any case it is unclear how the comparison was done and therefore it remains unclear whether this observation is in support of their conclusion. Later the author correctly state that, the observation of the existance of TBF is not entirely novel. In fact this observation was observed by others in this same dataset already. Thus, the authors reinvent the wheel, however, they do show the magnitude of the phenotype which has in my view not been recognized before. And they also [showed] that TBF occurs in various other cancers but whether this [is] also driven by dominant transcription is not studied.

Here, we would like to explain why and how we compared the extent of LOH between the chromosomes in our previous version. However, we have modified our LOH section substantially in the new version (see our response to Reviewer 2’s comment 1), emphasizing the different extent of LOH between the bridge vs. non-bridge arms. Therefore, the part that the Reviewer was concerned is no longer existent in the manuscript.

In the initial version of the manuscript, we compared the length of the chromosomal segments affected by the LOH between the chromosomes containing the amplified translocations (that lead to the TB amplification process) vs. the chromosomes not associated with the amplified translocations (outside of the complex rearrangement clusters). This comparison was to support that the chromosomes involved in the TB amplifications (or bridge breakage) are frequently subject to chromothripsis (likely in the next cell cycle), which is consistent with previous experimental observation (PMID 32299917). We apologize if our original description was not

clear enough. For clearer mechanistic explanation, we changed our focus of our LOH analysis to highlighting the extensive LOH in the bridge arms, which is critical in inferring the timing of the initial translocation (in response to Reviewer 2's comment). Our model indicates that the telomeric parts of the two bridge arms were lost during mitosis because they formed acentric fragments (PMID 3733881), explaining the LOH. Since this finding supports our sequential model of 1) dicentric chromosome formation by translocation, 2) bridge breakage and 3) further rearrangements potentially involving chromothripsis, and the BFB is inherently coupled to chromothripsis (shown in the previous paper from our group by Umbreit *et al. Science* 2020; PMID 32299917), we do not think the comparison with BFB could be helpful in this context. Further demonstration of transcription- or other epigenetic factor-associated breaks that lead to TB amplification in other cancer types is a great suggestion. However, in contrast to breast cancer, we do not have extensive tissue type-matched epigenomic datasets in many other tissue types, which makes this beyond the scope of our study.

4. One argument against the model: We and also others have observed that the SHANK2 locus and the adjacent TMEM4 locus is indeed frequently rearranged but that the translocations are reciprocal or at least the SHANK2 (and TMEM4) locus can be donor (5' prime fusion partner) and acceptor (3' prime fusion partner) of a translocation. (<https://www.biorxiv.org/content/10.1101/2021.05.17.441778v1>) Even though chromosomal breakage after ER binding may probably be indifferent whether the fusion occurs with the locus being donor or acceptor of the oncogenic event, I am not sure these observations fit the TBF model presented in figure 2f. At least I wonder whether the current analysis also revealed what was observed earlier and if not why not. And second if it was observed whether telemetric LOH observed is dependent on whether its fusion involved a locus as acceptor or as donor. One of the options seems to me less likely fitting the current model so I would like to understand how the previous findings fit the current model.

We are excited to see this comment. The findings the Reviewer mentions is an important observation that actually agrees well with our findings on the fragility of *SHANK2* and *TENM4* regions. We agree with Dr. Martens that the translocation can happen in either + or – orientations after the DNA break so that two simultaneous DSBs in different chromosomes could form either two monocentric chromosomes or a pair of one dicentric chromosome and an acentric fragment.

We have actually done a more extensive analysis for the observation that the Reviewer makes regarding *SHANK2* or *TENM4* fusions. We performed a DNA-level fusion analysis in our expanded dataset and found many in-frame and out-of-frame fusions involving *SHANK2* or

TENM4 and other frequently observed genes at the boundaries of amplicons. We described the details in the **Supplementary Note**. Below we describe the details.

Fusion analysis shows DNA-level fusions of SHANK2, TENM4, and the E2-responsive genes:

We analyzed the fusion genes in our 780 breast cancers and identified 38,306 gene fusion events (including in-frame, out-of-frame, and exon-skipping events). Surprisingly, the second most commonly fused gene in our cohort was *SHANK2* (248 events in 93 tumors; 12% in our cohort; **Supplementary Note Fig. 5a**) and the fourth was *TENM4* (187 events in 62 tumors; 8% in the cohort). *BCAS3*, which is frequently observed in the right border or inside of the 17q23 amplicon, was ranked first by the number of events (284 events in 77 tumors; 10% of the cohort), but the number of tumors harboring its fusion was less than that of *SHANK2*. Overall, these findings are consistent with what was observed in the published study.

We also used TCGA PanCan Atlas database of gene-gene fusion transcripts (n=25,664 fusion events; PMID 29617662). In this dataset, we found 48 mRNA-level fusion events involving *SHANK2*. 35 (73%) of them were from breast cancer samples, and seven (15%) fusions were found in bladder cancers. *TENM4* fusion was reported in 11 cases, including 9 (82%) breast cancer cases, less frequent than in our dataset. We also found 44 *BCAS* fusions and 32 (73%) were from breast cancer. These findings also support that these genes are fragile and their DSBs are often repaired by rearrangements. If they are repaired in an orientation forming a dicentric chromosome and an acentric fragment, this could trigger the TB amplification.

Correlation between the large telomeric LOHs and the orientation of the boundary SVs: As suggested, we examined if the presence of large telomeric LOH is associated with the orientation of most distal (farthest from the centromere) boundary SVs. We analyzed 292 chromosome arms showing an obvious footprint of TB amplifications where the most distal boundary SVs in an orientation supporting the dicentric chromosome formation (+ breakpoint for the q arms and – breakpoint for the p arms). Among these chromosome arms, about half of them (53%) showed telomeric LOHs. The remaining cases without telomeric LOHs showed complex genomic rearrangements, often rearranging the distal portion elsewhere (e.g., **Extended Data Fig. 3b**, in TCGA-A1-A0SM, the telomeric part of 17q was translocated to chromosome 1q). In these cases, the initial rearrangement event is likely complex involving multiple DNA DSBs and often chainable, reminiscent of chromoplexy (**Fig. 2d** and **Extended Data Figs. 3b**). Taken together, not all the cases of TB amplification showed extensive telomeric LOH. Because the initial rearrangement event forming the dicentric chromosome is frequently complex, the telomeric segments distal to the translocations were often rearranged to elsewhere, preserving their heterozygosity.

5. One other point which is not clear: Mostly inter-chromosomal bridges are discussed. I do not see why the model would favor inter-chromosomal bridge over intra-chromosomal bridges. Or are inter-chromosomal bridge also observed. In any case in prostate cancer the most frequent translocation (TMPRSS1-ERG) is intra-chromosomal and is considered to occur in an androgen dependent manner. Of course whether the TMPRSS1-ERG fusion or any other fusion in prostate cancer (strongly driven by AR) concerns the same mechanism is as far as I know unknown but I could not find a reason for the phenotype discussed in the paper not to occur intra-chromosomally. Can the authors comment?

Just a reminder, a chromosome bridge is formed when two centromeres of a dicentric chromosome are pulled to the opposite poles. Because this needs a dicentric chromosome, the only way to form a chromosome bridge by intra-chromosomal rearrangements is the fusion between the sister chromatids. This is often detected as fold-back inversion, which is the hallmark of the classical, chromatid-type, breakage-fusion-bridge (BFB) cycle (Fig. 2e). In our manuscript, we are describing another way to generate a dicentric chromosome, which necessarily involves an inter-chromosomal translocation (Fig. 2f).

We sometimes see the BFB initiating the amplification process in breast cancer. In these cases, we find fold-back inversions at their boundaries that are flanked by non-amplified segments. However, the prevalence of the fold-back inversions initiating the amplification is substantially lower than the translocations in breast cancer (Fig. 5c). In some other cancer types, including head-and-neck or lung squamous cell carcinomas, the amplicons built up by fold-back inversions are more frequently observed. We highlighted these findings in our updated pan-cancer analysis (Fig. 5).

Fig. 2

TMPRSS2-ERG and comparison between the prostate and breast cancers: This has been one of our major motivations for studying the genomic rearrangements in breast cancer. We believe that the mechanisms of early complex translocations in breast cancers, often leading to TB amplification, are analogous to what was described in prostate cancers (PMID 19962179, 20601956, and 23622249). However, their consequential genomic footprints are very different. In prostate cancer, the *TMPRSS2-ERG* fusion oncogene is described as one of the earliest events and is more common among early-onset prostate cancers (PMID 23410972). The close relationship between the breakpoints of this fusion oncogene and the androgen receptor binding has been previously suggested (PMID 19962179 and 20601956). *TMPRSS2-ERG* fusion also frequently exists as part of the chromoplexy chain in almost always near-diploid background (WGD is observed in only 4.5% of prostate cancers in the PCAWG cohort; 9 out of 199). Presumably, in the cellular context of normal prostate epithelial cells, catastrophic rearrangements after the bridge breakage would not be tolerable and might be negatively selected during the evolution (we described a similar phenomenon in non-smoker lung cancer patients in our previous study; PMID 31155235). In contrast, a similar process rearranges the breast cancer genomes, leading to massive focal amplification of the oncogenes. In our evolutionary analysis, TB amplification happens in the ancestral cells already harboring more than a thousand mutations in the genome. It is typically preceded by other CNAs as well as presumably by some driver mutations, including *PIK3CA* or *AKT1* mutations, which have been reported from the stage of ductal columnar epithelial cells (PMID 22460814) and usual ductal hyperplasia (PMID 24186142 and 26718977), even before exhibiting any pathological atypia. In addition, the antagonistic role of the estrogen receptor toward p53 signaling has also been suggested (PMID 20696891 and 23077249), which may engender tolerance to those massive genomic rearrangements leading to focal amplifications. However, in this context, the chain-like footprint of chromoplexy is disrupted by additional rearrangements and can no longer be clearly identifiable. In conclusion, genomic rearrangement processes in breast and prostate cancers have similar features, notably associated with sex hormone receptors. However, their timing and cellular contexts lead to different genomic outcomes, one as massive focal amplification in the aneuploid background and another as a translocation chain forming a fusion oncogene in the near-diploid background. We described this in the Discussion section of the **Main Text**.

6. In line 242, 2 alternative hypotheses are proposed to explain rearrangements (topoisomerase II-mediated breakage (REF6) and the formation of unstable DNA/RNA hybrids (R-loops) (REF7). *SHANK2* and *TMEM4* generate long transcripts and expressed in breast cancer and may be involved in R-loop, I wonder whether the authors can provide evidence against these 2 earlier proposed models are [or?] can these co-exist with the current model?

Source of fragility in *SHANK2* and *TENM4*: Based on the previous report showing the R-loop formation in late estrogen-responsive genes and their frequent breakage (PMID 27552054), we studied whether R-loop could explain the hotspots of amplification boundaries or the E2-induced translocations in our HTGTS experiments. It appears that R-loop formation is associated with amplification boundary hotspots in our LASSO regression model. However, it does not explain the distribution of E2-induced translocation in our experiment. Our direct inspection of the *SHANK2* and *TENM4* loci in DNA-RNA immunoprecipitation followed by high-throughput sequencing dataset (DRIP-seq; from Stork et al. study in *eLife* 2016; PMID 27552054) showed no notable difference in R-loop formation before and after E2 treatment (**Supplementary Note Fig. 6**). More details are below.

Supplementary Note Fig. 6

R-loop formation in several genomic regions of interest
SHANK2 neighborhood

TENM4 neighborhood

In our LASSO regression model for the amplification boundary hotspots, R-loop was one of the significantly associated variables with the boundary hotspots ($p = 0.029$; **Fig. 3a**). This remained significant in the ER+ diseases but showed no association in the ER- diseases. However, the E2-induced translocations in our HTGTS experiments were not significantly associated with R-loop (**Fig. 3f**), even though our experimental time frame was long enough to induce R-loop formation in the late estrogen-responsive genes that were described in the previous study (PMID 27552054; R-loop formation was dramatically increased between 2 and 24 hours after the E2 treatment). Instead, the HTGTS translocations were strongly associated with two factors – E2-ERα and

topoisomerase 2B binding, which could be a more prevalent mechanism of E2-induced DSBs (PMID 16794079 and 31110352). Both *SHANK2* and *TENM4* loci showed a numerical increase of the HTGTS translocations by the E2 treatment, but the amounts of increase were not statistically different from the background increase (E2 treatment increased the HTGTS translocations in general, as described in the **Main Text**). This is somewhat contradictory to our fusion analysis result, showing that these two regions were among the top 3 fusion hotspot genes in our cohort. We interpret that this discrepancy may indicate either a strong selection of the low-frequency translocations due to their selective advantage from their neighboring oncogenes (*CCND1* and *RSF1/PAK1*), or there might be another important source of DNA damage in these genes in addition to the E2-induced, ER α -mediated DSBs. Although these two genes have prominent E2-ER α peaks in their long introns, they are not listed as estrogen-responsive genes in a curated gene ontology database (MSigDB; *Cell Syst* 2015; PMID 26771021).

7. Line 260 states “These findings suggest a widespread ER-associated fragility in breast cancer genomes. The DNA segments harboring breast oncogenes are likely selected for focal amplifications. Even though the work indeed suggest that estrogen induced activation is required [,] evidence for this in this dataset is not provided.

Experimental validation of E2-induced ER α -mediated fragility: In our previous version of the manuscript, we tried to provide bioinformatic evidence of ER-associated fragility by analyzing the global associations between ER binding peaks in the CHIP-seq data and the SV breakpoints. We reported the highest correlation with the amplification boundary hotspots but also found a significant association with the genome-wide SV breakpoints, supporting the ER-associated fragility. These findings were reproduced in our expanded cohort with a more rigorous statistical modeling. We also acknowledge the importance of direct experimental validation to make a definitive conclusion here. Our HTGTS experiments showed a robust induction of translocations by the E2 treatment between the ER target genes, including the ones we frequently observe at the boundaries of focal amplifications (please see the response to Reviewer 1’ comment 2). The details were described in the **Main Text, Fig. 3, Extended Data Fig. 7.**

8. However, the *HER2* amplicon is present in breast cancer with ER positive and in ER negative disease. Assuming both types of cancer are equally well drive by *HER2* and ER negative-*HER2* positive disease being derived from a none-luminal progenitor, comparison of ER negative versus ER positive disease is a perfect case-control study for the observations in the *HER2* locus. Thus, if sufficient WGS data are available in current or other data sources, and if the hypothesis that ER binding to ER response elements is right, in ER negative disease the boundaries of the *HER2* amplicon would not be confined to the regions identified here. Has this comparison been done or is sufficient data lacking?

We appreciate this comment. It has been a major debate in this field if the ER⁻ breast cancers originate from non-luminal progenitors, especially basal cells, or they are phenotypically drifted from the luminal to the non-luminal lineage through oncogenesis (PMID 26266985). In addition, within the HER2⁺ breast cancers, ER⁺ has been associated with a lower rate of pathologic complete remission (pCR) compared to the ER⁻ diseases in the neoadjuvant clinical trials with dual HER2 blockade therapy with trastuzumab and pertuzumab (PMID 22153890, 23704196, and 27831502). Therefore, the question here would be whether we could find any notable differences in the pattern of rearrangements in HER2⁺/ER⁺ vs. HER2⁺/ER⁻, which may be associated with their different clinical behavior. In summary, we found no noticeable difference in the pattern of copy-number amplifications and their boundary translocations between the ER⁺/HER2⁺ vs ER⁻/HER2⁺ diseases. However, different partners of co-amplification might be associated with their clinical features. The result is described in the **Extended Data Fig. 2b, c**.

Similar pattern of TB amplification between the ER⁺ vs. ER⁻, HR-proficient subgroups:

ERBB2-amplified breast cancers were largely homologous recombination-proficient (only 6 out of 119 were classified as HR-deficient; 5%; **Extended Data Fig. 2d**). Among the HR-proficient tumors, we do not find any notable differences in the pattern of the TB amplifications. Characteristic amplification of the boundary translocations, adjacent segments exhibiting a ‘dual LOH’ pattern, and the dense bundle of intra-amplicon rearrangements were observed in both ER⁺ and ER⁻ tumors. Apparently, the size of amplicons and the location of their telomeric boundaries were also not different between these two groups (**Extended Data Fig. 2b**).

Different partners of TB HER2 amplifications depending on the ER status: However, we found differences in co-amplified regions depending on the ER status of the HER2⁺ tumors. For example, co-amplification of the 17q23 region (harboring *PPM1D*, *MIR21*, *USP32*, etc.) was frequent in the HER2⁺/ER⁺ tumors but was infrequent in the HER2⁺/ER⁻ tumors (**Extended Data Fig. 2b**; available in page 25 of this document). This is consistent with the previous observation that the 17q23 amplicon often plays a role as regulatory machinery by providing ER-driven enhancers. Amplification of this region could be more beneficial for the triple-positive tumors than those with HER2 amplification without active ER signaling. In line with this, co-amplifications of *ZNF703* or *ZNF217* were also often observed in the HER2⁺/ER⁺ tumors but very rarely in the HER2⁺/ER⁻ tumors (**Extended Data Fig. 2d**). Interestingly, the amplification of 6q21 region (harboring *QRSL1*, *MTRES1*, and *BEND3* – discussed more in detail in response to reviewer 1’s comment 1) was the most common co-amplification in the HER2⁺/ER⁻ tumors (observed in 11%, but only in 3% of the HER2⁺/ER⁺ tumors). These co-amplicons are oftentimes directly connected with the *ERBB2* amplicons through TB amplification, or indirectly through a complex genomic rearrangement cluster, or exist independently without connections. Taken together, we found different partners of co-amplification among the *ERBB2*-amplified breast cancers depending on their ER status. Some of the genes that were frequently co-amplified in the HER2⁺/ER⁺ tumors through TB amplification were the putative oncogenes beneficial in

the context of active ER signaling, which may explain the dependence of HER2+/ER+ tumors on both the HER2 and ER signaling and their different clinical behavior.

9. Another thing related to ER activity and the TBF phenotype: ER presence and ER activity are not fully correlated in ER-positive breast cancer. In the proposed model in the paper, ER activity rather than mere presence of ER would be driving the event. Within ER-positive disease further support of the model would be that tumors with an active ER would favor the phenotype to occur over ER positive BC in which ER is inactive. Since gene expression data for part of this cohort are available and ER activity can be inferred from those data, would it not allow the authors to find additional support for this hypothesis in the studied tumors. E.g. are these translocation fusion bridges involving an ER binding site more frequently seen in ER-positive tumors in which ER is (more) active?

We appreciate this insightful comment. We agreed with the hypothesis that ER activity might be driving the TB amplification, developed a practical score of ER activity using the expression level of estrogen-responsive genes, and applied it to our cohort. To our surprise, it showed rather a slight but statistically significant anti-correlation with the extent of chromosome arms affected by the TB amplification. Our analysis also provides an alternative explanation of why the ER activity is not positively correlated with TB amplification. We described this in the **Main Text**, **Fig. 4d, e**, and **Extended Data Fig. 9e**.

Fig. 4

Extended Data Fig. 9

Quantification of ER transcriptome activity in the RNA seq data: We used the published RNA sequencing dataset from the Sanger study by Nik-Zainal *et al.* (*Nature* 2016; PMID 27135926 and *Nat Commun* 2016; PMID 27666519) to quantify the activity of the ER-driven transcriptome. We defined the ER target genes based on the Hallmark gene sets in the MSigDB (PMID 26771021), which we also used in analyzing the HTGTS dataset. A total of 275 genes are listed in early and late estrogen-responsive genes in the MSigDB. This list included well-known ER target genes that have been extensively used in experimental settings, including *GREB1*, *TFF1*, and *PGR*. First, we tested if these genes were differentially expressed between the ER+ and ER- tumors (n=188 and 69, respectively, with 6 ER-unknown cases). 136 of them showed a significantly higher level of mRNA expression in the ER+ group compared to the ER- group in our cohort. We used these 136 individual genes as probes of ER activity and scored each tumor if they had an expression level of 50 percentile or higher in our cohort. We tested different cutoff values for defining the ER activity score (10, 30, 50, and 70), and the score based on the 50-percentile cut showed the best-separated distribution between the ER+ and ER- cases and a good spread within the ER+ cases (**Fig. 4d**).

Negative correlation between the ER activity score and the TB amplification: We tested if the ER activity score is correlated with the number of chromosome arm pairs involved in TB amplification events in each tumor (this information is also used in **Fig. 2b** and **Extended Data Fig. 2d**). Linear regression model showed a mild but statistically significant negative correlation (slope -3.75 , $p=0.02$) between the ER transcriptome activity score and the number of chromosome arm pairs involved in TB amplification (**Fig. 4d**), indicating that the ER activity rather decreases in the tumors with more extensive TB amplifications. We checked individual genes that are the component of our score and found that known ER target genes were often indeed negatively correlated with the presence/magnitudes of TB amplifications. For example, *TFF1*, one of the most commonly used ER target genes as a readout of the ER-mediated transcription in experimental settings, showed a significant negative correlation with the TB amplification (slope -0.61 , intercept 4.80 , $p=0.001$), and we found similar cases for many other estrogen-responsive genes (we visualized examples considered statistically significant based on $FDR < 0.1$ in **Extended Data Fig. 9e**). In contrast to many of these genes, we found that a few were positively correlated with the TB amplification, and one of them was *ASCL1* (slope 0.94 , $p = 0.0003$; **Extended Data Fig. 9e**), the master transcriptional regulator in the pro-neural lineage, often playing an important role in neuroendocrine differentiation of various types of tumors (PMID 25267614). *ASCL1* is a known ER target gene in the context of neurogenesis (PMID 29246927), and its expression is upregulated by estrogen treatment in the MCF7 cells (AACR abstract; *Cancer Res* 2009;69(24 Suppl):Abstract nr 2153). Other downstream neuroendocrine genes did not show significant trends (*SYP*, *CHGA*, *NCAMI*, *ENO2*, and *CALCA*), although we see general uptrends in *SYP* and *CHGA* (slope 0.20 , $p=0.09$ and slope 0.25 , $p=0.22$, respectively). We interpret that this may indicate disruption of the estrogen receptor-driven transcriptional program by the massive genomic changes through the TB amplification (**Fig. 4e**).

Association with neuroendocrine differentiation needs further studies, but it has been frequently reported in breast cancer, and its clinical significance has been in question (PMID 33531618).

10. Line 286. Hi-C data detects chromosomal contact sites, however, it also detect rearrangements. T47D has a highly rearranged genome. Thus, whether HiC in T47D is most suitable to determine if regions are in close proximity is questionable (Extended Data Fig. 7a) unless results were corrected for rearrangement present in T47D. Alternatively Hi-C data from another breast cancer cell line model with a (more) stable or at least less rearranged genome is preferably used for the proposed comparison.

Given that we now provide direct experimental validation of E2-induced translocations, we have moved the three-dimensional analysis to the **Supplementary Note**. We also note that the larger fraction of ER⁻ or HR-deficient tumors in our new cohort also introduces additional complexity to this analysis, since these samples are less likely similar to the cell lines used.

Hi-C analysis: We agree with Dr. Martens' concern on the limitation of the cell line models. In our initial manuscript, we analyzed Hi-C data of MCF7 and T47D cells to analyze the physical proximity of the regions. We presented our data based on the T47D dataset because the genome of T47D is less complex than MCF7. Computational calibration of Hi-C contact information using the genomic rearrangements is an area of active development, but no established tools yet. Outside of this study, we recently analyzed the Hi-C data of the HMEC cell line, which remains a relatively stable, near-diploid genome. However, this cell line does not express the ER. We discussed this issue with our collaborators, and apparently there is no better generic model for this purpose in the breast cancer research community.

In our expanded cohort of 780 breast cancers, our previous findings showing higher contact frequency of the amplification boundary hotspots in T47D cell Hi-C data were not statistically significant. This is mainly because of the increased heterogeneity between the tumors and the inclusion of many HR-deficient and ER⁻ tumors. Furthermore, the T47D Hi-C dataset was generated in an experiment without E2 treatment, which may explain the weak correlation. 3C-based sequencing analysis with or without E2 showed more interactions between the chromosome arms that were commonly involved in TB amplification (discussed in the following comment), reproducing the conclusion in our previous version. The findings are now described in **Supplementary Note**.

11. In line 294, it is stated that the 20 most recurrent chromosome arm pairs exhibit significantly higher interaction frequencies after E2 stimulation. I wonder whether the original data (REF 39) were corrected for copy number gain as these severe over-estimated

interactions over regions not having rearrangements in the studied cell line model. The question thus remains whether after this correction the significance of this observation remains.

Copy number correction: Yes, we corrected the copy number effect in our initial analysis. Because we directly compared the E2-treated dataset against the control dataset, both generated from the MCF7 cells, the copy number-associated effect was canceled out. The list of most frequently translocated arms was slightly changed due to our cohort expansion. Our new analysis verifies that the frequently translocated arms are indeed more interactive with each other in the three-dimensional dataset (**Supplementary Note Fig. 7a**).

Supplementary Note Fig. 7

12. One final question out of curiosity. Is ER just involved in the causing the breaks or does it subsequently facilitate selection of the event. Do the authors have data in support of the latter or can they eliminate the latter based on specific observations. This difference may have clinical impact with regard of the treatment of these cancers.

John Martens

Functional impact of TB amplification: This issue partly overlaps with the previous comment regarding the ER activity. Although our analysis showed an unexpected negative correlation between the ER transcriptome activity score and the presence and magnitudes of TB amplifications, we think this does not nullify the hypothesis of ER activity-driven selection of TB amplification. The lack of correlation is rather an anticipated consequence of massive genomic rearrangements that lead to gene amplifications, disruption of the neighborhood, and potentially subsequent rewiring of the regulatory network. In addition to amplifying the oncogenes, the TB amplification could contribute to the alteration of the transcriptional program.

Reviewer Reports on the First Revision:

Referees' comments:

Referee #1 (Remarks to the Author):

The authors have addressed my comments and I have no further concerns. The manuscript appears even stronger with the experimental validation, additional computational analyses, and extensive supporting data. Congratulations!

Referee #2 (Remarks to the Author):

The authors have comprehensively addressed my comments, and I remain enthusiastic about its publication.

Referee #3 (Remarks to the Author):

No additional comments. My concerns have been properly addressed.